# GOLUM-CNP v1.0: a data-driven modeling of carbon, nitrogen and phosphorus cycles in major terrestrial biomes

Yilong Wang[1,*], Philippe Ciais[1], Daniel Goll[1], Yuanyuan Huang[1], Yiqi Luo[2,3,4], Ying-Ping Wang[5], A. Anthony Bloom[6], Grégoire Broquet[1], Jens Hartmann[7], Shushi Peng[8], Josep Penuelas[9,10], Shilong Piao[8,11], Jordi Sardans[9,10], Benjamin D. Stocker[10], Rong Wang[12], Sönke Zaehle[13], Sophie Zechmeister-Boltenstern[14]

[1] Laboratoire des Sciences du Climat et de l'Environnement, CEA-CNRS-UVSQ- Université Paris Saclay, Gif-sur-Yvette, France

[2] Center for Ecosystem Science and Society, Northern Arizona University, Flagstaff, AZ, USA

[3] Department of Earth System Science, Tsinghua University, Beijing, China

[4] Department of Microbiology and Plant Biology, University of Oklahoma, Norman, OK, USA

[5] CSIRO Oceans and Atmosphere, PMB #1, Aspendale, Victoria, Australia

[6] Jet Propulsion Laboratory, California Institute of Technology, Pasadena, CA, USA

[7] Institute for Geology, KlimaCampus, Universität Hamburg, Bundesstrasse 55, D-20146 Hamburg, Germany

[8] Sino-French Institute for Earth System Science, College of Urban and Environmental Sciences, Peking University, Beijing, China

[9] CSIC, Global Ecology Unit CREAF-CSIC-UAB, Bellaterra, Catalonia, Spain

[10] CREAF, Cerdanyola del Vallès, Catalonia, Spain

[11] Institute of Tibetan Plateau Research, Chinese Academy of Sciences, Beijing, China

[12] Department of Global Ecology, Carnegie Institution for Science, Stanford, CA, USA

[13] Max Planck Institute for Biogeochemistry, Jena, Germany

[14] University of Natural Resources and Life Sciences Vienna, Institute of Soil Research, Department of Forest and Soil Sciences, Vienna, Austria

*Corresponding to: Yilong Wang[1] (yilong.wang@lsce.ipsl.fr)

**Abstract.** Global terrestrial nitrogen (N) and phosphorus (P) cycles are coupled to the global carbon (C) cycle for net primary production (NPP), plant C allocation and decomposition of soil organic matter, but N and P have distinct pathways of inputs and losses. Current C-nutrient models exhibit large uncertainties in their estimates of pool sizes, fluxes and turnover rates of nutrients, due to a lack of consistent global data for evaluating the models. In this study, we present a new model-data fusion framework called Global Observation-based Land-ecosystems Utilization Model of Carbon, Nitrogen and Phosphorus (GOLUM-CNP) that combines the CARbon DAta MOdel fraMework (CARDAMOM) data-constrained C-cycle analysis with spatially explicit data-driven estimates of N and P inputs and losses and with observed stoichiometric ratios. We calculated the steady-state N- and P-pool sizes and fluxes globally for large biomes. Our study showed that new N inputs from biological fixation and deposition supplied >20% of total plant uptake in most forest ecosystems but accounted for smaller fractions in boreal forests and grasslands. New P inputs from atmospheric deposition and rock weathering supplied a much smaller fraction of total plant uptake than new N inputs, indicating the importance of internal P recycling within ecosystems to support plant growth. Nutrient-use efficiency, defined as the ratio of gross primary production (GPP) to plant nutrient uptake, were diagnosed from our model results and compared between biomes. Tropical forests had the lowest N-use efficiency and the highest P-use efficiency of the forest biomes. An analysis of sensitivity and uncertainty indicated that the NPP-allocation fractions to leaves, roots and wood contributed the most to the uncertainties in the estimates of nutrient-use efficiencies. Correcting for biases in NPP-allocation fractions produced more plausible gradients of N- and P-use efficiencies from tropical to boreal ecosystems and highlighted the critical role of accurate measurements of C allocation for understanding the N and P cycles.

## 1 Introduction

Nitrogen (N) and phosphorus (P) cycling are tightly coupled with the global carbon (C) cycle (Cleveland et al., 2013; Elser et al., 2007; Gruber and Galloway, 2008; Ver et al., 1999; Turner et al., 2018) in terrestrial ecosystems. N and P availability affects vegetation productivity, growth and other processes (Norby et al., 2010; Sutton et al., 2008; Vitousek and Howarth, 1991). N and P also affect soil C by nutrient controls on the mineralization of litter and soil organic matter (Gärdenäs et al., 2011; Melillo et al., 2011). Global vegetation models suggest that the coupling between the C, N and P cycles is among the major factors determining projected changes in the terrestrial C balance under scenarios of climate change and rising atmospheric $CO_2$, because additional productivity will only be realized if plants can increase their uptake or recycling of nutrients (Hungate et al., 2003; Sun et al., 2017; Wang and Houlton, 2009; Zaehle et al., 2015). Estimates of the magnitudes of these responses of ecosystems in the future, however, are highly uncertain (Peñuelas et al., 2013; Wieder et al., 2015).

Nutrients are important for understanding the current perturbation and future projections of the global C cycle, so several Dynamic Global Vegetation Models (DGVMs) have incorporated terrestrial N cycling (Goll et al., 2012; Medvigy et al., 2009; Parton et al., 2010; Thornton et al., 2007; Wang et al., 2001, 2010; Weng and Luo, 2008; Xu-Ri and Prentice, 2008; Yang et al., 2009; Zaehle et al., 2014; Zaehle and Friend, 2010). Fewer models have incorporated the cycling of P and its interactions with C dynamics (Goll et al., 2012, 2017a; Wang et al., 2010). Many of the underlying processes are not fully understood, and comprehensive data for evaluation are lacking to constrain the representation of some key processes (Zaehle et al., 2014), so model structure, the processes included and the prescribed parameters differ widely among DGVMs (Zaehle and Dalmonech, 2011). For example, some models assume constant stoichiometry (N:C and P:C ratios) in plant tissues (Thornton et al., 2007; Weng and Luo, 2008), while others have a flexible stoichiometry (Wang et al., 2010; Xu-Ri and Prentice, 2008; Yang et al., 2009; Zaehle and Friend, 2010). For the N cycle, for instance, some models do not include losses of gaseous N from denitrification (Medvigy et al., 2009), some use the "hole-in-the-pipe" approach to simulate the denitrification flux (Thornton et al., 2007; Wang et al., 2010), assuming it is proportional to net N mineralization, while others calculate this flux as a function of soil N-pool size and soil conditions (temperature, moisture, pH, etc.) (Parton et al., 2010; Xu-Ri and Prentice, 2008; Zaehle and Friend, 2010). For the P cycle, for instance, Jahnke (2000) estimated that the global total amount of soil P was 200 Pg and that the P contained in plants was 3 Pg, based on empirical P content of soils (0.1%) and soil thickness (60 cm). These estimates were questioned by recent studies from Wang et al. (2010) and Goll et al. (2012), who estimated that P in plants ranged between 0.23 and 0.39 Pg and that P in soil was only 26.5 Pg based on P:C ratios derived from more comprehensive stoichiometric data sets. Furthermore, terrestrial ecosystem models are usually only evaluated for specific ecosystems or at a limited number of sites (Goll et al., 2017a; Yang et al., 2014a). The application of these models for simulations with global coverage is thus highly uncertain (Goll et al., 2012; Wang et al., 2010; Zhang et al., 2011).

A growing number of data sets in recent decades have addressed many aspects of the nutrient cycles and their interactions with C dynamics. For example, Zechmeister-Boltenstern et al. (2015) synthesized the stoichiometry in different ecosystem compartments and highlighted the latitudinal gradients of plant, litter and soil stoichiometry. Liu et al. (2017) evaluated soil net N mineralization among different ecosystems at the global scale and found that net N mineralization decreased with increasing latitude. They also found that the N mineralization at higher latitudes is more sensitive to temperature changes than at lower latitudes, indicating potential alleviation of N limitation for plants' productivity at boreal regions under global warming. Yang et al. (2013) provided spatially explicit estimates of different forms of soil P globally and thus made it possible to assess the P content that is available for plant uptake. These data help to improve the understanding of the global terrestrial biogeochemical cycles across large climatic and ecological gradients and can in principle be combined to provide an integrated analysis of terrestrial C, N and P cycles. Estimates of C, N and P cycles consistent with all these data sets, however, have not yet been successfully provided due to the difficulties in combining

these data sets with different uncertainties and inconsistent spatial/temporal representations.
We present a new global data-driven diagnostic of C, N and P pools and fluxes, called GOLUM-CNP (Global
Observation-based Land-ecosystems Utilization Model of Carbon, Nitrogen and Phosphorus) which is based on the
assumption that these cycles are equilibrated with present day conditions (see below for limitations of this approach). The
goals of this study are to: 1) establish a global data-driven diagnostics of C, N and P fluxes and pools in order to compare
nutrient use efficiencies, nutrient turnover rates and other relevant indicators across biomes; and 2) provide a new dataset
that can be used to evaluate the results of global terrestrial biosphere models with consistent state of C, N and P cycles. In
GOLUM-CNP, the C, N and P cycles were estimated for different biomes assuming steady state with present-day input of
carbon (NPP), nitrogen (N deposition and N fixation) and phosphorus (P deposition and release from rock weathering) (see
Sect. 3.2). The reason for this steady-state computation lies in the fact that only few global long-term observations associated
with N and P cycles are available and are insufficient to constrain a transient simulation under the model framework. For
example, field-scale manipulation experiments have shown that warming, elevated atmospheric $CO_2$, and N and P
fertilization can drive changes in stoichiometry and nutrient resorptions (Sistla and Schimel, 2012; Sardans et al., 2012;
Sardans and Peñuelas, 2012; Mayor et al., 2014; Yang et al., 2014b; Yuan and Chen, 2015; Sardans et al., 2016; Sardans et
al., 2017) in terrestrial ecosystems, but these data are insufficient to infer these changes in terrestrial ecosystems during the
past decades. As more data becomes available, the model framework can be adjusted to simulate a transient present day state.
Although, the steady-state assumption hampers the comparison of stocks with present day observations, a direct comparison
with simulated steady states of DGVM is possible as these model can simulate the steady-state for present day conditions.
Starting from a CARbon DAta MOdel fraMework (CARDAMOM) data-constrained analysis of the terrestrial C cycle
(Bloom et al., 2016), which is based on the Data Assimilation Linked Ecosystem Carbon Model version two (DALEC2,
Bloom and Williams, 2015; Williams et al., 2005) and on observations of biomass, soil C, leaf area index (LAI) and fire
emissions, we incorporated observed stoichiometric ratios (C:N:P) in each pool, N and P external input fluxes,
transformations and losses in ecosystems and the fraction of gaseous losses of N to total (gaseous and leaching) losses of N
from a global dataset of $^{15}$N measurements in soils. Although the diagnostics is presented for steady state, the methods used
to compute fluxes and pools are generic and could be extended to non-steady state (see Sect. 2 and equations in Appendix A-
C) when more data will become available in the future (see Sect. 5.3).
We first present the model structure (Sect. 2) and the data sets used to derive its outputs consisting of pools, fluxes and
turnovers of C, N and P (Sect. 3). The model results and their sensitivities to the input observation-based data sets are then
further analyzed in Sect. 4. In Sect. 5, we show examples of the application of this sensitivity analysis to identify the major
differences in the results from our model framework and a synthesis of *in situ* measurements, and a qualitative example of
how to compare the model and the independent data. These differences identify critical observations to reduce uncertainty in
global C and nutrient cycling and highlight the future demand for model development, calibration and evaluation.
**2    Model structure**
GOLUM-CNP describes the C, N and P cycles in natural (i.e. non-agricultural) terrestrial ecosystems (Fig. 1). We used
the same C pools and fluxes as in the CARDAMOM diagnostic (see Sect. 2.1 for details) to describe the C cycle and we
computed associated N and P pools and fluxes. Biomass is divided into three pools: foliage, fine roots and wood. The wood
pool includes woody stems, branches and coarse roots. The litter pool in Fig. 1 corresponds to fine litter from leaves and fine
roots. Soil organic matter (SOM) receives C from fine litter and woody biomass. Two additional pools not present in
CARDAMOM are added, representing soil inorganic N and labile soil P. These inorganic N and labile soil P pools are
assumed to represent nutrients accessible by plants (see Sect. 2.1 and 2.2). Of note is that these inorganic N and labile soil P
pools represent an integration of various forms of N and P. For example, P has various forms in the soil and can be
transformed between those forms (Wang et al., 2007; Yang and Post, 2011). Some forms of organic P (e.g. bicarbonate Po in

Hedley method, Yang and Post, 2011) can easily be mineralized and thus were implicitly included in our labile soil P pool. Other forms of P that are not easily accessible to plants are referred to as "occluded P" and labile soil P can become occluded P (Wang et al., 2010; Goll et al., 2017a). Fluxes connecting the pools are described by the differential equations given in Appendices A-C. An overview of the C, N and P cycles and their interactions are presented in the following sections. A full list of the symbols and their definitions is given in Table 1.

## 2.1 C cycle

The C cycle in the GOLUM-CNP model is based on the DALEC2 model (Bloom et al., 2016; Bloom and Williams, 2015). We used a similar structure to define the C pools of GOLUM-CNP but grouped the DALEC2 foliar and labile vegetation C pools into a single foliar pool (Fig. 1). Net primary production (NPP) is allocated to the three biomass pools. The outgoing fluxes from biomass pools include losses from fire, the transfer of foliage and root detritus to litter and the transfer of wood debris directly to the SOM pool. The outgoing fluxes from litter include losses from fire and decomposition. A fraction of decomposed litter is respired and returned to the atmosphere as $CO_2$, the remaining fraction being converted to SOM. The SOM pool loses C by fire and decomposition. Differential equations governing the dynamics of C pools are given in Appendix A.

## 2.2 N cycle

The N cycle in GOLUM-CNP is coupled to the C cycle: the pool sizes of N are determined by the C-pool sizes and their respective N:C ratios; the N fluxes from different pools are determined by the N-pool sizes and corresponding turnover rates. The N cycle includes a specific soil inorganic-N pool in addition to the five pools of the C cycle. The inputs of N to ecosystems include atmospheric N deposition and N fixation ($N_d+N_{fix}$ in Fig. 1), both of which are assumed to enter the inorganic-N pool. The total N-fixation flux in this study includes both symbiotic and asymbiotic fixation (see Sect. 3.1), separately estimated from a previous study (see Sect. 3.1). We do not separate the two fixation processes and assume that they together contribute to the inorganic-N pool, although these two pathways of N fixation are differed in terms of the relationships between $N_2$-fixing microorganisms and plants. We did not consider the flux of N mobilized from near-surface rocks, although a recent paper by Houlton et al. (2018) pointed out this flux may be an important N sources in montane and high-latitude ecosystems. N uptake ($F_N$) by plants is assumed to be solely from the inorganic-N pool. Organic N is an important N supply for plants (Näsholm et al., 2009)in boreal-forest and tundra ecosystems (Schimel and Bennett, 2004; Schimel and Chapin, 1996; Zhu and Zhuang, 2013), but the quantitative importance of this process is still unknown for other ecosystems globally. We thus ignored the uptake of organic soil N. N uptake by plants from the inorganic-N pool is modeled from the N:C ratio of NPP allocated to biomass pools minus the resorbed N. In the real world, N is only resorbed at the end of the growing season or leaf lifespan and then stored in plant organs and remobilized during the next growing season. Here, because our model does not have a sub-annual time step, rates of resorption described by a resorption coefficient (Appendix B) are assumed to be constant over time. We also assumed that N is not resorbed from fine roots or wood, because evidence for this process is inconclusive (Gordon and Jackson, 2000; Zechmeister-Boltenstern et al., 2015). N mineralization is modeled along with litter and SOM decomposition. N immobilization due to the uptake of inorganic soil N by soil organisms is modeled to match the higher N:C ratio of the SOM pool than its donor (wood and litter) pools. Loss of N from ecosystems occurs through fire, denitrification and leaching. The N lost due to fire is assumed to be emitted only in gaseous form, because the proportion of N retained in the residual ash is very small (Niemeyer et al., 2005; Qian et al., 2009). We consider the gaseous loss of inorganic N from denitrification but ignore the volatilization of N in the form of $NH_4^+$. This flux usually occurs at a soil pH >8 (Freney et al., 1983) or after application of N fertilizers (Yang et al., 2009), and $NH_3$ emissions from soils under non-agricultural vegetation are relatively small globally (5 Tg N y$^{-1}$, Bouwman et al., 1997; Houlton et al., 2015), representing <5% of total gaseous loss, so the omission of $NH_4^+$ N volatilization will not introduce large biases in our model

for most regions. The dynamics of N in the pools are summarized by the differential equations in Appendix B.
**2.3 P cycle**
The P cycle, like the N cycle, is also coupled to the C cycle; the dynamics of P in the pools are described by the
differential equations in Appendix C. The external inputs of P to ecosystems include atmospheric P deposition and P
released from P-bearing minerals by chemical weathering ($P_d+P_w$ in Fig. 1). P from deposition and rock weathering enters
the soil inorganic-P pool. The structure of the P cycle is the same as for the N cycle described above for foliar-P resorption,
P released from the decomposition of litter and SOM and the immobilization of inorganic soil P by soil organisms. Inorganic
P, unlike inorganic N, can be sorbed onto/into soil particles and subsequently become occluded. This form is assumed to be
unavailable to plants. We modeled the flux from the labile soil P to occluded P with a constant rate. Loss pathways of P
include fire, leaching and conversion to occluded P. Notably, not all P mobilized by fire is emitted in gaseous form, but is
partly retained in the residual ash (Niemeyer et al., 2005; Qian et al., 2009). We used a constant fraction of 75% (Niemeyer
et al., 2005; Qian et al., 2009) to model the P retained in the residual ash during a fire, and this fraction of P enters the labile
soil P pool.
**3    Methods**
**3.1 Input data sets**
All parameters used as inputs for the calibration of GOLUM-CNP are listed in Table 1. A steady state was assumed to
infer remaining variables (also listed in Table 1). The estimates of fluxes and C-pool sizes were based on mass balances, and
the estimates of N- and P-pool sizes were derived from the C-pool size and stoichiometric data (see below and Appendix E).
We used the C fluxes and turnover times of C pools derived from CARDAMOM for the C cycle (Bloom et al., 2016), which
offered a data-consistent analysis of terrestrial C cycling on a global $1°\times1°$ grid for 2001-2010 by optimizing the DALEC2
model parameters to match the state and process variables with the global observations of MODIS LAI (Myneni et al., 2015),
soil C (Hiederer and Köchy, 2011), burned area (Giglio et al., 2013) and tropical biomass (Saatchi et al., 2011). Although the
CARDAMOM data-driven analysis only reported the C pools and fluxes, the impacts of N and P on the C cycle have been
implicitly reflected in CARDAMOM through the constraints by some of the observations. For example, the availability of N
and/or P limits the growth of vegetation and thus the LAI observed (Klodd et al., 2016; Reich et al., 2010); the N and P
contents in soil control the decomposition of soil C and thus the soil C pool observed (Manzoni et al., 2010). In this sense, it
is appropriate to use C cycle from CARDAMOM as inputs to estimate the pool and fluxes of N and P.
Different indices have been used to describe nutrient cycling from different perspectives (soil, individual plant,
vegetation, ecosystem, etc) (Augusto et al., 2017; Cleveland et al., 2013; Gill and Finzi, 2016). In this study, we focused on
the openness, nutrient use efficiencies and the residence time (Sect. 3.2) which are defined at ecosystem scale and thus
correspond to the scale at which DGVMs are typically defined. For the presentation of results, we distinguish seven biomes:
tropical rainforests (TRF), temperate deciduous forests (TEDF), temperate coniferous forests (TECF), boreal coniferous
forests (BOCF), tundra (TUN), tropical/C4 grasslands (TRG) and temperate/C3 grasslands (TEG). Note that similar
empirical land-cover maps have been also used in previous studies to simulate C, N and P cycles (Cleveland et al., 2013;
Wang et al., 2010). We applied observed biome-specific N:C ratios for each pool from the synthesis by Zechmeister-
Boltenstern et al. (2015).
We used the spatially explicit estimates of N deposition (Wang et al., 2017) for 2001-2010 (Fig. S2a), which were
evaluated with globally distributed *in-situ* measurements. The spatially explicit estimate for N fixation (Fig. S2b) was taken
from the CABLE model simulation for 2001-2010 (Peng et al., submitted) with a N fixation model developed by Wang et al.
(2007). The simulation result matches the relative abundance of $N_2$-fixing legumes in different ecosystems. Globally, the N

fixation is 116 Tg N yr$^{-1}$ and is within the range of empirical estimates (100-290 Tg N yr$^{-1}$; Cleveland et al., 1999; Galloway et al., 2004), but larger than the estimate of 44 Tg N yr$^{-1}$ by Vitousek et al. (2013) for pre-industrial conditions. The large range (44-290 TgN yr$^{-1}$) in the estimates of nitrogen fixation reflects both a paucity of measurements of N fixation, as well as incomplete understanding of the biophysical and biochemical controls on N fixation. And to our knowledge, CABLE simulation is the only product that has spatially explicit and processed-based estimates of N fixation, and is therefore used in this study. The resorption coefficients of leaves for the seven biomes were derived from the N:C ratios of leaves and leaf litter reported by Zechmeister-Boltenstern et al. (2015). The rate of loss of inorganic N by leaching was determined from data for total soil moisture and runoff (Eq. B7). The spatially explicit estimate of total soil moisture was derived from the European Centre for Medium-Range Weather Forecasts (ECMWF) Interim Reanalysis (ERA-Interim/Land; Albergel et al., 2013; Balsamo et al., 2015). Gridded soil water content was provided in ERA-Interim/Land in four discretized layers until 2.89 m below ground, which is the soil depth we considered for the total soil moisture. Although some uncertainties exist at grid scale, the large-scale patterns in soil moisture from ERA-Interim/Land are consistent with other products (Rötzer et al., 2015), enabling us to use it to represent the large-scale spatial gradients in soil water moisture. The global gridded estimate of runoff data was obtained from the Global Runoff Data Centre (GRDC, http://www.grdc.sr.unh.edu/), which is constrained by observed river discharges from 663 stations globally. We used observation-based estimates of the fraction of N lost by denitrification to the total inorganic-N loss (denitrification + leaching) pathways (Goll et al., 2017b) to calibrate the denitrification-loss flux. This fraction of denitrification loss ($f_{denit}$) was derived using a process-based statistical model fitted to global soil $\delta^{15}$N data sets, based on the distinct $^{15}$N fractionation effect of denitrification versus loss from leaching (Bai et al., 2012; Houlton and Bai, 2009).

We constrained the P cycle using spatially explicit estimates by Wang et al. (2017) for P deposition for 2001-2010 (Fig. S3a). Spatially explicit estimates of P input from rock weathering (Fig. S3b) were derived from data for river discharge and the chemical composition of minerals by Hartmann et al. (2014). The P:C ratios and resorption coefficients for the seven biomes were obtained from Zechmeister-Boltenstern et al. (2015). Only a fraction of total inorganic P can be lost by leaching, and this fraction of dissolved inorganic P in total labile P was derived based on the observations of Hedley soil P fractions as the resin-extractable P divided by total labile P reported by Yang and Post (2011) for the twelve USDA soil orders. The constant rate at which inorganic P becomes strongly sorbed ($f_{sorb}$, Eq. C6) was fixed at 0.04 y$^{-1}$ (Goll et al., 2017a).

Zechmeister-Boltenstern et al. (2015) only reported the stoichiometric ratios and the N and P resorption coefficients for seven large non-agricultural biomes, but other input variables were grid-based products. A land-cover map was used to aggregate the grid-based C-cycle variables from Bloom et al. (2016) into the biomes used by Zechmeister-Boltenstern et al. (2015). The land-cover map was derived from the dominant land-cover type for each grid cell for the globe, excluding croplands, from the land-cover map of the Climate Change Initiative (LC_CCI) established by the European Space Agency (ESA) (Bontemps et al., 2013) at 0.25 °×0.25 ° resolution. Specifically, we used the 2010 map to classify all grid cells into one of the seven non-agricultural biomes of Zechmeister-Boltenstern et al. (2015) (Fig. 2), following the methodology presented by Poulter et al. (2015).

**3.2 Model integration and output diagnostics**

We applied the model framework described in Sect. 2 to derive a data-driven estimate of steady-state C, N and P cycling. A steady state indicates that annual mean input fluxes for all pools are assumed to be balanced by annual mean outgoing fluxes, with the annual mean outgoing fluxes from organic pools calculated as the quotient of the pool sizes to the corresponding turnover times. Assuming that all pool sizes were in a steady state, the left side of the equations in Appendices A-C (Eqs. A1-A5, B1-B6 and C1-C6) are all equal to zero. Adding the constraints in Appendix D (Eqs. D1-D11), we derived a system with 28 equations and 28 unknown variables (Table 1), thereby defining all unknowns in GOLUM-CNP. The unknown variables were solved by applying the 33 global spatially explicit observation-based estimates listed in Table 1

in these equations (Appendix E). The set of equations of the GOLUM-CNP model was solved for each 0.25 °×0.25 ° grid cell
using biome-mean N:C and P:C stoichiometric ratios, grid-cell specific values of C variables from Bloom et al. (2016) and
the gridded external N- and P-input and -output fields described above. In this computation, some processes were only
solved by mass balance and the steady-state assumption instead of explicitly being calibrated. For example, we did not
explicitly simulate various pathways of N and P mineralization and immobilization. The N and P mineralization fluxes are
computed as the product of the decomposition of C in litter and SOM and their respective stoichiometries, and N and P
immobilization fluxes are computed by mass balance to match the higher N:C and P:C ratios in the SOM pool compared to
the ratios in inputs to SOM from wood and litter decomposition. For instance, N and P mineralization were computed as the
difference between nutrient demand of vegetation and the sum of external inputs and resorption in Cleveland et al. (2013,
Eqs. S5 and S6), assuming that the nutrients available to plants in soil do not change significantly at current stage. Some
variables were computed by mass balance and do not rely on steady state assumption, e.g. the uptake of N ($F_N$) and P ($F_P$) by
plants (Table 1). Such computations based on mass balance and steady-state assumptions allow us to have a diagnostic
modelling framework, but at the same time capture observations of carbon fluxes, pools and pool stoichiometries.
The inputs of the C cycle from the original CARDAMOM dataset were provided as probability distributions, while
other datasets were provided only as mean values. In this study, we compute the GOLUM-CNP using the mean values of all
the input datasets to represent the mean behaviour of the C, N and P cycling.
We present the C, N and P pools and the fluxes between them for each biome. We also aggregated the results at the
global scale and compared them with previous studies. We calculated some ecologically relevant quantities from the
GOLUM-CNP output. We defined the *openness* of N and P cycles as NO and PO that were calculated as the ratio of
nutrients inputs ($I_x$, $X \in \{N,P\}$, taken as the sum of deposition ($N_d$) and biological fixation ($N_{fix}$) for N and as the sum of
deposition ($P_d$) and release from rock weathering ($P_w$) for P, over the amount of nutrients in production, taken as the sum of
uptake from inorganic N or labile soil P pools ($F_x$, $X \in \{N,P\}$) and resorbed nutrients ($RSB_x$, $X \in \{N,P\}$), leading to:

$$XO = \frac{I_x}{F_x + RSB_x} \tag{1}$$

The openness quantifies how much nutrients is from external inputs, which is similar but not strictly equal to the
"proportion of new NPP fueled by new nutrient inputs". In general, the "openness" used this study and the "proportion of
new NPP fueled by new nutrient inputs" in Cleveland et al. (2013) both quantify the ratios between fluxes that are related to
external inputs and the "total" fluxes, but "openness" used in this study was defined from nutrient cycles, while the index
used in Cleveland et al. (2013) was defined from NPP-carbon. In addition, the practical computation of the openness in this
study are slightly different from that of Cleveland et al. (2013), which was quantitatively compared in the supplementary
material S1.
The mean residence time of N and P for the entire ecosystem ($\tau_{X,eco}$, $X \in \{N,P\}$) was defined as the ratio of total
modeled pool mass (including plant, litter, SOM and inorganic pools) to all outgoing fluxes. The sum of all steady-state
outgoing fluxes was set equal to the sum of external input fluxes, so we calculated the mean residence time of N and P by:

$$\tau_{N,eco} = \frac{\sum_{i=1}^{5} N_i + N_{inorg}}{N_d + N_{fix}}$$

$$\tau_{P,eco} = \frac{\sum_{i=1}^{5} P_i + P_{inorg}}{P_d + P_w} \tag{2}$$

The *nutrient-use efficiencies* (NUE and PUE) were defined by:
$$XUE = \frac{GPP}{Fx} = \frac{NPP / f_{NPP}}{Fx}$$ (3)
where $F_x$ ($X \in \{N,P\}$) is the annual uptake of inorganic soil N or P by plants, and $f_{NPP}$ is the ratio of NPP to gross primary
production (GPP) from CARDAMOM. Our model used NPP as the input C flux for ecosystems, but we used GPP instead of
NPP in Eq. (3) to calculate XUE for comparing with the estimates based on *in situ* measurements by Gill and Finzi (2016),
which were also based on GPP. We thus used $f_{NPP}$ only as an external variable in our modeling framework, and $f_{NPP}$ was not
targeted when evaluating the sensitivities and uncertainties of the results (see below).
We tested the steady-state sensitivity (SS) of the model results to the observational data sets (inputs of the model listed
in Table 1) by linearizing the GOLUM-CNP model and its solver for calculating the first-order partial derivative of all
outputs relative to each input parameter:
$SS = \partial\mathbf{O}/\partial\mathbf{I}$ (4)
where $\mathbf{I}$ is the vector of the input variables, and $\mathbf{O}$ is the vector of the output variables. This approach directly provided a
sensitivity matrix, which allowed us to test the effect of the accuracy of the measurement of each input variable on the model
results for the N and P cycles. This method was similar to the "one-at-a-time" (OAT) approach used for sensitivity analysis
in previous C-N coupled modeling studies (Orwin et al., 2011; Shi et al., 2016; Zaehle and Friend, 2010) but did not require
running simulations by changing the inputs one at a time. This approach did not fully explore the possible range of values for
a given parameter, but provided comparable SS values for different parameters, which is useful when the full uncertainty
ranges of some parameters are unknown, e.g. uncertainty due to the inconsistent definitions between the measured pools in
the real world and the conceptual pools in the model, or the large uncertainty due to sparse observations for some biomes.
The input parameters had distinct magnitudes (and units), so we used the relative sensitivities, e.g. $SS = \partial\mathbf{O}/\mathbf{O}/(\partial\mathbf{I}/\mathbf{I})$, to
compare the sensitivities to different model inputs. For the sensitivity analysis, an SS of 1 indicates that a 1% increase (or
decrease) in a model input produces a 1% increase (or decrease) in the model output, and an SS of -0.5 indicates that a 1%
increase (or decrease) in the model input produces a -0.5% decrease (or increase) in the model output. The results of this
sensitivity analysis could be further used to investigate the sources of uncertainty in the outputs and to evaluate variances of
the model outputs using error propagation:

$$\mathbf{\varepsilon}_{\mathbf{O},i} = \frac{\partial\mathbf{O}}{\partial\mathbf{I}_i}\mathbf{\varepsilon}_{\mathbf{I},i}$$
$$\mathbf{\varepsilon}_{\mathbf{O}} = \sum_{i=1}^{n}\frac{\partial\mathbf{O}}{\partial\mathbf{I}_i}\mathbf{\varepsilon}_{\mathbf{I},i}$$
$$\Sigma_{\mathbf{O}} = E(\mathbf{\varepsilon}_{\mathbf{O}}^{\mathsf{T}}\mathbf{\varepsilon}_{\mathbf{O}})$$
(5)

where $\varepsilon_{\mathbf{I},i}$ is the error in the $i$th input data, $\varepsilon_{\mathbf{O},i}$ is the error propagated from the error in input $i$, $\varepsilon_{\mathbf{O}}$ is the error that accounts for
errors in all input data, E represents the expectation of a variable and $\varepsilon_{\mathbf{O}}$ is the covariance matrix whose diagonal entries are
the variances of the outputs.
**3.3 Adjustments of CARDAMOM C cycle**
In CARDAMOM, there was no explicit separation between forests and grasslands and CARDAMOM provided low
woody biomass in grassland dominated regions (Saatchi et al., 2011; Williams et al., 2013), while grasslands are considered
as biomes and have no woody biomass in GOLUM-CNP. In order to represent the grassland biomes in GOLUM-CNP and to
conserve the global NPP from CARDAMOM, we approximated the C-cycle state of the non-forest biomes (TRG, TEG and
TUN) by partitioning half of CARDAMOM woody NPP to foliar NPP and half to fine roots, in order to better represent
grassland C, N and P cycling across these biomes.
The CARDAMOM terrestrial C analysis did not assume steady states. Our goal, however, was to describe the steady

states of C, N and P cycling, because few global long-term observations associated with N and P were available to constrain the model. We recalculated the C cycle based on a subset of the CARDAMOM results. Specifically, we used NPP and turnover times of the C pools for 2001-2010 (Table 1) and recalculated the steady-state sizes of these pools and the transfers of C between the pools represented in Fig. 1, solving Eqs. A1-A5 with their left sides as zeros. This steady-state transformation of the CARDAMOM C cycle is assessed in Sect. 4.1.

## 4    Results

### 4.1 Steady state C cycle

Table 2 shows the global C-pool sizes and main fluxes of the steady-state C cycle transformed from CARDAMOM under the climate conditions of 2001-2010.. Although the steady-state C stocks do not exactly represent the C stocks at present day, the differences between the steady-state transformed pool sizes and the original non-steady-state CARDAMOM results were within 10% for most C pools and fluxes. The largest differences were for biomass (foliar, fine-root and wood) pools. The larger foliar and fine-root pools in the steady-state GOLUM-CNP model were due to adjustments done for grass dominated grid cells for which CARDAMOM provided some wood growth inconsistent with the biome distribution used in GOLUM-CNP. In these cases, we allocated wood growth from CARDAMOM into growth of fine root and foliage. These pools in GOLUM-CNP, however, remained within the [5, 95th] percentile range of the original CARDAMOM values. The pool size for global woody biomass was 37% smaller in the steady-state model (469 Pg) than the original CARDAMOM results but remained within its inter-quartile range (364-984 Pg). The differences between the gridded maps from original CARDAMOM and GOLUM-CNP are shown in Fig S1. The steady-state transformed C stocks in biomass, litter and SOM were within the 25th and 75th percentiles of the original CARDAMOM results at more than 90% of forest grid cells, indicating that our steady-state transformed C stocks are close to the actual C stocks at present day, given the large uncertainties in the state-of-the-art estimates. Due to the adjustment made for the grass dominated grid cells (see above), the C pools for grassland differ more strongly than forest-dominated area from the original CARDAMOM.

### 4.2 Steady-state nutrient stocks and fluxes

Figure 3 and Fig. 4 summarizes the stocks and fluxes of N and P for the seven biomes and for the globe. The uptake fluxes of N and P were largest for tropical forests, mainly driven by the large NPP of this biome. Rates of N and P uptake were lower for temperate and boreal forests than tropical forests and for non-forest biomes than forests. The pool sizes of N and P in plants tended to decrease from tropical to boreal regions, following the C-pool sizes and their observed stoichiometries. Conversely, N and P contents in litter were larger for boreal forests, temperate grasslands and tundra ecosystems than the other biomes, mainly due to a longer turnover of the litter pool in these biomes. The N-pool size of SOM was also larger in boreal forests, temperate grasslands and tundra than the other biomes. The P-pool size in SOM, however, was smaller for boreal forests and tundra than the other biomes, consistent with the differences between the N:C and P:C ratios of boreal biomes compared to other biomes (Table S1). Inorganic N and labile soil P pools and leaching rates of N and P were higher in tropical forests, where runoff was higher than in the other biomes. Semi-arid tropical grassland (TRG) had high losses of N and P by fire and a low loss from leaching. The internal N and P fluxes within ecosystems were usually much larger than the external input fluxes and the output fluxes for all biomes, highlighting the dominant role of internal cycling of N and P, which differed from C cycles where NPP and losses by respiration were larger than any internal C flux.

Here we compared the estimates of N and P stocks for global terrestrial biosphere with other studies. Our estimate of N in plants (3.9 Pg N) was close to the estimate modeled by Zaehle et al. (2013) (3.5 Pg N), and was within the range of other studies, from 1.8 Pg N by Yang et al. (2009) and 6.57 Pg N by Wang et al. (2010) (6.57 Pg N). Our estimate of N in litter

and SOM was lower than the estimate of 65 Pg N by Xu-ri et al. (2008) and Yang et al. (2009), but smaller than the estimate of 126 Pg N by Wang et al. (2010). Our estimate of the P mass in plants (0.17 Pg P) was smaller than the estimates modeled by Wang et al. (2010) (0.39 Pg P) and Goll et al. (2012) (0.23 Pg P). Our estimate of the litter P mass (0.03 Pg P) was similar to the estimate of 0.04 Pg P by the CABLE model (Wang et al. 2010) but was two-fold lower than the estimate (0.08 Pg P) modeled by Goll et al. (2012).

The rate of total N input (deposition and fixation) aggregated to global scale was 0.19 Pg N $y^{-1}$ and equated (by construction) to the steady-state rate of total N loss. Total N uptake by plants was 0.68 Pg N $y^{-1}$. Our estimate of N denitrification was 0.10 Pg N $y^{-1}$, consistent with the independent estimate of global soil denitrification of 0.12 Pg N $y^{-1}$ by Seitzinger et al. (2006) and within the range reported by other studies, from 0.04 Pg N $y^{-1}$ (Houlton and Bai, 2009) to 0.29 Pg N $y^{-1}$ (Galloway et al., 2013). The global loss of N was 0.05 Pg from fire and 0.04 Pg N $y^{-1}$ from leaching, the latter being similar to the independent estimates by Galloway et al. (2004, 2013) of 0.013-0.18 Pg N $y^{-1}$ and by Houlton and Bai (2009) of 0.09 Pg N $y^{-1}$.  Globally, the loss of N by fire accounted for 26% of the total N loss. The total input of P to the terrestrial ecosystem was 0.007 Pg P $y^{-1}$, 86% from deposition (range from 71% for BOCF to 92% for TRG); only a small fraction was from rock weathering (ranging from 8 to 29% across biomes). The loss of P is mainly from leaching and the loss by fire accounted for only 18% of the total P loss, much smaller than the fraction for N.

**4.3 Implications for ecological research**

Figure 5 shows the latitudinal distribution of foliar N:P ratios in our model. Thisresult reflects the distribution of the seven biomes and respective C:N and C:P ratios – both of which are prescribed here. Foliar N:P ratios decreased on average from low to high latitudes. Estimates from previous studies also followed this trend (Kerkhoff et al., 2005; McGroddy et al., 2004; Reich and Oleksyn, 2004) based on foliar measurements. The mean N:P ratios in our study were in the middle of the range of observations for all latitudes. The results of GOLUM-CNP better indicated the high N:P ratios between 20 ° to 40 °, where grassland is the dominant biome, than the monotonic regressions (colored lines in Fig. 4) derived by Reich and Oleksyn (2004) and Kerkhoff et al. (2005) for foliar data, implying that the use of stoichiometries at the scale of large biomes can identify the general features of the spatial gradients of N and P cycling.

Figure 6a and 6b show the distribution of the openness (defined as the ratio of new nutrient inputs to the total plant uptake of nutrients, Sect. 3.2) for N and P in different ecosystems and Fig. S4a and S4b show the gridded maps of these indices. New N in forest ecosystems (sum of deposition and biological fixation) accounted for 10% (BOCF) to 51% (TECF) of the total plant uptake of N, and new P (due to deposition and rock weathering) accounted for only 3.5% (BOCF) to 15% (TRF) of the total plant uptake of P. The openness of both N and P in grassland ecosystems decreased from the tropics to high latitudes. The residence times of N and P in ecosystems were much longer than those of C (Table S2) and decreased from the tropics to boreal areas (Fig. 5c, 5d, S5a and S5b).

The openness and residence times of N and P together allow to assess the relative importance of external inputs and internal cycling to support plant growth. For example, TECF are characterized by a more open N cycle and a longer N residence times compared to TRF. The difference in the openness of N cycle indicates that the TECF intends to invest more resources to obtain N from external inputs than TRF. The differences in residence times indicate that N is more efficiently conserved within the ecosystem in TECF compared to TRF, and such a conservation within ecosystems is primarily driven by differences in the turnover of dead organic matter (Fig. 3). The  P cycle is less open than N cycle in all ecosystems, highlighting the importance of ecosystem P recycling within ecosystems to support plant growth.

Figure 7 shows the diagnosed nutrient-use efficiencies from GOLUM-CNP outputs for the seven biomes and Fig. S6a and S6b show the gridded maps of nutrient-use efficiencies. Among forest biomes, tropical forest had the lowest NUE and the highest PUE compared to other forest biomes (Fig. 7a), consistent with the higher P and lower N stresses in tropical ecosystems (Gill and Finzi, 2016; Reich and Oleksyn, 2004). The values of NUE and PUE were similar to each other for

TEDF, TECF and BOCF. Nutrient-use efficiencies were about 3-fold lower for non-forest biomes (Fig. 7b) than forest
biomes, and both NUE and PUE decreased from tropical/C4 grassland to tundra.
**4.4 Sensitivity analysis**
Figure 8 shows the mean sensitivity of the nutrient-uptake fluxes ($F_N$ and $F_P$), nutrient-use efficiencies (NUE and PUE),
pool sizes of inorganic N and P ($N_{inorg}$ and $P_{inorg}$), N and P openness, residence times of N and P in the ecosystem ($\tau_{N,eco}$ and
$\tau_{P,eco}$) and residence times of N and P in plants ($\tau_{N,plant}$ and $\tau_{P,plant}$) to the input variables for the tropical-rainforest biome
(TRF). The sensitivities were similar for the other biomes (Figs. S7-S12). The uptake of nutrients in GOLUM-CNP was
determined by NPP, NPP-allocation fractions, observation-based nutrient:C ratios and resorption coefficients (Eqs E7 and
E18), so N uptake for tropical forest (Fig. 8a) was highly sensitive to NPP (1.0), NPP-allocation fractions (0.3) and the N:C
ratio (0.4) of the woody pool, and P uptake was sensitive to NPP (1.0) and foliar variables (0.5 for $\gamma_{C,1}$ and 0.5 for $\rho_{P,1}$; see
Table 1 for the definition of these variables). The nutrient-use efficiencies, defined in Eq. (1) as the ratio between GPP and
the nutrient-uptake fluxes (Eq. 3), were negatively sensitive to the input variables mentioned above. Estimates of the
openness of N and P were sensitive to input fluxes, NPP, NPP-allocation fractions and stoichiometric inputs. The residence
times of nutrients in the ecosystem were influenced by variables affecting vegetation growth (e.g. NPP and allocation
fractions of NPP) and those affecting inputs (e.g. deposition, N fixation and P release from rock weathering) to about equal
extent. They were also very sensitive to variables related to soil, e.g. the N:C and P:C ratios in soil and residence times of
soil. This reflect the large stocks of C and nutrients in soils than in the vegetation. The residence times of nutrients in whole
plants ($\tau_{N,plant}$ and $\tau_{P,plant}$) were more sensitive to the variables affecting woody biomass than those affecting foliage and fine
roots. The sensitivity of residence times in the ecosystem and whole plants suggested that the nutrient cycling in the
terrestrial biosphere were primarily determined by the largest pools.
**5    Discussion**
We developed a new observation-based modeling framework of global terrestrial N and P cycling built on a data-driven
C-cycle model and observed N:C and P:C stoichiometric ratios in different pools spatially averaged at the scale of large
biomes and observation-based estimates of the external input and output fluxes of N and P. This model was then used to
estimate the pool sizes and fluxes in N and P cycles and indicators of the coupling between nutrient and C cycling, including
nutrient openness, residence times in ecosystems and nutrient-use efficiencies. The data-driven estimates of steady-state
global C, N and P cycles are the first that are fully consistent with a large set of global observation-based data sets, under the
condition of current climate, deposition and $CO_2$ concentration. The indicators for the coupling between nutrient and C
cycling, which are emerging properties of GOLUM-CNP, are used to evaluate the capabilities of GOLUM-CNP to capture
observed patterns among biomes. We found that there are some differences between our data-driven estimates and previous
studies about the nutrient efficiencies at biome scales (Gill and Finzi, 2016) and the openness (Cleveland et al., 2013). In this
section, we discussed the major uncertainties in our model and show how these uncertainties affect the computation of
nutrient efficiencies and the openness (Sect. 5.1). Of note is that most of our discussions are for the C cycle (based on the
sensitivity analysis, see below), and since CARDAMOM is the only data-driven C cycle to our knowledge, the modifications
of the CARDAMOM dataset we made in this section are more qualitative and diagnostic rather than deterministic. Such an
example highlights some important variables that should be investigated or considered in future data-driven products.
**5.1 Sensitivity to C variables**
Our estimates of nutrient-use efficiencies differed significantly from those estimated from *in situ* measurements (red
squares and diamonds in Fig. 7) by Gill and Finzi (2016), particularly the values of PUE for all biomes and NUE for
temperate and boreal forests. NUE and PUE were determined by NPP-allocation fractions, stoichiometric ratios, resorption

coefficients and fractions of fire in the total outgoing C flux (Eqs. 3 and E7, where NPP is canceled by the division). The CARDAMOM observation-based analysis of the C cycle is the basis of the GOLUM-CNP modeling framework, so that errors and uncertainties in CARDAMOM for the C cycle translate into errors and uncertainties of GOLUM-CNP. Quantitatively, the sensitivity analysis (Figs. 8, S7-S12) indicated that $F_N$ and $F_P$, and thus NUE and PUE, were most sensitive to the NPP-allocation fractions (especially to woody biomass) and foliar stoichiometry. We applied the sensitivity matrix (Eq. 5) to further calculate the contribution of variances from each of these input variables, in which the uncertainties in the NPP-allocation and fire fractions were obtained from CARDAMOM and the uncertainties (1-sigma) in the N:C and P:C stoichiometric ratios and resorption coefficients were assumed to be 40%. This 40% uncertainty was larger than the uncertainty (20%) of the N:C ratios used by Wang et al. (2010), so our estimate of the contribution of uncertainties from the stoichiometric ratios was relatively large. The contribution of these different sources of uncertainty to the variances of NUE and PUE is shown in Fig. 9 for temperate coniferous forests whose NUE and PUE deviated the most from the estimate by Gill and Finzi (2016). Fig. 9 shows that the NPP-allocation fractions were the largest contributors to the total variances in NUE and PUE, which totaled >80%.

The NPP-allocation coefficients in CARDAMOM were only constrained indirectly by the satellite observations of LAI and tropical aboveground biomass. The uncertainty of the CARDAMOM allocation fractions was thus substantial, especially for non-tropical biomes where no biomass data were used (allocation-fraction $25^{th} – 75^{th}$ percentile ranges are typically >50% of the mean). For example, the mean fraction of NPP allocated to woody biomass in CARDAMOM was >60% in most grids (Fig. S13a), which is rare for field measurements (Chen et al., 2013; Doughty et al., 2015). The mean allocation of NPP to fine roots may have been underestimated, characterized by too long a turnover time in CARDAMOM (range from <1 to 10 y) compared to field measurements (<3 y for all ecosystems, Gill and Jackson, 2000; Green et al., 2005). The CARDAMOM results indicated a turnover time of leaves in temperate and boreal biomes of <1 y, while Reich et al. (2014) indicated that the typical life span of conifer needles in evergreen coniferous forests depended on temperature and ranged from 2.5 to >10 y, this inconsistency being attributed by Bloom et al. (2016) to the potential roles of seasonal MODIS LAI biases and to the presence of understory vegetation across high-latitude ecosystems (Heiskanen et al., 2012).

Considering these inconsistencies between mean CARDAMOM values and *in situ* measurements, we conducted an additional experiment in which the CARDAMOM fields were further adapted: 1) the mean NPP-allocation fractions to woody biomass and the turnover time of woody biomass was divided by 1.5 to make sure that the NPP-allocation fractions to woody biomass fall in the range of field measurements, 2) the foliage turnover time of TECF and BOCF forests and associated NPP-allocation fractions were adjusted (keeping foliar biomass not changed) to match *in situ* observations (Reich et al., 2014) based on the fitted relationship between the needle longevity and mean annual temperature from Reich et al. (2014), assuming that understory vegetation plays a minimal role in C, N and P cycling and 3) Since in CARDAMOM, the NPP was constrained by GPP (GPP being constrained by the observation of LAI and the relationship between LAI and GPP) and the observation of biomass, additional adjustments were made to conserve the total NPP and pool sizes estimated from CARDAMOM by allocating the residual NPP after the modifications from step 1 and 2 to fine roots, and adjusting the turnover time of fine roots to conserve exactly the pool size of CARDAMOM (see Figs. S13-S16 for the adjusted variables and original CARDAMOM values).. Fig. 10 shows the NUE and PUE from this new experiment based on this modified version of the C cycle from CARDAMOM. The NUE and PUE were lower than those in Fig. 7 for the forest biomes, especially for TECF and BOCF, which tended to decrease PUE from tropical to boreal forest. This distribution of PUE among the biomes in Fig. 10 better matched the differences between biomes presented by Gill and Finzi (2016). Remaining inconsistencies could be attributed to the different methods used in this study and by Gill and Finzi (2016) are different. For example, Gill and Finzi (2016) notably used the net mineralization rates of N and P to approximate plant uptake, because their differences were an order of magnitude smaller than net nutrient mineralization. These authors used *in situ* measurements of net N mineralization but used a statistical model to estimate P mineralization based on a soil-order-specific

soil-P pool due to the lack of data (Yang and Post, 2011) and a regression between soil-P turnover times and mean annual
temperature. Their estimate of plant uptake was thus independent of vegetation stoichiometry, which differed from our study.
Gill and Finzi (2016) also used bootstrapping to sample the NPP and net N (or P) mineralization from independent studies.
Their estimates of NUE and PUE were thus not based on paired data, so their estimates may contain some sampling errors.

## 5.2 Uncertainty of nutrient-cycle openness

The distribution of nutrient-cycle openness in the seven biomes was presented in Sect. 4.3 and Fig. 6. Our estimate of a
small openness of N and P in BOCF, and that the openness was smaller for the P than the N cycle, were consistent with the
estimates by Cleveland et al. (2013). Our estimates of the N openness, however, were about twice as large as the estimates of
Cleveland et al. (2013). This difference was due to the larger deposition fluxes in our study (globally 72 Tg N $y^{-1}$) than those
used by Cleveland et al. (2013) (33 Tg N $y^{-1}$; from Dentener, 2006), because Wang et al. (2017) used an atmospheric model
with higher horizontal resolution and an updated inventory of reactive-N (e.g. $NO_x$ and $NH_3$) emission (Wang et al., 2017)
and also because Cleveland et al. (2013) assumed that only 15% of deposited N was available to plants. Cleveland et al.
(2013) demonstrated that changing the fraction of biologically available deposited N to 100% did not significantly change
the openness, because N-deposition fluxes were generally smaller than N fixation and accounted for a small fraction of
external N inputs in their study. Our estimates of P openness were also larger than those of Cleveland et al. (2013), which we
attributed to the large differences in the estimates of P deposition between the two studies. Cleveland et al. (2013) used P
deposition (0.26 Tg $yr^{-1}$) from Mahowald et al. (2008), which were an order of magnitude lower than recent estimates from
Wang et al. (2017) used in this study (5.8 Tg $yr^{-1}$), because Wang et al. (2017) revised the contribution of anthropogenic P
emissions and P in particles with diameters >10 μm (Wang et al., 2015, 2017). We also found that the P-cycle openness
decreased from the tropics to the boreal region, in contrast to the results by Cleveland et al. (2013). This also derives from
the differences in the spatial gradients of P deposition in the two studies. Mahowald et al. (2008) found that P deposition was
largest in northern Africa and that P deposition was within the same order of magnitude for tropical and temperate forests.
Wang et al. (2017), however, found that P deposition was much larger over tropical forests than other regions. The
contrasting spatial gradients in P deposition was likely due to the different models of atmospheric transport used by Wang et
al. (2017) and Mahowald et al. (2008). More importantly, most stations measuring total P deposition are in temperate regions,
and measurements of P deposition over tropical forests are very limited (Mahowald et al., 2008; Wang et al., 2017), so the
estimates of P deposition in the tropics were not well constrained by *in situ* observations and thus had large uncertainties.
Differences in the spatial gradients in nutrient-cycle openness between our study and the study by Cleveland et al. (2013)
demonstrated the impact of uncertain input data sets on the estimate of ecologically relevant quantities. The quantitative
assessment of the uncertainties in our estimates of openness, however, was difficult, because the potential uncertainties in
these data sets were not systematically evaluated within and between different estimates, and should therefore be addressed
in future studies.

## 5.3 Future research and data needs

Our estimates of global N and P cycles were at the scale of large biomes. Recent studies of N and P cycles have relied
on biome-specific stoichiometry (Cleveland et al., 2013; Wang et al., 2010). Stoichiometry, however, is also highly variable
within biomes (Reich and Oleksyn, 2004). For example, Kattge et al. (2011) found that 40% of the variability in foliar N
content was within species (finer scale than that of large biomes) and suggested that these stoichiometric ratios may be better
represented by future trait-based estimates rather than fixed species-specific values. Some improvements have been made on
the variation of stoichiometric ratios across climatic and ecological gradients within and across biomes, and on the
contribution of plant traits and environmental conditions to these variations (Dong et al., 2017; Han et al., 2005; Meyerholt
and Zaehle, 2015). However, it is still not sufficient to derive a globally gridded overview of the N and P cycles on current

knowledge. A better understanding of the stoichiometric variability and its drivers is still needed in terms of not only representing the large-scale gradients but also reducing the uncertainties at local scale. New and spatially interpolated stoichiometric data sets should partly overcome this problem, although uncertainties in the interpolation will need to be carefully propagated on GOLUM-CNP outputs.

We assumed that all terrestrial ecosystems were at a steady state for 2001-2010 due to a lack of global constraints on the dynamics of N and P cycling over a long period. Terrestrial ecosystems, however, are not currently at steady states (Luo, 2017; Luo and Weng, 2011), due to climate change, increasing atmospheric $CO_2$, anthropogenic disturbance etc, (Friedlingstein et al., 2006; Sitch et al., 2015). Zaehle (2013) reported that the terrestrial biosphere has accumulated 1.2 Pg N and 134.0 Pg C since the pre-industrial period. Wang et al. (2017) also found that N and P deposition have changed dramatically over time. The simulations by different models varies considerably, e.g. the responses of the biosphere to the increasing atmospheric $CO_2$ (Zaehle et al., 2014) and thus in future projections, because the current data sets have had little success in constraining all key processes in most DGVMs. Our results contribute to  evaluating models simulating global biogeochemical cycles. Although our steady-state C pool sizes (given the NPP and residence time at the condition of current climate) were within the [25, 75th] percentile range of the original non-steady-state CARDAMOM results (Fig. S1) at most grid cells, the biomass C stocks at 5%-10% of forest grid cells exceed the uncertainty range of CARDAMOM. In addition, independent remote-sensing estimates for 30 °N to 80 °N were 4.76 ±1.78 kg C $m^{-2}$ for mean forest C density and 79.8 ±29.9 Pg C for total forest C (Thurner et al., 2014), which were lower than the GOLUM-CNP estimates (6.51 kg C $m^{-2}$ for mean forest C density across pixels defined as forest in Fig. 2, and 181 Pg C for total forest C) for this region. This inconsistency was largely due to the fact that northern temperate and boreal forests may deviate substantially from their equilibrium for the current NPP (Pan et al., 2011), because of climate change and elevated $CO_2$. Residual overestimation could be also due to the fact that biomass removal by harvesting and from disturbance other than fires was not explicitly constrained in CARDAMOM and thus not represented in GOLUM-CNP. A transient simulation of N and P cycling will be needed in future studies as more constraints on N and P cycles emerge to study the effects of climate change, increasing $CO_2$ levels and disturbance on N and P cycles and their feedbacks. In such a transient simulation, a key process would be to simulate both the short-term and long-term responses of plants to the changing environment, e.g. how the plants would react when the inorganic N or labile soil P was not sufficient. Different models assumed different hypotheses under these conditions. For instance, N:C and P:C ratios are fixed and the photosynthesis rate is reduced to meet the low uptake of nutrients in Thornton et al., (2007). In Wang et al. (2010), the N:C and P:C ratios in biomass can vary within certain ranges, insufficient nutrient uptake would first result in a low concentration of N and/or P in plant tissues and the low concentration of nutrient would then limit the photosynthesis according to an empirical relationship between nutrient concentration and NPP. Similarly, when the N:C and P:C ratios in litter change, the decomposition rate of litter would change as a result of altered activity of microbes (Manzoni et al., 2017). In the future, more data are required to test these hypotheses and the transient simulation of next version of GOLUM-CNP should incorporate these interactions between the plants and environments.

In addition, some processes, such as the N inputs from rock weathering (Houlton et al., 2018) were not considered in this study, because 1) as stated in Houlton et al. (2018), it is still unknown how much of rock-released N can be used by plants when rock weathering happens deep beneath the soils; 2) in GOLUM-CNP, adding rock N inputs has the same effect than N fixation and N deposition (Eqs. B6 and E17); and 3) the estimate of total input of N to ecosystems (188 Tg N $yr^{-1}$) in this study are already at the higher end of the estimate (mean 147 Tg $yr^{-1}$, and range between 99.1 and 185.1 Tg $yr^{-1}$) of Houlton et al. (2018), even if rock N inputs are not accounted for, due to our larger estimates of N fixation and N deposition than Houlton et al. (2018). In the future, the rock N inputs and the fraction of these N inputs are accessible to plants should be further quantified and the quantity of total N inputs to the ecosystems should be reconciled between different studies. With these improvements, the future development of data-driven GOLUM-CNP should take into all these processes and fluxes.

The model structure of GOLUM-CNP is mainly described by the inputs (NPP for C cycle, N deposition and fixation for
N cycle, P deposition and release from rock weathering for P cycle) and residence times. Most DGVMs (e.g. Goll et al.,
2012, 2017a; Medvigy et al., 2009; Parton et al., 2010; Thornton et al., 2007; Wang et al., 2010; Weng and Luo, 2008; Xu-Ri
and Prentice, 2008; Yang et al., 2009; Zaehle et al., 2014; Zaehle and Friend, 2010) can be summarized by these two
components, although these models have more processes and use complex equations to describe the dynamics controlling
carbon and nutrient distribution among pools and the turnover of each pool. In this context, the output of the GOLUM-CNP
provides a traceable tool that can be used in the future to compare the results between GOLUM-CNP and different DGVMs.
As DGVMs are capable of computing the steady state of the biogeochemical cycles for present conditions, a direct
comparison between GOLUM-CNP estimate and DGVMs' estimates is possible.
At last, the sensitivity matrix presented in Sect. 4.4 provides a useful tool for assessing the uncertainties in model
outputs by propagating the uncertainties in the model inputs. We applied this method to quantitatively assess the sources of
uncertainties in the estimated nutrient-use efficiencies (Sect. 5.1 and Fig. 9), but we also found that the uncertainties for
some other quantities were currently difficult to obtain, because the estimates of uncertainties were not available for all
spatially explicit input data. This sensitivity analysis can be used in future studies to quantify the contribution of each input
data set to the uncertainty in other model outputs, to characterize the dominant sources of uncertainties in the estimated C, N
and P processes, to identify the major differences between different models (e.g. GOLUM-CNP versus DGVMs) and thus to
identify priorities for future data syntheses to fill the largest gaps in uncertainty. Future studies that provide global data sets
will need to include systematic evaluations and spatially explicit estimates of uncertainties in their data sets.
**6    Concluding remarks**
This study is a first attempt to combine observation-based estimates of C, N, P fluxes and pools in terrestrial ecosystems
into a consistent (steady-state) diagnostic model. Although there are considerable uncertainties in our results due to uncertain
and incomplete carbon cycle and nutrient observations, the main findings are: 1) external inputs of P from outside the
ecosystem contributes to a smaller plant P uptake than that of N, indicating a more important role of internal P recycling than
that of internal N recycling in supporting plant growth, 2) tropical forests have the lowest N use efficiency and the largest P
use efficiency, suggesting the adaptive response of this biome to the low P availability in the tropics. The structure of
GOLUM-CNP is analogous to most other process-based DGVMs describing carbon and nutrient interactions. The output of
the GOLUM-CNP provides a traceable tool and can be used in the future to test the performance of complex DGVMs in the
simulation of interactions between C, N and P cycling.
**Code and data availability**
The source code and the map of the classification of seven large biomes are included in the Supplement. For the other
datasets that are listed in Table 1, it is encouraged to contact the first authors of the original references.
**Appendix A Equations for carbon cycle**
The carbon cycle framework is based on DALEC2 model (Bloom and Williams, 2015), except that we combined the
labile and foliage pools together since the labile pool in DALEC2 only transfer to foliage. There are five pools in the C cycle
(1: foliage; 2: fine roots; 3: wood; 4: litter; 5: SOM). The equations governing the change of C pools are given by:
$\frac{dC_1}{dt} = -\tau_1^{-1}C_1 + \gamma_1 Fc$                                                     (A1)
$\frac{dC_2}{dt} = -\tau_2^{-1}C_2 + \gamma_2 Fc$                                                     (A2)
$\frac{dC_3}{dt} = -\tau_3^{-1}C_3 + \gamma_3 Fc$                                                     (A3)
$\frac{dC_4}{dt} = \tau_1^{-1} C_1 (1 - f_{fireC,1}) + \tau_2^{-1} C_2 (1 - f_{fireC,2}) - \tau_4^{-1} C_4$        (A4)
$\frac{dC_5}{dt} = \tau_3^{-1} C_3 (1 - f_{fireC,3}) + \eta \tau_4^{-1} C_4 - \tau_5^{-1} C_5$        (A5)
The definitions of the symbols are listed in Table 1.
**Appendix B Equations for nitrogen cycle**
There are five organic N pools and one inorganic soil N pool. The N cycle are described by the following equations:
$\frac{dN_1}{dt} = -\tau_1^{-1} N_1 (1 - \varepsilon_1) + \beta_1 Fn$        (B1)
$\frac{dN_2}{dt} = -\tau_2^{-1} N_2 + \beta_2 Fn$        (B2)
$\frac{dN_3}{dt} = -\tau_3^{-1} N_3 + \beta_3 Fn$        (B3)
$\frac{dN_4}{dt} = \tau_1^{-1} N_1 (1 - \varepsilon_1)(1 - f_{fireC,1}) + \tau_2^{-1} N_2 (1 - f_{fireC,2}) - \tau_4^{-1} N_4$        (B4)
$\frac{dN_5}{dt} = \tau_3^{-1} N_3 (1 - f_{fireC,3}) + \eta \tau_4^{-1} N_4 + N_{imob} - \tau_5^{-1} N_5$        (B5)
$\frac{dN_{inorg}}{dt} = \tau_5^{-1} N_5 (1 - f_{fireC,5}) + \tau_4^{-1} N_4 (1 - \eta - f_{fireC,4}) + N_d + N_{fix} - N_{imob} - f_{leach} N_{inorg} - f_{denit} N_{inorg} - Fn$

12        (B6)

The definitions of the symbols are listed in Table 1.
In Eq. B6, the fraction of inorganic N ($f_{leach}$) that is lost due to leaching is computed by soil water ($\Theta$) and the sum of
drainage and surface runoff (q). We use the spatially explicit estimate of daily soil moisture derived from the European
Centre for Medium-Range Weather Forecasts (ECMWF) Interim Reanalysis (ERA-Interim/Land; Albergel et al., 2013;
Balsamo et al., 2015) (see Table 1), and the global gridded estimate of monthly mean runoff data from the Global Runoff
Data Centre (GRDC, http://www.grdc.sr.unh.edu/). Since the runoff data only have a monthly time step, we use the same
value of runoff for each day within one month. The leaching fraction at annual scale is thus computed by:
$f_{leach} = \sum_{d=1}^{365} \frac{q_i}{\Theta_i + q_i}$        (B7)
Of note is that in this computation, $f_{leach}$ can exceed one, meaning that the turnover time of inorganic N pool is smaller
than one year (Wang et al., 2010).
**Appendix C Equations for phosphorus cycle**
There are five organic P pools and one inorganic soil P pool. The P cycle are described by the following equations:
$\frac{dP_1}{dt} = -\tau_1^{-1} P_1 (1 - \theta_1) + \varphi_1 Fp$        (C1)
$\frac{dP_2}{dt} = -\tau_2^{-1} P_2 + \varphi_2 Fp$        (C2)
$\frac{dP_3}{dt} = -\tau_3^{-1} P_3 + \varphi_3 Fp$        (C3)
$\frac{dP_4}{dt} = \tau_1^{-1} P_1 (1 - \theta_1)(1 - f_{fireC,1}) + \tau_2^{-1} P_2 (1 - f_{fireC,2}) - \tau_4^{-1} P_4$        (C4)
$\frac{dP_5}{dt} = \tau_3^{-1} P_3 (1 - f_{fireC,3}) + \eta \tau_4^{-1} P_4 + P_{imob} - \tau_5^{-1} P_5$        (C5)
$\frac{dP_{inorg}}{dt} = \tau_5^{-1} P_5 (1 - f_{fireC,5}) + + \tau_4^{-1} P_4 (1 - \eta - f_{fireC,4}) + P_d + P_w + 0.75 Fire_P - P_{imob} - f_{leach} f_{dissolve} P_{inorg} -$
$f_{sorb} P_{inorg} + Fp$        (C6)
Where $Fire_P$ represent the P in the ecosystem that suffers from fire events:
$Fire_P = \tau_1^{-1} P_1 (1 - \theta_1) f_{fireC,1} + \tau_2^{-1} P_2 f_{fireC,2} + \tau_3^{-1} P_3 f_{fireC,3} + \tau_4^{-1} P_4 f_{fireC,4} + \tau_5^{-1} P_5 f_{fireC,5}$        (C7)
**Appendix D Additional constraints**
1) Under steady-state, the N:C and P:C ratios for the plants and soil are assumed to be constant, so that $N_i$ and $P_i$ can be
calculated by the production of the C pool size from CARDAMOM and the stoichiometry ratios for each pool from
Zechmeister-Boltenstern et al. (2015), except litter which has different definitions in CARDAMOM and Zechmeister-
Boltenstern et al. (2015):
$N_i = \rho_{N,i} C_i \ \ (i = 1,2,3,5)$         (D1-D4)
$P_i = \rho_{P,i} C_i \ \ (i = 1,2,3,5)$         (D5-D8)
2) The fraction of NPP, $F_N$ and $F_P$ allocations sum up to 1:
$\beta_1 + \beta_2 + \beta_3 = 1$         (D9)
$\varphi_1 + \varphi_2 + \varphi_3 = 1$         (D10)
3) The fraction of gaseous loss of N due to denitrification to the total inorganic N loss should satisfy the estimates by
using global $\delta^{15}N$ observations ($f_{gasN}$, Goll et al., 2017b):
$\dfrac{f_{denit} N_{inorg}}{f_{leach} N_{inorg} + f_{denit} N_{inorg}} = f_{gasN}$         (D11)
**7    Appendix E Solutions under steady-state assumption**
$C_i = F_c \gamma_{C,i} \tau_i \ \ (i = 1,2,3)$         (E1-E4)
$C_4 = \left[ \dfrac{C_1}{\tau_1} \left(1 - f_{fireC,1}\right) + \dfrac{C_2}{\tau_2} \left(1 - f_{fireC,2}\right) \right] \tau_4$         (E5)
$C_5 = \left[ \dfrac{C_3}{\tau_3} \left(1 - f_{fireC,3}\right) + \dfrac{C_4}{\tau_4} \left(1 - f_{fireC,4}\right) \right] \tau_5$         (E6)
$F_N = F_c \left[ \rho_{N,1} \gamma_{C,1} \left(1 - f_{fireC,1}\right)\left(1 - \varepsilon_{N,1}\right) + \rho_{N,1} \gamma_{C,1} f_{fireC,1} + \rho_{N,2} \gamma_{C,2} + \rho_{N,2} \gamma_{C,3} \right]$         (E7)
$\gamma_{N,2} = \dfrac{\rho_{N,2} C_2}{\tau_2 F_N}$         (E8)
$\gamma_{N,3} = \dfrac{\rho_{N,3} C_3}{\tau_3 F_N}$         (E9)
$\gamma_{N,1} = 1 - \gamma_{N,2} - \gamma_{N,3}$         (E10)
$N_i = \rho_{N,i} C_i \ \ (i = 1,2,3,5)$         (E11-E14)
$N_4 = \dfrac{\frac{\rho_{N,1} C_1}{\tau_1}(1 - f_{fireC,1}) + \frac{\rho_{N,2} C_2}{\tau_2}(1 - f_{fireC,2})}{\frac{C_1}{\tau_1}(1 - f_{fireC,1}) + \frac{C_2}{\tau_2}(1 - f_{fireC,2})} C_4$         (E15)
$N_{imorb} = \eta \left( \rho_{N,5} - \dfrac{N_4}{C_4} \right) \dfrac{C_4}{\tau_4} + \left( \rho_{N,5} - \dfrac{N_3}{C_3} \right) \dfrac{C_3}{\tau_3} \left(1 - f_{fireC,3}\right)$         (E16)
$N_{inorg} = \dfrac{N_d + N_{fix} - \sum_{i=1}^{5}\left(\frac{N_i}{\tau_i} f_{fireC,i}\right)}{f_{leach}}$         (E17)
$F_P = F_c \left[ \rho_{P,1} \gamma_{C,1} \left(1 - f_{fireC,1}\right)\left(1 - \varepsilon_{P,1}\right) + \rho_{P,1} \gamma_{C,1} f_{fireC,1} + \rho_{P,2} \gamma_{C,2} + \rho_{P,2} \gamma_{C,3} \right]$         (E18)
$\gamma_{P,2} = \dfrac{\rho_{P,2} C_2}{\tau_2 F_P}$         (E19)
$\gamma_{P,3} = \dfrac{\rho_{P,3} C_3}{\tau_3 F_P}$         (E20)
$\gamma_{P,1} = 1 - \gamma_{P,2} - \gamma_{P,3}$         (E21)
$P_i = \rho_{P,i} C_i \ \ (i = 1,2,3,5)$         (E22-E25)
$P_4 = \dfrac{\frac{\rho_{P,1} C_1}{\tau_1}(1 - f_{fireC,1}) + \frac{\rho_{P,2} C_2}{\tau_2}(1 - f_{fireC,2})}{\frac{C_1}{\tau_1}(1 - f_{fireC,1}) + \frac{C_2}{\tau_2}(1 - f_{fireC,2})} C_4$         (E26)
$P_{imorb} = \eta \left( \rho_{P,5} - \dfrac{P_4}{C_4} \right) \dfrac{C_4}{\tau_4} + \left( \rho_{P,5} - \dfrac{P_3}{C_3} \right) \dfrac{C_3}{\tau_3} \left(1 - f_{fireC,3}\right)$         (E27)

1 $$P_{inorg} = \frac{P_d + P_w - \sum_{i=1}^{5}(\frac{P_i}{\tau_i} f_{fireC,i})}{f_{leach} f_{dissolve} + f_{sorb}}$$ (E28)

## Acknowledgement

The idea of GOLUM was initially discussed at a workshop held at the Northwest Agricultural and Forestry, China. We are grateful for the financial support of the workshop by the State Key Laboratory of Soil Erosion and Dryland Farming on the Loess Plateau of Northwest A & F University and the National Basic Research Programme of China grant 2013CB956602. Funding was provided by the Laboratory for Sciences of Climate and Environment (LSCE), CEA, CNRS and UVSQ. PC, DG, SP, JP and JS acknowledge support from the European Research Council Synergy grant ERC-2013-SyG-610028 IMBALANCE-P. Contribution by AAB was carried out at the Jet Propulsion Laboratory, California Institute of Technology, under a contract with the National Aeronautics and Space Administration. BDS was funded by ERC H2020-MSCA-IF-2015, grant number 701329.

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

**Table 1** Global spatially explicit observation-based estimates of model variables used as input data sets and the unknowns estimated in this study (including the symbols for each
 variable/parameter).

| Variable | Definition | Description | Computation method | References |
|---|---|---|---|---|
| **Inputs: carbon cycle** | | | | |
| $F_c$ | NPP | Spatially resolved model-data fusion estimates | Input | CARDAMOM; Bloom et al., 2016 |
| $\tau_{i=1,2,3,4,5}$ | Residence time of foliage, fine roots, wood, litter and SOM | Spatially resolved model-data fusion estimates | Input | CARDAMOM; Bloom et al., 2016 |
| $\gamma_{C, i=1,2,3}$ | Fraction of NPP allocated to foliage, fine roots and wood | Spatially resolved model-data fusion estimates | Input | CARDAMOM; Bloom et al., 2016 |
| $f_{fireC, i=1,2,3,4,5}$ | Fraction of fire to total outgoing flux from foliage, fine roots, wood, litter and SOM | Spatially resolved model-data fusion estimates | Input | CARDAMOM; Bloom et al., 2016 |
| $\eta$ | Fraction of litter outflux that enters SOM | Spatially resolved model-data fusion estimates | Input | CARDAMOM; Bloom et al., 2016 |
| **Inputs: nitrogen cycle** | | | | |
| $\rho_{N, i=1,2,3,5}$ | N:C ratio in foliage, fine roots, wood and SOM | Biome-scale synthesis based on *in situ* measurements | Input | Zechmeister-Boltenstern et al., 2015 |
| $f_{leach}$ | Fraction of inorganic N (or P) lost due to leaching (Eq. B7) | Spatially resolved reanalysis by model; Model result, scaled to match measurements | Input | Balsamo et al., 2015 Fekete et al., 2002 |
| $\varepsilon_{N, 1}$ | Resorption coefficient of N in foliage | Biome-scale synthesis based on *in situ* measurements | Input | Zechmeister-Boltenstern et al., 2015 |
| $N_d$ | N deposition | Spatially resolved model result, scaled to match *in situ* measurements | Input | Wang et al., 2017 |
| $N_{fix}$ | N fixation | Spatially resolved model result, scaled to match the estimates of NPP and N:C ratios | Input | Peng et al., submitted |
| $f_{gas}$ | Fraction of denitrification to the total loss of inorganic N | Spatially resolved process-based statistical model result | Input | Goll et al., 2017b |
| **Inputs: phosphorus cycle** | | | | |
| $\rho_{P, i=1,2,3,5}$ | P:C ratio in foliage, fine roots, wood and SOM | Biome-scale synthesis based on *in situ* measurements | Input | Zechmeister-Boltenstern et al., 2015 |
| $f_{dissolved}$ | Fraction of labile soil P that is dissolved in the soil water | *In situ* measurements, averaged based on soil order | Input | Yang and Post, 2011 |
| $f_{sorb}$ | Fraction of inorganic P that is transformed to strongly sorbed P | Assumed constant | Input | Goll et al., 2017a |
| $\varepsilon_{P, 1}$ | Resorption coefficient of P in foliage | Biome-scale synthesis based on *in situ* measurements | Input | Zechmeister-Boltenstern et al., 2015 |
| $P_d$ | P deposition | Spatially resolved model result, scaled to match *in situ* measurements | Input | Wang et al., 2017 |
| $P_d$ | P weathering | Spatially resolved model result, scaled to match observed data | Input | Hartmann et al., 2014 |
| **Unknowns estimated from mass balance assuming steady state** | | | | |
| $C_{i=1,2,3,4,5}$ | C pool of foliage, fine roots, wood, litter and SOM | Pools | Based on steady-state assumption | |

| | | | |
|---|---|---|---|
| $F_N$ | N uptake from inorganic-N pool by vegetation | Flux | Mass balance approach based on NPP (input) and stoichiometry ratios (input) |
| $\gamma_{N, i=1,2,3}$ | Fraction of $F_N$ allocated to foliage, fine roots and wood | Allocation fractions | Mass balance approach based on NPP (input) and stoichiometry ratios (input) |
| $N_{i=1,2,3,4,5}$ | N in foliage, fine roots, wood, litter and SOM | Pools | Mass balance approach based on stoichiometry ratios (input) and steady-state C pools ($C_{i=1,2,3,4,5}$), assuming N:C ratios do not change over time |
| $N_{imob}$ | N immobilization flux | Pools | Based on steady-state assumption that stoichiometry ratios (input), litter C and soil C do not change at annual scale |
| $f_{denit}$ | Annual denitrification rate | Rate | Mass balance approach, assuming annual mean inorganic N pool size does not change at annual scale |
| $N_{inorg}$ | Inorganic-N pool | Pool | Based on steady-state assumption that inorganic N do not change at annual scale |
| $F_P$ | P uptake from inorganic-P pool by vegetation | Flux | Mass balance approach based on NPP (input) and stoichiometry ratios (input) |
| $\gamma_{P, i=1,2,3}$ | Fraction of Fp allocated to foliage, fine roots and wood | Allocation fractions | Mass balance approach based on NPP (input) and stoichiometry ratios (input) |
| $P_{i=1,2,3,4,5}$ | P in foliage, fine roots, wood, litter and SOM | Pools | Mass balance approach based on stoichiometry ratios (input) and steady-state C pools ($C_{i=1,2,3,4,5}$) |
| $P_{imob}$ | P immobilization flux | Flux | Based on steady-state assumption that stoichiometry ratios (input), litter C and soil C do not change at annual scale |
| $P_{inorg}$ | Inorganic-P pool | Pool | Based on steady-state assumption that labile P do not change at annual scale |

1  **Table 2** Global annual mean C-pool sizes, NPP and heterotrophic-respiration fluxes in the C-cycle model assuming steady
2  states under the climate conditions of 2001-2010, compared to the means and percentile ranges from the original
3  CARDAMOM results during 2001-2010.

| | This study | Original CARDAMOM | | | | |
| --- | --- | --- | --- | --- | --- | --- |
| | | 5th percentile | 25th percentile | Mean | 75th percentile | 95th percentile |
| Foliage-pool size (Pg C) | 23 | 3.2 | 7 | 15 | 21 | 34 |
| Fine-root-pool size (Pg C) | 27 | 1.9 | 5 | 18 | 25 | 56 |
| Wood-pool size (Pg C) | 493 | 193 | 364 | 755 | 984 | 1850 |
| Litter-pool size (Pg C) | 20 | 1.3 | 4 | 22 | 26 | 88 |
| SOM-pool size (Pg C) | 1421 | 749 | 1100 | 1557 | 1882 | 2771 |
| NPP (Pg C y$^{-1}$) | 52.5 | Not given | 39 | 52 | 63 | Not given |
| Fire (Pg C y$^{-1}$) | 1.5 | Not given | 1.3 | 1.7 | 2.0 | Not given |
| Heterotrophic respiration (Pg C y$^{-1}$) | 51 | Not given | 37 | 54 | 67 | Not given |

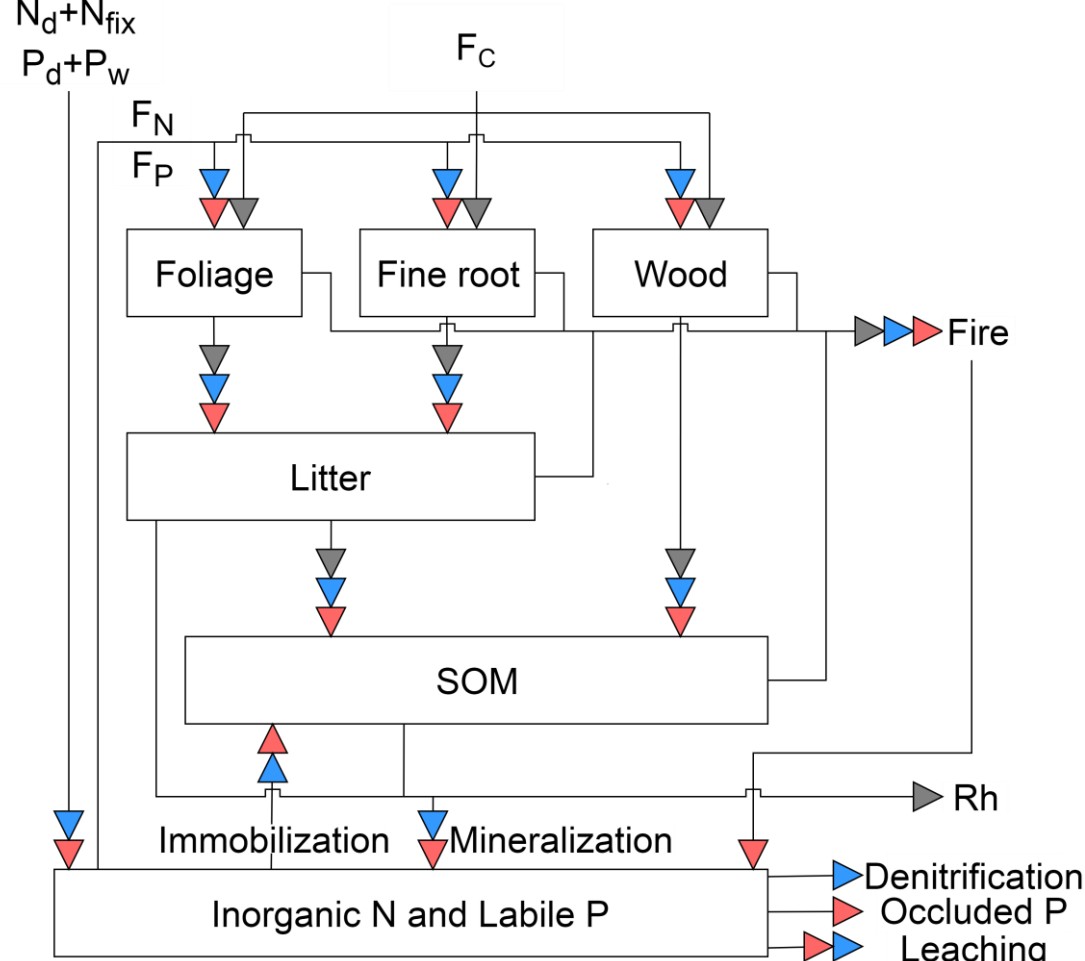

**Figure 1** Schematic representation of the pools and fluxes in the C, N and P cycles within GOLUM-CNP. The gray, blue and
red arrows represent C, N and P fluxes, respectively. Plants are divided into foliar, fine root and wood pools, where the wood
pool includes woody stems and coarse roots. Litter and soil are two separate pools. The inorganic pool represents the nutrient
sources in the soil that are available for plant uptake. Arrows between the pools represent the directions of C, N and P flow
between pools. External inputs of N are atmospheric deposition ($N_d$) and biological N fixation ($N_{fix}$). External inputs of P are
atmospheric deposition ($P_d$) and P released by rock weathering ($P_w$). $F_C$ is net primary production (NPP). $F_N$ and $F_P$ are plant
uptake of N and P from the inorganic N and labile P pools, respectively. $R_h$ is release of C due to heterotrophic respiration.
Mineralization of N and P is modeled along with litter and SOM decomposition, and N and P immobilization is modeled by
a flux from the inorganic pool to SOM. External losses of N occur by fire, leaching and denitrification. External losses of P
occur by fire, leaching and transfer to occluded P in the soil.

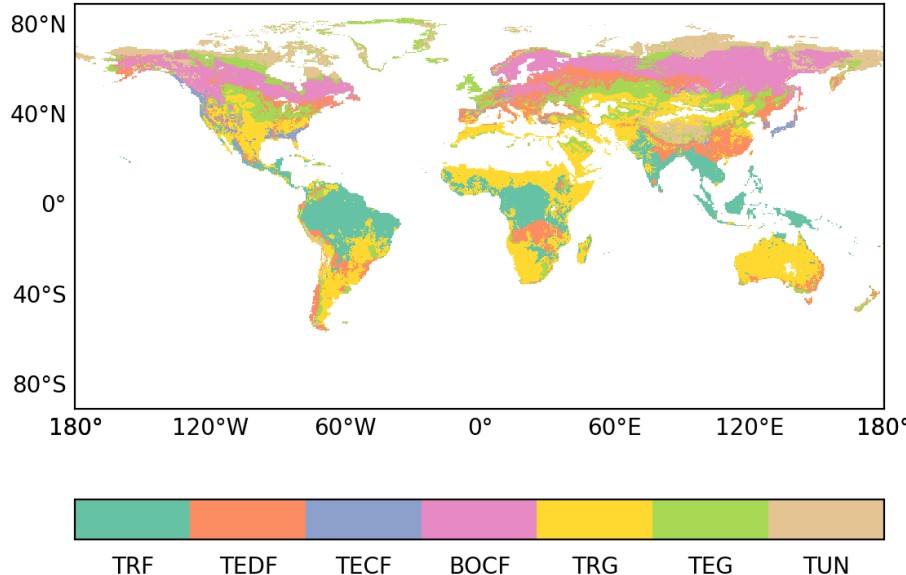

**Figure 2** ESA CCI land-cover map classified into the seven large biomes for which average N:C and P:C ratios for each carbon pool are available, at 0.25 °×0.25 ° resolution: tropical rainforests (TRF), temperate deciduous forests (TEDF), temperate coniferous forests (TECF), boreal coniferous forests (BOCF), tropical/C4 grasslands (TRG), temperate/C3 grasslands (TEG) and tundra (TUN).

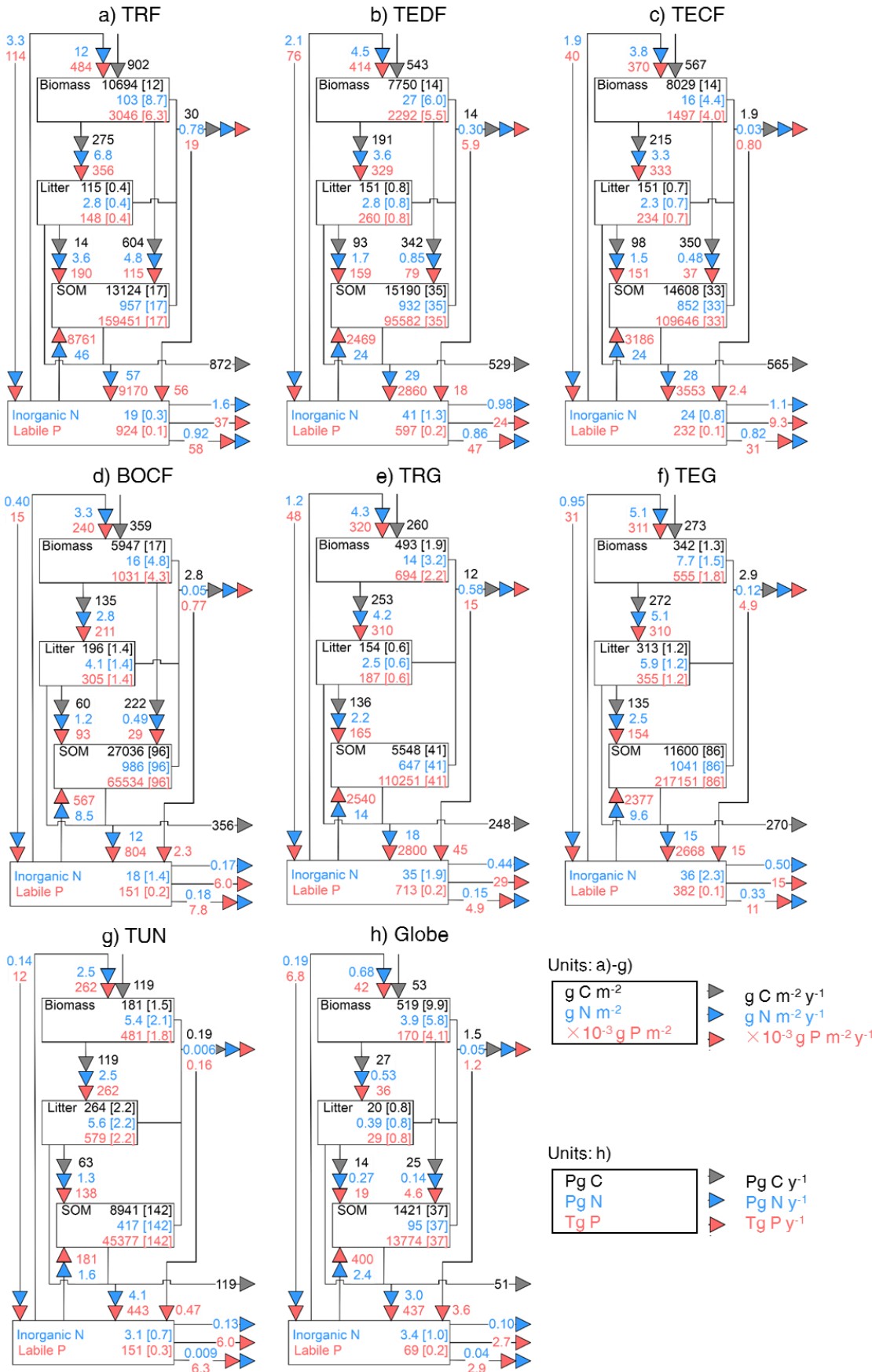

**Figure 3** Fluxes (numbers along arrows), mean residence times (in parentheses) and pool sizes of the N (blue) and P (red)
cycles in the terrestrial biosphere at steady state for the large biomes (a-g) and globe (h). The targeted biomes are tropical
rainforests (TRF, a), temperate deciduous forests (TEDF, b), temperate coniferous forests (TECF, c), boreal coniferous
forests (BOCF, d), tropical/C4 grasslands (TRG, e), temperate/C3 grasslands (TEG, f) and tundra (TUN, g).

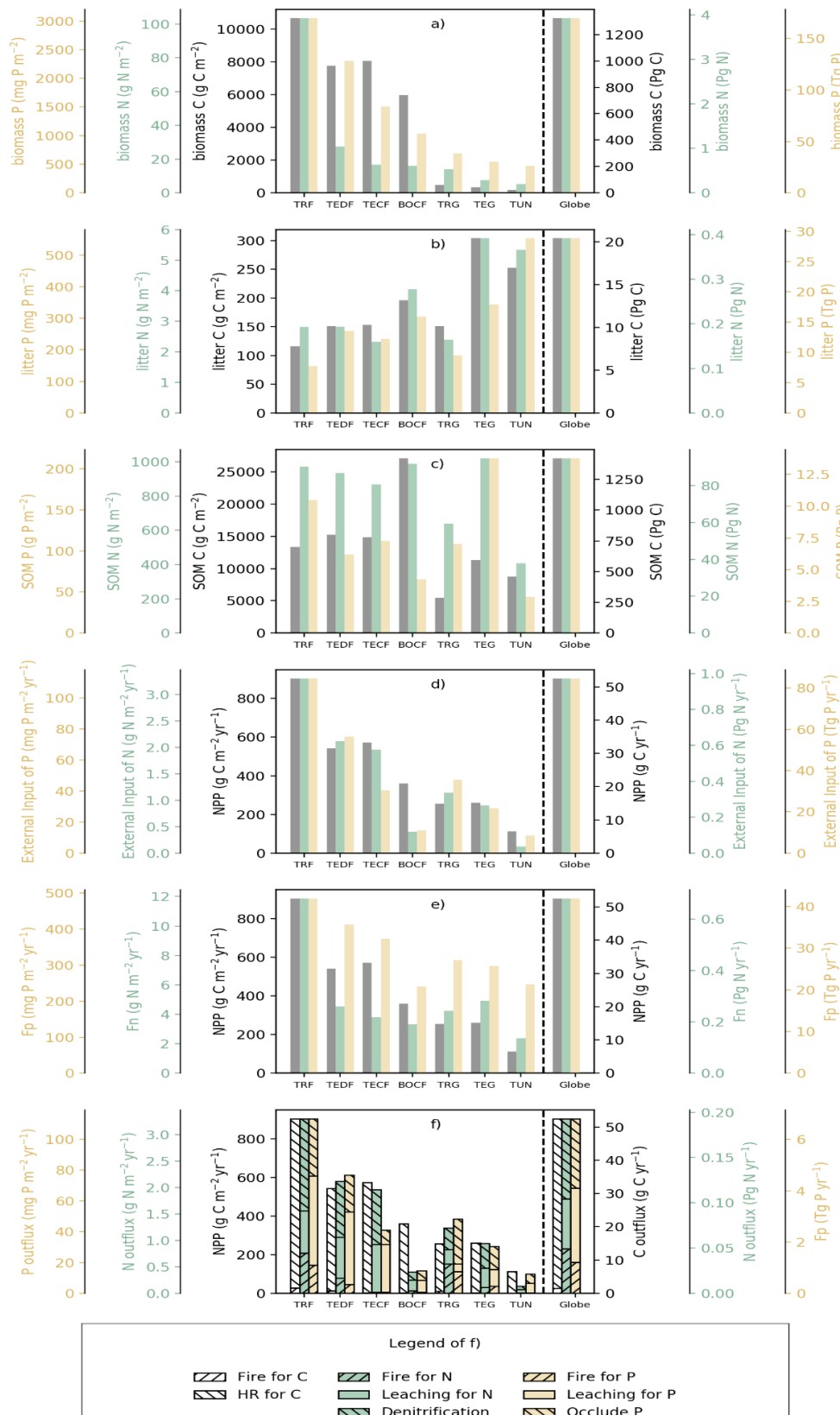

2 **Figure 4** Pool sizes and fluxes of C (black), N (green) and P (yellow) computed from GOLUM-CNP.

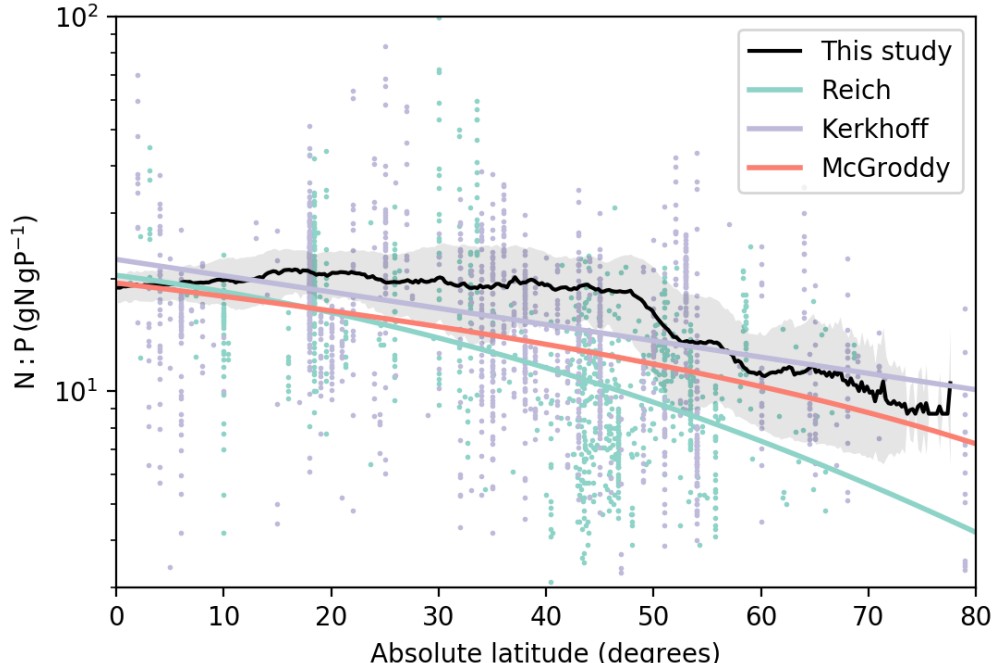

**Figure 5** Relationship between foliar N:P ratios (gN gP$^{-1}$) and absolute latitude. The black line is the mean N:P ratios from
this study, and the shaded area is the one-sigma standard deviation of the N:P ratios for specific latitude. Colored lines are
the regression trends of foliar N:P ratios as a function of absolute latitude from Reich and Oleksyn (2004; green), Kerkhoff
et al. (2005; blue) and McGroddy et al. (2004; red). Dots are the raw data that Reich and Oleksyn (2004; green) and
Kerkhoff et al. (2005; blue) used to derive their regression trends.

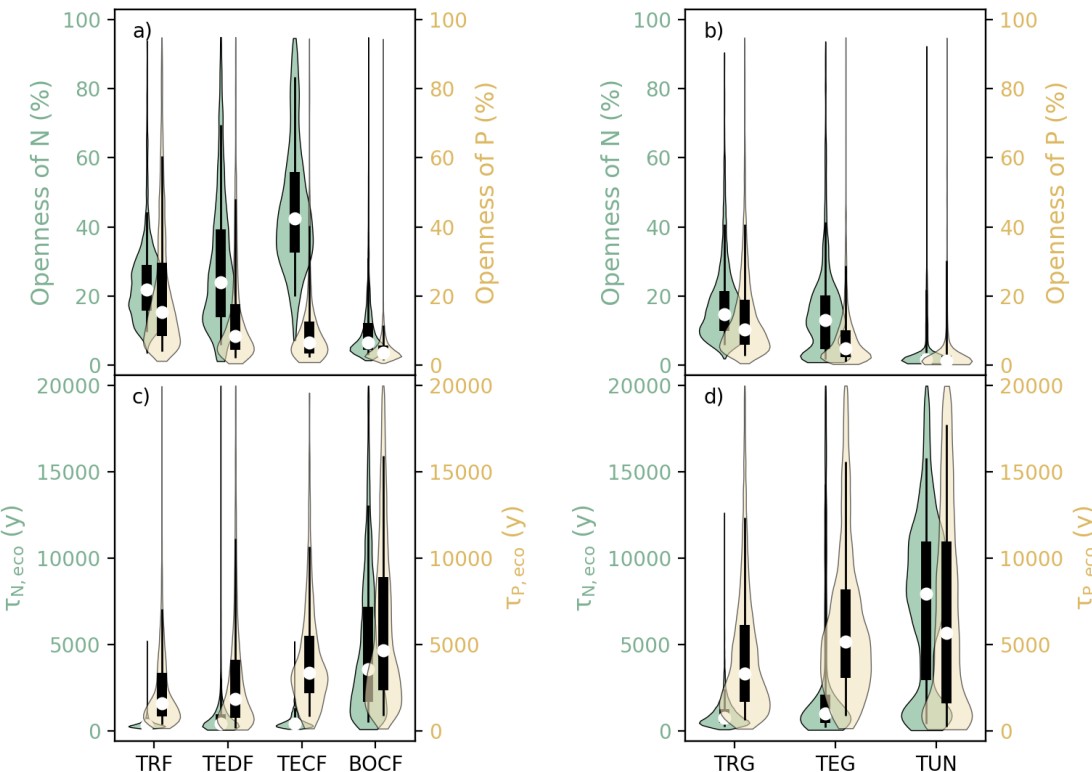

**Figure 6** Violin plots of the openness of N and P cycling (the percentage of total plant uptake of N and P attributed to new nutrient inputs) for a) forest and b) grassland biomes. Residence times of N ($\tau_{N,eco}$) and P ($\tau_{P,eco}$) in c) forest ecosystems and d) grassland biomes. Open circles are medians of all grid cells within each biome, with balloons representing the probability density distribution of each value. Black whiskers indicate interquartile (thick) and 95% confidence intervals (thin). The biomes are tropical rainforests (TRF), temperate deciduous forests (TEDF), temperate coniferous forests (TECF), boreal coniferous forests (BOCF), tropical/C4 grasslands (TRG), temperate/C3 grasslands (TEG) and tundra (TUN).

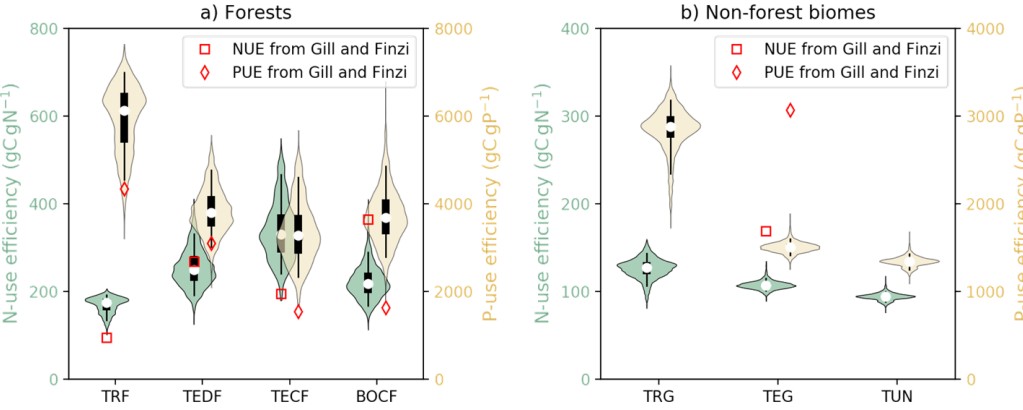

**Figure 7** Violin plots of N- and P-use efficiencies (NUE and PUE, the nutrient uptake by plants divided by GPP) of seven biomes. Open circles are medians of all grid cells within each biome, with balloons representing the probability density distribution of each value. Black whiskers indicate interquartile (thick) and 95% confidence intervals (thin). a) Forest biomes, including tropical rainforests (TRF), temperate deciduous (TEDF), temperate coniferous (TECF) and boreal coniferous forests (BOCF). b) Grassland biomes, including tropical/C4 (TRG), temperate/C3 grasslands (TEG) and tundra (TUN). Red squares (NUE) and diamonds (PUE) are the independent estimates from site observations and other generic data sets compiled and harmonized by Gill and Finzi (2016) based on site measurements of GPP and net N/P mineralization.

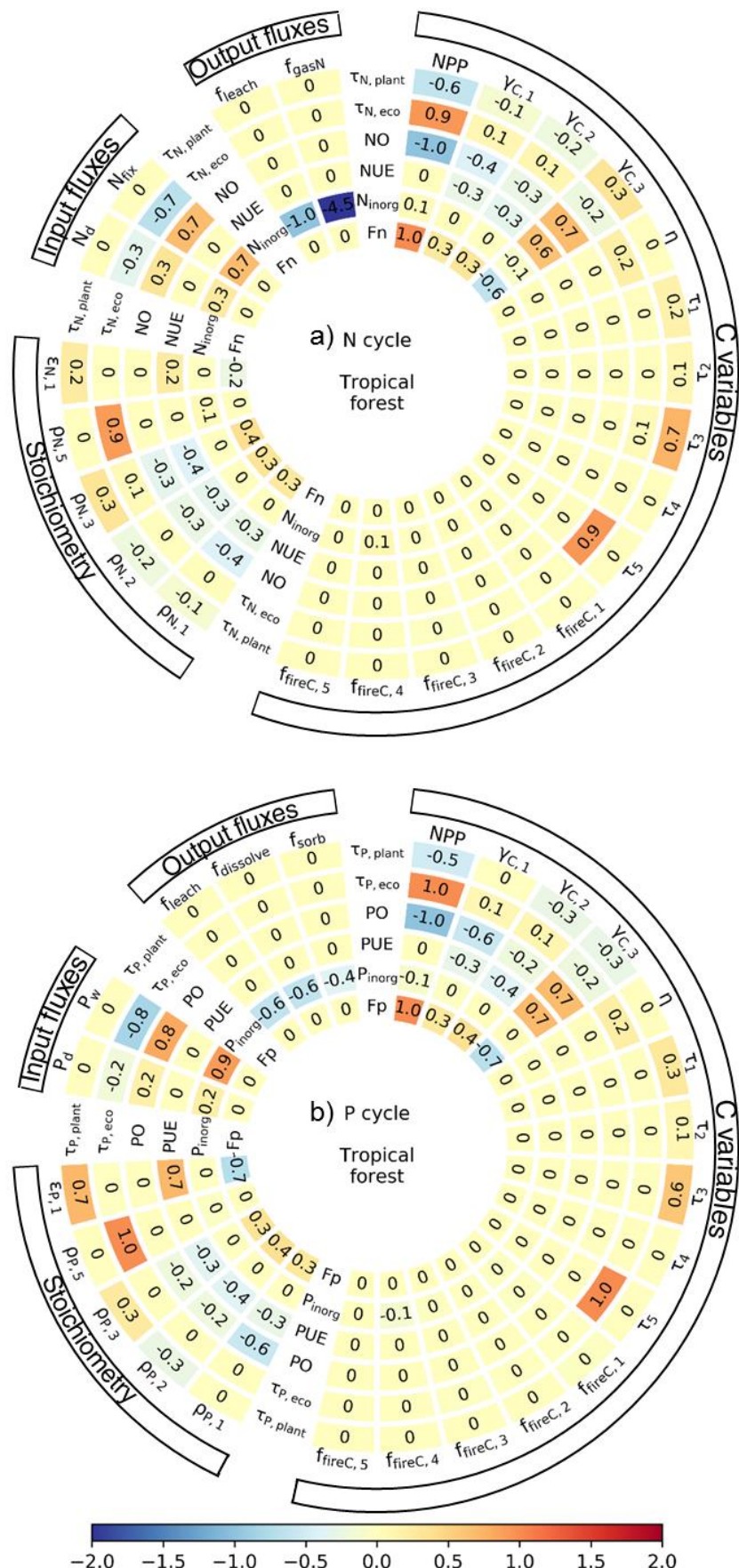

**Figure 8** Mean **s**ensitivity of the estimates of rates of nutrient uptake, inorganic nutrients, nutrient-use efficiencies, openness, turnover time of nutrients in the ecosystem and turnover time of nutrients in plants to the input variables for tropical forest. Results for other biomes are shown in Figs. S7-S12.

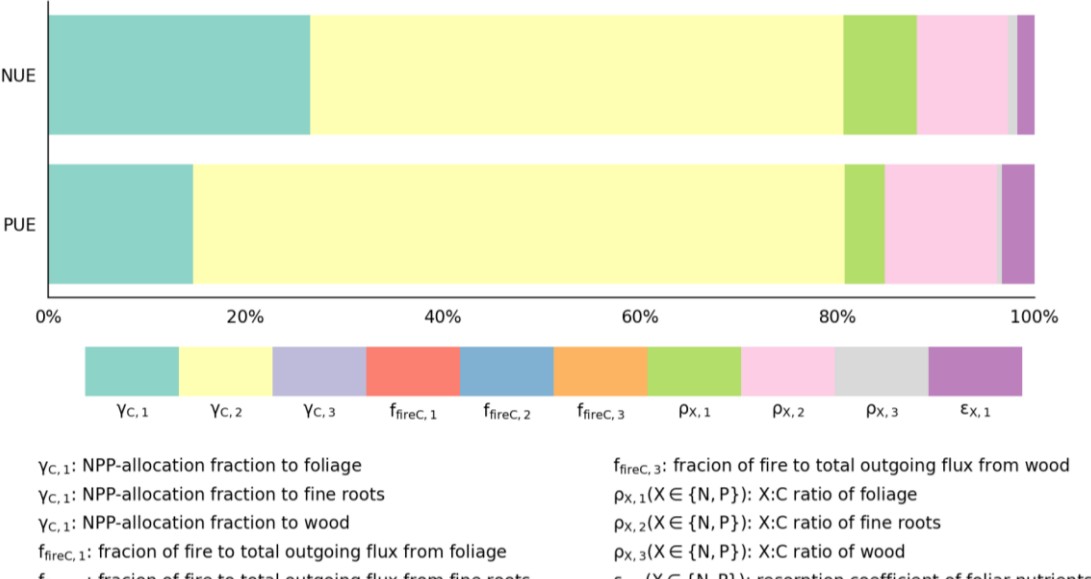

$\gamma_{C,1}$: NPP-allocation fraction to foliage
$\gamma_{C,1}$: NPP-allocation fraction to fine roots
$\gamma_{C,1}$: NPP-allocation fraction to wood
$f_{fireC,1}$: fracion of fire to total outgoing flux from foliage
$f_{fireC,2}$: fracion of fire to total outgoing flux from fine roots

$f_{fireC,3}$: fracion of fire to total outgoing flux from wood
$\rho_{X,1}(X \in \{N,P\})$: X:C ratio of foliage
$\rho_{X,2}(X \in \{N,P\})$: X:C ratio of fine roots
$\rho_{X,3}(X \in \{N,P\})$: X:C ratio of wood
$\varepsilon_{X,1}(X \in \{N,P\})$: resorption coefficient of foliar nutrients

**Figure 9** Contribution of input data to the variance of the estimates of nutrient-use efficiencies ($X \in \{N,P\}$) for temperate coniferous forests. $\gamma_{C, i=1,2,3}$ are NPP-allocation fractions to foliage, fine roots and wood, respectively. $f_{fireC, i=1,2,3}$ are fractions of fire to total outgoing flux from foliage, fine roots and wood, respectively. $\rho_{x, i=1,2,3}$ ($X \in \{N,P\}$) are X:C ratios of foliage, fine roots and wood, respectively. $\varepsilon_{X, 1}$ ($X \in \{N,P\}$) is the resorption coefficient of foliar nutrients.

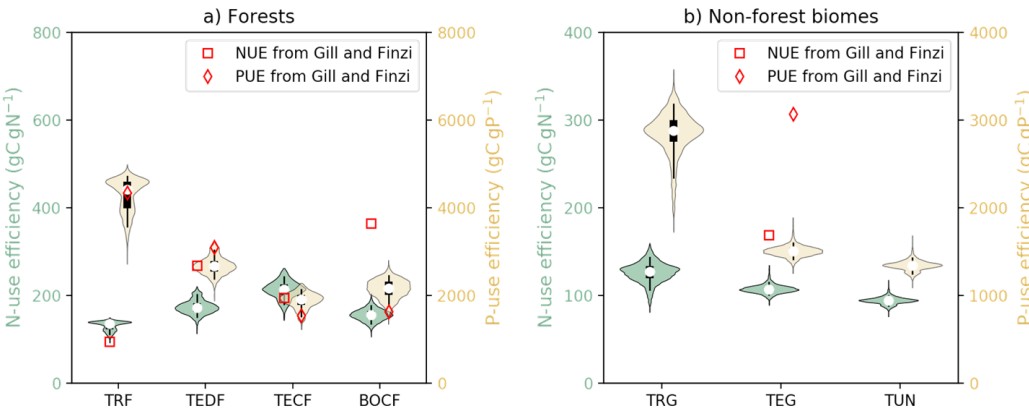

Figure 10 Violin plots of the nutrient-use efficiencies of the seven biomes from the experiment in which the allocation fraction of NPP to woody biomass and to leaves in coniferous forests is reduced. Open circles are the medians of all grid cells within each biome, with balloons representing the probability density distribution of each value. Black whiskers indicate interquartile (thick) and 95% confidence intervals (thin). The biomes are tropical rainforests (TRF), temperate deciduous forests (TEDF), temperate coniferous forests (TECF), boreal coniferous forests (BOCF), tropical/C4 grasslands (TRG), temperate/C3 grasslands (TEG) and tundra (TUN). The red squares (NUE) and diamonds (PUE) are the independent estimates from site observations and other generic data sets compiled and harmonized by Gill and Finzi (2016) based on site measurements of GPP and net N/P mineralization.