# Peer review of "GOLUM-CNP v1.0: a data-driven modeling of carbon, nitrogen and 1"

_Geoscientific Model Development, 2018_

## Referee Comment (RC1) · Anonymous Referee #1 · 15 May 2018

Overall this is an interesting and detailed summary of an improvement to an existing model. Though not the first to put in a N cycle, the P cycle is relatively novel and there is clearly diligent work done by the authors to ensure values are appropriately backed up by data where possible. There are a few points that I think need clarifying and some aspects of the model that seem to me a bit odd, and therefore need provisos about the appropriate (or inappropriate) use of the model. Some of the conclusions about openness are a bit of a stretch given the model setup. But with some extra information discussing the limitations, this will be a worthwhile model description.

Major points

[Figure]

The issue of this being an equilibrium model for the present day/recent past is a concern to me. It isn't sufficiently explained how an equilibrium estimate (including anthropogenic N deposition) is valid in equilibrium. I can see the justification if it's pre-industrial (excluding anthropogenic N deposition), but it doesn't make sense to me as it is. In particular, the openness of the system seems to me to be almost completely determined by the assumption of equilibrium. If the outputs and inputs are balanced (i.e. the equations are solved to 0, as is stated on P7 L32), then surely the store size is at least partly determined by something we know to be wrong. Given that the equilibrium assumption should increase the carbon storage, it's odd that in Table 2 (assuming these are present-day values), the NPP and the soil and vegetation pools are all smaller than many other models and global estimates suggest.

Reading between the lines, it seems that the N fixation is about 120Tg/year. How does this square with other estimates, e.g. Vitousek et al. 2013 (44Tg/year)? Since it's not discussed where this N fixation number came from (and the reference isn't available), or where the N deposition number came from, it makes it difficult to give much credibility to the openness discussions which rely on these.

The relationship between GOLUM-CNP and CARDOMON is opaque and needs to be clarified. It is particularly unclear with regard to what the relationship between the code provided and CARDOMON is. For example, does this code work independently? Or does it need CARDOMON to run? If CARDOMON is part of the code provided, which parts are new and which are CARDOMON?

It's not explained what the intended use of this model is. It's essential early on in the paper to have some examples of use, as well as specific limits on what it shouldn't be used for, (particularly given the limitation of it being an equilibrium model). This is slightly covered right at the end of the paper, but needs to be earlier and more extensive.

The code is very dense, making it very difficult to read. Code should have comments

every 1 - 10 lines, depending on how interpreted/dense, etc. the code is. Ideally, code should be commented so that, if you stripped out the actual code or if you didn't know python at all, you could re-write it in another language just from the comments. I was also a bit surprised not to see any functions used.

Minor points

P.2 L6-7. Yes, but also water, light etc. are essential controls - if there's no light and water it doesn't matter how much N or P there is, nothing will grow.

P.2 L17 - 30. This seems to mix up P and N fertiliser. It would be better to keep the two issues separate wherever possible.

P18. L28. Reference to Peng missing.

P21. Table 2. When is this table referring to? (Pre-industrial? Present day?) It needs to be specified.

P22. Figure 1. The caption would be more useful if the terms at the top of the figure were defined first, and then worked downwards.

P24. Figure 3. This is a very difficult to read. A table or a series of bar plots would be much better.

P28. Figure 7. Two points need to be addressed for this figure. First that the text is so small that it is impossible to read printed A4. That's true of the ones in the SI too. Second that it's ironic that a red-green color scheme is used, despite one of the authors being color-blind. I just... Other color schemes are available.

P29. Figure 8. It would be a courtesy to your readers to include in the key what YC1, etc. are. It could literally just go beneath the current labels.

---

## Referee Comment (RC2) · Anonymous Referee #2 · 15 May 2018

This manuscript presented a simple model for coupled terrestrial carbon-nitrogen-phosphorus cycles. A model-data fusion approach was applied to estimate steady-state values for selected model state variables, fluxes, and parameters. In particular, the estimated quantities allowed for computation of carbon, nitrogen, and phosphorus pool sizes, the openness of the N and P cycles, characteristic turnover times, and nutrient use efficiencies.

In principle, I think that estimation of these quantities using model-data fusion is a worthwhile and interesting idea. However, I think there is substantial room for improvement in both the presentation and the approaches used. Here are a few general

comments:

1. The paper contains no statement of objectives, questions, or hypotheses. Including any or all of these features would help to orient the reader. But without them, the paper is difficult to evaluate because the goals of the paper are unstated.

2. The model is meant to be applied to natural ecosystems, and so the model-data fusion approach should be valid in grid cells that are dominated by nature ecosystems. However, the problem is that very few such grid cells remain (Ellis and Ramankutty 2008). I would therefore expect biases to emerge when the natural ecosystem model is applied to grid cells that are biogeochemically influenced by humans. I am bothered that anthropogenic effects are not accounted for in the global and (natural) biome scale averages computed by the authors.

3. In a way, not enough information is presented. I would be interested in seeing spatial maps of at least some quantities and some confirmation that the model-data fusion approach is satisfactory on the grid cell level. I am skeptical of the biome- and global-level results without having a better idea of the grid cell level information used to construct them.

4. In another way, too much information is presented, making the paper confusing (see Figs. 3, 7). It would help to have a better-defined narrative arc. While many numbers were presented, I think the authors could do a better job in drawing attention to new, qualitative insights.

5. I would like to know approximately what level of bias the steady-state assumption incurs. These ecosystems are unlikely to be at steady state for many reasons ($CO_2$ fertilization, changes in nutrient deposition, climate extremes such as droughts, etc.).

6. The model is simple, which is okay for a first attempt. But there does not seem to be any substantial discussion of how to make the model more realistic by including other known fluxes and feedbacks (especially appropriate in 5.3, Future research and data

needs).

7. The manuscript needs a careful proofreading by a fluent English speaker. Starting with the title, "model" would make more sense than "modeling".

Specific comments Page 2, lines 17-21: Not clear if you are talking about N cycling, P cycling, or both Page 2, lines 28-30: Too bad that this is not discussed further Page 3, lines 17-19, state that an inorganic P pool was added to the model, and it is assumed that this P is accessible to plants. However, in reality, very little inorganic P is accessible to plants. Page 3, line 31: What do you think about N from rock weathering (Houlton et al. 2018)? Page 3, lines 37-38: N fixation is not entirely controlled by plants. In fact, the authors mentioned asymbiotic fixation a few lines above. Page 4, lines 21-22: much inorganic P is neither strongly sorbed, nor immediately available to plants. See, for example, Yang et al. (2013). Page 5, lines 2-6: why the biome-scale analysis? Soil parent material is perhaps the most important factor governing nutrient limitation (Augusto et al. 2017) and has substantial sub-biome scale variability. Page 5, lines 12-13: I am concerned at the ECMWF soil moisture product. Is it any good? Doesn't it have fixed layer depths? I'm not sure if that is appropriate here. Page 8, lines 3-4: There is not much take-away here. I guess it is not too surprising that the number falls within the large range. But what is the range of your computed number? Is the ranage reduced or increased compared to CARDAMOM? Page 8, lines 16-19: All of this seems like speculation, more appropriate for a "discussion" section than for "Results". Page 11, lines 4-18: There are also a fair amount of "results" in the "Discussion" section. Furthermore, these are some very ad hoc ways of making corrections when unexpected results are obtained. Page 11, lines 40-41: I do not see where this was shown.

Augusto, L., D. L. Achat, M. Jonard, D. Vidal, and B. Ringeval. 2017. Soil parent material—A major driver of plant nutrient limitations in terrestrial ecosystems. Global Change Biology.

Ellis, E. C., and N. Ramankutty. 2008. Putting people in the map: anthropogenic biomes of the world. Frontiers in Ecology and the Environment 6:439-447.

Houlton, B., S. Morford, and R. Dahlgren. 2018. Convergent evidence for widespread rock nitrogen sources in Earth's surface environment. Science 360:58-62.

Yang, X., W. M. Post, P. E. Thornton, and A. Jain. 2013. The distribution of soil phosphorus for global biogeochemical modeling. Biogeosciences 10:2525.

---

## Author Comment (AC1) · 25 Jul 2018

**Response to comments on "GOLUM-CNP v1.0: a data-driven modeling of carbon, nitrogen and phosphorus cycles in major terrestrial biomes" by Y. Wang et al.**

We thank the referee for reviewing our manuscript. Please find attached a point-by point reply to each of the comments raised by the referee with legible text and figures organized along the text.

Overall this is an interesting and detailed summary of an improvement to an existing model. Though not the first to put in a N cycle, the P cycle is relatively novel and there is clearly diligent work done by the authors to ensure values are appropriately backed up by data where possible. There are a few points that I think need clarifying and some aspects of the model that seem to me a bit odd, and therefore need provisos about the appropriate (or inappropriate) use of the model. Some of the conclusions about openness are a bit of a stretch given the model setup. But with some extra information discussing the limitations, this will be a worthwhile model description.

**Response:**

We would like to thank the referee for the valuable comments and suggestions for improving our manuscript. Following the reviewer's comments, we carefully revised our manuscript. Please find below the point-to-point responses (in black) to all referee comments (in blue). For your convenience, changes in the revised manuscript are highlighted with dark red. All the pages and line numbers correspond to the original version of text. Of note GOLUM-CNP is not an improvement to an existing model, but an independent model that uses the outputs from CARDAMOM and incorporate the data-driven estimates of N and P cycles (see detailed response to Comment 3).

**Major points**

1. The issue of this being an equilibrium model for the present day/recent past is a concern to me. It isn't sufficiently explained how an equilibrium estimate (including anthropogenic N deposition) is valid in equilibrium. I can see the justification if it's preindustrial (excluding anthropogenic N deposition), but it doesn't make sense to me as it is. In particular, the openness of the system seems to me to be almost completely determined by the assumption of equilibrium. If the outputs and inputs are balanced (i.e. the equations are solved to 0, as is stated on P7 L32), then surely the store size is at least partly determined by something we know to be wrong. Given that the equilibrium assumption should increase the carbon storage, it's odd that in Table 2 (assuming these are present-day values), the NPP and the soil and vegetation pools are all smaller than many other models and global estimates suggest.

**Response:**

We develop a steady-state model as a first attempt of a CNP data-driven diagnostic model. The objective of this study is not to reproduce the present day state, but to provide a steady-state estimates of C, N and P cycle with current input of C, N and P, stoichiometry of

N:C and P:C ratios, and residence times of different pools, because there is a lack of data to constraint a transient state (see point 1). As a result, we choose to constrain an equilibrium state. Although it does not correspond in some aspects to the present day state, it is still useful for evaluating global models as these model are able to simulate a equilibrium state for present day conditions and a direct comparison is possible (see point 2 and point 3).

- 1) Although we described processes in the C, N and P cycles by a set of differential equations (Eqs. A1-A5, B1-B7, C1-C7), only few global long-term observations associated with N and P were available to constrain a transient simulation. For example, the synthesis of stoichiometries in different pools are based on published literature during the last four decades, and almost no data are available for preindustrial period; field-scale manipulation experiments have shown that warming, elevated atmospheric CO2, and N and P fertilization can drive changes in stoichiometries and nutrient resorption in terrestrial ecosystems (Sistla and Schimel, 2012; Sardans et al., 2012; Sardans and Pe ñuelas, 2012; Mayor et al., 2014; Yang et al., 2014; Yuan and Chen, 2015; Sardans et al., 2016; Sardans et al., 2017), but these data are insufficient to infer these changes in terrestrial ecosystems during the past years. As a result, for a data-driven model framework, we computed the equilibrium estimates and this equilibrium corresponds to the state where the inputs of C, N and P (e.g. NPP for C cycle, N deposition and fixation for N cycle, P deposition and release of P by rock weathering for P cycle) and residence times equal to the estimates under present-day conditions.
- Some previous studies that have investigated the changes in C, N stocks between pre-2) industrial period and present day. Thornton et al. (2007) investigated the C stock changes from pre-industrial times and present day (1976-2000) using CLM-CN model. They found the C stocks in vegetation, litter and SOM increased by 35, 1 and 10 Pg, which equal to only 5%, 6% and 3% of the respective initial C stocks in 1850. Zaehle et al. (2010) estimated the C stock changes and N stock changes in vegetation and SOM, using the O-CN terrestrial biosphere model. By excluding the impact by land use change, that vegetation C stocks increased by 62 Pg and soil C stocks increased by 39 Pg from 1860 to 2002, being 13% and 3% of initial C stocks for vegetation and SOM respectively. The N stocks increased by 376 Tg and 2836 Tg in vegetation and SOM, being 11% and 3% of initial N stocks. Zaehle et al. (2013) further accounted for the effect of land use change on the global C and N cycles. The results also showed that the cumulative effect of anthropogenic disturbance from preindustrial times are small (<5%) on the natural C and N cycles. The evidences for the change of global P cycling from pre-industrial times are quite limited and highly uncertain (Goll et al., 2012). At local scales, some places are experiencing significant anthropogenic disturbances, mostly through deposition and fertilizer (Jornard et al., 2015). However, the extent to which natural ecosystems are affected by these disturbances are not clear (Wang et al., 2017). Given these state-of-theart estimates about the magnitude of anthropogenic disturbance on natural C, N and P cycles and large uncertainties, it is still useful to take steady-state estimates as a diagnostic to evaluate DGVMs in simulating global C, N and P cycling (Wang et al., 2010; Xu-ri et al., 2008; Yang et al., 2009; Lawrence et al., 2011). The estimates based on steady state assumptions listed above match well with a wide range of *in-situ* observations in recent years.

3) CARDAMOM is the only spatially explicit data-driven product of C cycle, which does not rely on a steady-state assumption. In this study, our aim is to develop a steady-state model (see point 1) in which a consistency between different datasets are achieved (which is a novelty). We chose to use the variables in the CARDAMOM products: NPP, residence times and fire fractions, which were constrained by global long-term observations of MODIS LAI and MODIS burned area. For other C variables whose direct constraints do not have a global coverage in CARDAMOM (such as biomass), we recalculated them based on the equilibrium assumption. With this steady state C cycle, we compute the associated N and P pools and fluxes. For example, the N and P in SOM pool equal to the product of N:C/P:C ratios and equilibrated C content.

In summary, given the three concerns mentioned above, we computed an equilibrium estimate of C, N and P cycle, where the inputs of C, N and P into the terrestrial ecosystems (NPP, N and P deposition, N fixation and release of P by rock weathering), stoichiometries of N:C and P:C ratios, the residence times of different pools and the land cover are held constant as the period 2001-2010. Although this steady-state estimates deviate from the actual state of present-day disturbed cycles, we compare the GOLUM-CNP with the original CARDAMOM to show that the steady-state estimates in GOLUM-CNP are still within the range of such large uncertainties from data on the present day state.

To address the referee's concern about steady-state assumption in a quantitative way, we compare the recomputed pool sizes for C with the original CARDAMOM estimates (Fig. S1 in the revised supporting information). The steady-state transformed pool sizes are within the [25, 75th] percentile range of the original CARDAMOM results at more than 90% forest grid cells. The major part of the deviation comes from the treatment of grassland in GOLUM and are not associated to the steady-state assumption: In GOLUM-CNP grasslands are considered as a distinct biome, while the original CARDAMOM did not use land cover information and provided some woody biomass pools for grassland dominated regions. As GOLUM does not consider woody biomass for grass, we transfer wood growth from CARDAMOM into nonwoody tissue for grasslands. This makes our C pools more different from the original CARDAMOM at grassland-dominated pixels than at forest-dominated pixels. However, although the biomass at grassland grid cells in GOLUM-CNP are much lower than the biomass in CARDAMOM, the litter and SOM pool sizes at grassland grid cells in GOLUM-CNP are still within the [5, 95th] percentile range of the original CARDAMOM results (Fig. S1b and S1c in the revised supporting information). In addition, we also mention in the revised manuscript: although GOLUM-CNP is presented for steady state in this study, the methods and equations used to compute fluxes and pools are generic and could be extended to non-steady state. In the future, when more data will become available, a transient version of GOLUM-CNP will incorporate the new data and improved understandings in C, N and P cycles.

We outline in the revised paper that some variables like the *openness* do not strongly rely on the equilibrium assumption. The openness is defined as:

$$XO = \frac{I_x}{F_x + RSB_x}$$
(1)

where  $I_x$  (X  $\in$  {N,P}) is the new nutrient inputs, i.e. deposition (Nd) and biological fixation  $(N_{fix})$  for N cycle and deposition  $(P_d)$  and rock weathering  $(P_w)$  for P cycle. These variables are in fact the nutrient inputs to the system, which are derived from observations and are not estimated from equilibrium assumption.  $F_x$ +RSBx represent the total uptake of nutrients by plant, which are determined by NPP, allocation fractions of NPP to different vegetation pools and stoichiometries in different vegetation pools. These variables are also the inputs to GOLUM-CNP and represent the non-steady state situation under current climate condition from original CARDAMOM without equilibrium assumption (as discussed above, only the C store sizes are computed based on equilibrium assumption). In consequence, this computation is actually based on a mass-balance framework and driven by fluxes as observed in present day transient state. However, we do assume that the stoichiometries of N:C and P:C ratios do not change significantly during the period considered (2001-2010), although small changes in stoichiometry in fast-turnover plant tissue is being observed on such a time scale (Jonard et al. 2015). Such an approach is common and is similar to that of Cleveland et al. (2013). In Cleveland et al. (2015), they computed NPPnew and NPPrecycle where N:C and P:C ratios are the same in these two types of NPP. In our study, we use new nutrient input (i.e.  $I_x$ ) and total nutrient uptake (i.e.  $F_x$ +RSBX) rather than two types of NPP.

Under the equilibrium assumption, conceptually, the pool sizes do not represent the current stock. However, our estimates under equilibrium assumption are very close to observation-based estimates of NPP and C stocks (Table R1) and are all within the range of other estimates. In fact, our estimates of biomass and SOM stocks are a little bit larger than the state-of-the-art estimates listed in Table R1. In addition, the C stocks are not necessarily larger than the dynamic state. For example, Jones et al. (2009) showed that when the climate forcing was held constant after 2050, the equilibrated forest cover in a region of Amazon forest will be smaller than the dynamic state in 2050, resulting less equilibrated biomass than the dynamic biomass in this region. Given these evidences, we do not think that our results about the steady state are significantly underestimated.

| Table R1  | Computed NPP | and C stocks | under | equilibrium | assumption | compared | with |
|-----------|--------------|--------------|-------|-------------|------------|----------|------|
| published | estimates.   |              |       |             |            |          |      |

| udy Other                              |
|----------------------------------------|
| 54 (Zhao et al., 2005)                 |
| 450 [380-536] (Erb et al., 2018)       |
| 450 [375-540] (Bar-on et al., 2018)    |
| $1408 \pm 154^1$ (Batjes et al., 2016) |
|                                        |

1 This value represent the C stock between the depth 0-100 cm, generally correspond to the biologically active depth

To address all the points raised by the referee, in the revised manuscript, we revised the manuscript:

 Page 2 line 37 – page 3 line 6: "We present a new global data-driven diagnostic of C, N and P pools and fluxes, called GOLUM-CNP (Global Observation-based Landecosystems Utilization Model of Carbon, Nitrogen and Phosphorus) which is based on

the assumption that these cycles are equilibrated with present day conditions (see below for limitations of this approach). The goals of this study are to: 1) establish a global datadriven diagnostics of C, N and P fluxes and pools in order to compare nutrient use efficiencies, nutrient turnover rates and other relevant indicators across biomes; and 2) provide a new dataset that can be used to evaluate the results of global terrestrial biosphere models with consistent state of C, N and P cycles. In GOLUM-CNP, the C, N and P cycles are estimated for different biomes assuming steady state with present-day input of carbon (NPP), nitrogen (N deposition and N fixation) and phosphorus (P deposition and release from rock weathering) (see Sect. 3.2). The reason for this steadystate computation lies in the fact that only few global long-term observations associated with N and P cycles are available and are insufficient to constrain a transient simulation under the model framework. For example, field-scale manipulation experiments have shown that warming, elevated atmospheric CO2, and N and P fertilization can drive changes in stoichiometries and nutrient resorption (Sistla and Schimel, 2012; Mayor et al., 2014; Yang et al., 2014; Yuan and Chen, 2015) in terrestrial ecosystems, but these data are insufficient to infer these changes in terrestrial ecosystems during the past decades. As more data becomes available the model framework can be adjusted to simulate a transient present day state. Although, the steady-state assumption hampers the comparison of stocks with present day observations, a direct comparison with simulated steady states of DGVM is possible as these model can simulate the steady-state for present day conditions.

Starting from a CARbon DAta MOdel fraMework (CARDAMOM) ...we incorporated observed stoichiometric ratios (C:N:P) in each pool, N and P external input fluxes, transformations and losses in ecosystems and losses and observation basedinformation for the fraction of gaseous losses of N to total (gaseous and leaching) losses of N from a global dataset of 15N measurements in soils. Although the diagnostics is presented for steady state, the methods used to compute fluxes and pools are generic and could be extended to non-steady state (see Sect. 2 and equations in Appendix A-C) when more data will become available in the future (see Sect. 5.3)."

Sect. 5.3 page 12 lines 35-37: "... Our results are a first step for evaluating global biogeochemical cycles. Although our steady-state C pool sizes (given the NPP and residence time at the condition of current climate) were within the [25, 75th] percentile range of the original non-steady-state CARDAMOM results (Fig. S1) at most grid cells, the biomass C stocks at 5%-10% of forest grid cells exceed the uncertainty range of CARDAMOM. In addition, independent remote-sensing estimates for 30 % to 80 % were 4.76 ± 1.78 kg C m-2 for mean forest C density and 79.8 ± 29.9 Pg C for total forest C (Thurner et al., 2014), which were lower than the GOLUM-CNP estimates (6.51 kg C m-2 for mean forest C density across pixels defined as forest in Fig. 2, and 181 Pg C for total forest C) for this region. This inconsistency was largely due to the fact that northern temperate and boreal forests may deviate substantially from their equilibrium for the current NPP (Pan et al., 2011), because of climate change and elevated CO2. Residual overestimation could be also due to the fact that biomass removal by harvesting and from disturbance other than fires was not explicitly constrained in CARDAMOM and thus not represented in GOLUM-CNP. A transient simulation of N and P cycling

will be needed in future studies..."

- We also specify the computation method for the output variables, to indicate whether these variables depend or not on the equilibrium assumption (Table 1 in the revised manuscript).

2. Reading between the lines, it seems that the N fixation is about 120Tg/year. How does this square with other estimates, e.g. Vitousek et al. 2013 (44Tg/year)? Since it's not discussed where this N fixation number came from (and the reference isn't available), or where the N deposition number came from, it makes it difficult to give much credibility to the openness discussions which rely on these.

**Response:**

The estimates of global BNF from synthesis and extrapolation are highly uncertain. Cleveland et al. (1999) estimated of biome- and global-level BNF based on scaling up of *in situ* measurements of BNF. They applied average plot-level BNF rates measured within each biome to the biome as a whole, assuming empirical range of values for the cover of plants with potential N-fixing symbioses. They estimated that global BNF by natural ecosystems was ~195 Tg N yr-1 (with a range 100-290 Tg N yr-1) during pre-industrial. The estimates by Cleveland et al. (1999) were criticized by Galloway et al. (2004) and later Sullivan et al. (2014) as too high, in particular for tropical regions. Galloway et al. (2004) suggested a lower global BNF to be 128 Tg N yr-1; Wang et al. (2007) made the estimate of BNF using the principle of resource optimization, and their global estimate for the year 1900 was 125 Tg N yr-1 (Wang and Houlton, 2009). Vitousek et al. (2013) incorporated information on N fluxes with 15N relative abundance data and estimated pre-industrial N fixation was 44 TgN yr-1. The large range (44-290 TgN yr-1) in the estimates of BNF mentioned above reflects both a paucity of measurements of N fixation, as well as an incomplete understanding of the biophysical and biochemical controls on BNF.

In this study, we used estimates of biological N fixation from the CABLE model simulation by Peng et al. with a N fixation model developed by Wang et al. (2007), and used by Wang and Houlton (2009). This work by Peng et al. still is under review for Global Biogeochemical Cycles. We agree that there are considerable uncertainties about the magnitude of BNF. Vitousek (2013) estimated the total global N fixation from ranges from 40 to 100 Tg N yr-1 for the preindustrial time. Our estimated rate of 116 Tg N/yr-1 for the present days is still close to the upper limit of Vitouske (2013). Much of the difference may result from much higher rates used in this study for tropical forests than the recent estimate by Sullivan et al. (2014) that was based on the measurements at sites in southwest Costa Rica. However, estimates of BNF from Vitousek et al. (2013) are only available at global scale. BNF from CABLE, a process-based model validated for observations for diverse terrestrial biomes (Wang et al., 2007; Houlton et al., 2008) is to our best knowledge the only available global spatially explicit estimates of BNF besides the heavily criticized estimates by Cleveland et al. (1999), and is therefore used in this study. We discussed the uncertainties of BNF in the revised manuscript on page 5 line 7: "We used the spatially explicit estimates of N deposition (Wang et al., 2017) for 2001-2010, which were evaluated with globally distributed in-situ measurements. The spatially explicit N fixation was taken from the CABLE model simulation for 2001-2010 (Peng et al., submitted) with a N fixation model developed by

Wang et al. (2007). The simulation result matches the relative abundance of N2-fixing legumes in different ecosystems. Globally, the N fixation was 116 Tg N yr-1, within the range of empirical data (100-290 Tg N yr-1; Cleveland et al., 1999; Galloway et al., 2004), but was larger than the estimate of 44 Tg N yr-1 by Vitousek et al. (2013) for pre-industrial. The large range (44-290 TgN yr-1) in the estimates of nitrogen fixation reflects both a paucity of measurements of N fixation, as well as incomplete understanding of the biophysical and biochemical controls on N fixation. However, as mentioned in Sect. 2.2, our estimates of total N inputs (N deposition + N fixation) are consistent with the estimate from Houlton et al. (2018). And to our knowledge, CABLE simulation is the only product that has spatially explicit and processed-based estimates of N fixation, and is therefore used in this study. The resorption coefficients of leaves..."

3. The relationship between GOLUM-CNP and CARDOMON is opaque and needs to be clarified. It is particularly unclear with regard to what the relationship between the code provided and CARDOMON is. For example, does this code work independently? Or does it need CARDOMON to run? If CARDOMON is part of the code provided, which parts are new and which are CARDOMON?

**Response:**

The code of GOLUM-CNP is not just an extension of CARDAMOM, but a new model that used the outputs of CARDAMOM, i.e. data-driven C variables. The relationship between GOLUM-CNP and CARDAMOM are: 1) the C pools and fluxes in GOLUM-CNP follows that of CARDAMOM after steady state transformation, except that we group foliar and vegetation labile C into a single pool because labile C is not a measurable pool and no observation data were available to separate it from foliar in terms of stoichiometry. The N and P cycles in GOLUM-CNP were added on top of this C cycle model by adding two pools (soil inorganic N and P) and associated N and P fluxes; 2) GOLUM-CNP used some of the CARDAMOM results as inputs which are listed and explained in Table 1. These variables are the key parameters that describe the C cycle and influence the estimates of status of nutrient cycles. For instance, the residence time of C pools were used in the calculation of N and P fluxes (Eqs. E15, E16, E26 and E27).

To precise the relationship between GOLUM-CNP and CARDAMOM in the manuscript, we made the following revisions:

- We added on page 3 line 8: "The GOLUM-CNP-framework describes the C, N and P cycles in natural (i.e. non-agricultural) terrestrial ecosystems (Fig. 1). We used the same C pools and fluxes as in The C cycle follows the model structure of the CARDAMOM diagnostic (see Sect. 2.1 for details) to describe the C cycle and we computed associated N and P pools and fluxes. Biomass is divided into three pools: foliage, fine roots and wood..."
- We revised the sentence on page 4 line 35: "...the N and P contents in soil control the decomposition of soil C and thus the soil C pool observed (Manzoni et al., 2010). In this sense, it is appropriate to use C cycle from CARDAMOM as inputs to estimate the pool and fluxes of N and P."

The code must be run with variables from CARDAMOM (and also other required datasets, see "Inputs" in Table 1). Following the referee's suggestion, we comment the code

**thoroughly. Please refer to the code resubmitted for detail.**

4. It's not explained what the intended use of this model is. It's essential early on in the paper to have some examples of use, as well as specific limits on what it shouldn't be used for, (particularly given the limitation of it being an equilibrium model). This is slightly covered right at the end of the paper, but needs to be earlier and more extensive.

**Response:**

Thank you for this suggestion to improve the manuscript. We agree with the referee this information is quite important for the readers. In the revised manuscript, we explained the objects of this study in the introduction and discussed the equilibrium assumption in Sect. 5.3 (see answer to Major comment 1).

We remind the readers that current GOLUM-CNP results are steady-state estimates, when comparing our results with other DGVMs, it is better to run the DGVMs to steady state with present-day inputs of C, N and P. We revised the manuscript by:

- Adding a paragraph in Sect. 5.3 about this point: "The model structure of GOLUM-CNP is mainly described by the inputs (NPP for C cycle, N deposition and fixation for N cycle, P deposition and release from rock weathering for P cycle) and residence times. Most DGVMs (e.g. Goll et al., 2012, 2017a; Medvigy et al., 2009; Parton et al., 2010; Thornton et al., 2007; Wang et al., 2010; Weng and Luo, 2008; Xu-Ri and Prentice, 2008; Yang et al., 2009; Zaehle et al., 2014; Zaehle and Friend, 2010) can be summarized by these two components, although these models have more processes and use complex equations to describe the dynamics controlling carbon and nutrient distribution among pools and the turnover of each pool. In this context, the output of the GOLUM-CNP provides a traceable tool that can be used in the future to compare the results between GOLUM-CNP and different DGVMs. As DGVMs are capable of computing the steady state of the biogeochemical cycles for present conditions, a direct comparison between GOLUM-CNP estimate and DGVMs' estimates is possible."
- Revising page 12 line 38 to page 13 line 3: "At last, Thethe sensitivity matrix presented in Sect. 4.4 provides a useful tool for assessing the uncertainties in model outputs by propagating the uncertainties in the model inputs. We applied this method to quantitatively assess the sources of uncertainties in the estimated nutrient-use efficiencies (Sect. 5.1 and Fig. 8), but we also found that the uncertainties for some other quantities were currently difficult to obtain, because the estimates of uncertainties were not available for all spatially explicit input data. This sensitivity analysis can be used in future studies to quantify the contribution of each input data set to the uncertainty in other model outputs, to characterize the dominant sources of uncertainties in the estimated C, N and P processes, to identify the major differences between different models (e.g. GOLUM-CNP versus DGVMs) and thus to identify priorities for future data syntheses to fill the largest gaps in uncertainty. Future studies that provide global data sets will need to include systematic evaluations and spatially explicit estimates of uncertainties in their data sets."
- Revising the conclusion on page 13 lines 10-17: "... The structure of GOLUM-CNP is analogous to most other process-based LSMs DGVMs describing carbon and nutrient interactions (e.g. Goll et al., 2012, 2017a; Medvigy et al., 2009; Parton et al., 2010;

Thornton et al., 2007; Wang et al., 2010; Weng and Luo, 2008; Xu-Ri and Prentice, 2008; Yang et al., 2009; Zaehle et al., 2014; Zaehle and Friend, 2010), although thesemodels have more processes and use complex equations to describe the dynamicscontrolling carbon and nutrient distribution among pools and the turnover of each pool. The output of the GOLUM-CNP provides a traceable tool and, in which a consistencybetween different datasets of global C, N and P cycles has been achieved. Such aframework can thus-be used in the future to test the performance of these complex LSMs DGVMs in the simulation of interactions between C, N and P cycling."

5. The code is very dense, making it very difficult to read. Code should have comments every 1 - 10 lines, depending on how interpreted/dense, etc. the code is. Ideally, code should be commented so that, if you stripped out the actual code or if you didn't know python at all, you could re-write it in another language just from the comments. I was also a bit surprised not to see any functions used.

**Response:**

We write detailed comments in the new code. In fact, we have the main code "Globe\_ss\_Anal.py" and another two files incorporating three modules: 1) ncload.py, in which the module "ncload" reads the netcdf4 files and associated variables very efficiently; 2) ss\_Anal.py, which make the computation at each grid cell. In ss\_Anal.py, two functions ("tree" and "grass") making the computation for forest and grassland are defined separately. In the main code Globe\_ss\_Anal.py, "ncload" are called at the beginning of the code to read all the variables; "tree" and "grass" are called according to the dominant biomes of each grid cell.

**Minor points**

1. P.2 L6-7. Yes, but also water, light etc. are essential controls - if there's no light and water it doesn't matter how much N or P there is, nothing will grow.

**Response:**

Thank you for noting this inappropriate expression. We revised the sentence as: "...in terrestrial ecosystems. N and P availability affects vegetation productivity, growth and other processes (Norby et al., 2010; Sutton et al., 2008; Vitousek and Howarth, 1991)."

2. P.2 L17 - 30. This seems to mix up P and N fertiliser. It would be better to keep the two issues separate wherever possible.

**Response:**

Thank you for the suggestion. We rephrase this paragraph: "... Many of the underlying processes are not fully understood, and comprehensive data for evaluation are lacking to constrain the representation of some key processes (Zaehle et al., 2014), so model structure, the processes included and the prescribed parameters differ widely among DGVMs (Zaehle and Dalmonech, 2011). For example, some models assume constant stoichiometry (N:C and P:C ratios) in plant tissues (Thornton et al., 2007; Weng and Luo, 2008), but others have a flexible stoichiometry (Wang et al., 2010; Xu-Ri and Prentice, 2008; Yang et al., 2009; Zaehle and Friend, 2010). For the N cycle, for instance, some models do not include losses of gaseous N due to-from denitrification (Medvigy et al., 2009), some use the "hole-in-the-pipe" approach to simulate the denitrification flux (Thornton et al., 2007; Wang et al., 2010),

assuming it is proportional to net N mineralization, and others calculate this flux as a function of soil N-pool size and soil conditions (temperature, moisture, pH, etc.) (Parton et al., 2010; Xu-Ri and Prentice, 2008; Zaehle and Friend, 2010). For models of the terrestrial P cycle, for instance, Jahnke (2000) estimated a global amount of soil P of 200 Pg and that P contained in plants was 3 Pg, based on empirical P content of soils (0.1%) and soil thickness (60 cm). These estimates were questioned by Wang et al. (2010) and Goll et al. (2012), who estimated that P in plants ranged between 0.23 and 0.39 Pg and that P in soil was only 26.5 Pg based on P:C ratios derived from more comprehensive stoichiometric data sets. Furthermore, thesemodels-terrestrial ecosystem models are usually only evaluated for specific ecosystems or at a limited number of sites (Goll et al., 2017a; Yang et al., 2014). The application of these models for simulations with global coverage is thus highly uncertain (Goll et al., 2012; Wang et al., 2010; Zhang et al., 2011)."

**3. P18. L28. Reference to Peng missing.**

**Response:**

Please see the response to the 2nd major point. And we added the reference accordingly.

**4. P21. Table 2. When is this table referring to? (Pre-industrial? Present day?) It needs to be specified.**

**Response:**

We revised the caption of Table 2 into: "Global annual mean C-pool sizes, NPP and heterotrophic-respiration fluxes in the C-cycle model assuming steady states under the climate conditions of 2001-2010." We also revised the text in the manuscript on page 8 line 3: "Table 2 shows the global C-pool sizes and main fluxes of the steady-state C cycle transformed from CARDAMOM under the climate conditions of 2001-2010, which are compared with the means and percentile ranges from the original non-steady-state CARDAMOM results during 2001-2010. The differences between the steady-state transformed pool sizes…"

**5. P22. Figure 1. The caption would be more useful if the terms at the top of the figure were defined first, and then worked downwards.**

**Response:**

Thank you for the suggestion. We reorganized the sentences in the caption: "**Figure 1** Schematic representation of the pools and fluxes in the C, N and P cycles within GOLUM-CNP. The gray, blue and red arrows represent C, N and P fluxes, respectively. Plants are divided into foliar, fine root and wood pools, where the wood pool includes woody stems and coarse roots. Litter and soil are two separate pools. The inorganic pool represents the nutrient sources in the soil that are available for plant uptake. Arrows between the pools represent the directions of C, N and P flow between pools. External inputs of N are atmospheric deposition (Nd) and biological N fixation (Nfix). External inputs of P are atmospheric deposition (Pd) and P released by rock weathering (Pw). FC is net primary production (NPP). FN and FP are plant uptake of N and P from the inorganic N and labile soil P pools, respectively. Rh is release of C due to heterotrophic respiration. Mineralization of N and P is modeled along with litter and SOM decomposition, and N and P immobilization is modeled by a flux from the inorganic pool to SOM. External losses of N occur by fire, leaching and denitrification. External losses

**of P occur by fire, leaching and transfer to occluded P in the soil."**

**6. P24. Figure 3. This is a very difficult to read. A table or a series of bar plots would be much better.**

**Response:**

We tried to use tables or bar plots (Fig. R1) as the referee suggested. There are many variables involved, and we want to separate the seven biomes to see their major differences in the C, N and P cycle. As shown in Fig. R1, such figures will need 6 y-axis, and a very long space to put 6 figures. Even though we tried to optimize the spacing between figures, it is hard to put it on one A4 page and the texts are small. In addition, with the original plots, we depict the full C, N and P cycles including the pool sizes, residence times of different pools and fluxes. But with bar plots, only a small subset of the variables could be shown in limited space so that the readers will lose the whole pictures of the C, N and P cycles when reading these bar plots only. It is the same case to use a series of tables, because readers would need to link the variables from the tables in their own mind. At last, we decide to use the same figure, but rearrange them. We put the figures in a 3 by 3 grid so that all figures and numbers in the figures are larger and more easily to be read (please see the revised manuscript). And for readers who are more interested in reading bar plots, we also provide these bar plots in the supplementary material.

---

## Author Comment (AC2) · 25 Jul 2018

**Response to comments on "GOLUM-CNP v1.0: a data-driven modeling of carbon, nitrogen and phosphorus cycles in major terrestrial biomes" by Y. Wang et al.**

We thank the referee for reviewing our manuscript. Please find attached a point-by point reply to each of the comments raised by the referee with legible text and figures organized along the text.

This manuscript presented a simple model for coupled terrestrial carbon-nitrogen-phosphorus cycles. A model-data fusion approach was applied to estimate steady-state values for selected model state variables, fluxes, and parameters. In particular, the estimated quantities allowed for computation of carbon, nitrogen, and phosphorus pool sizes, the openness of the N and P cycles, characteristic turnover times, and nutrient use efficiencies.
In principle, I think that estimation of these quantities using model-data fusion is a worthwhile and interesting idea. However, I think there is substantial room for improvement in both the presentation and the approaches used. Here are a few general comments:

**Response:**

We would like to thank the referee for the valuable comments and suggestions for improving our manuscript. Following the reviewer's comments, we carefully revised our manuscript. Please find below the point-to-point responses (in black) to all referee comments (in blue). For your convenience, changes in the revised manuscript are highlighted with dark red. All the pages and line numbers correspond to the original version of text.

1. The paper contains no statement of objectives, questions, or hypotheses. Including any or all of these features would help to orient the reader. But without them, the paper is difficult to evaluate because the goals of the paper are unstated.

**Response:**

Thank you for the suggestion. We revise the paragraph in the introduction (page 2 line 37 – page 3 line 6):

"We present a new global data-driven diagnostic of C, N and P pools and fluxes, called GOLUM-CNP (Global Observation-based Land-ecosystems Utilization Model of Carbon, Nitrogen and Phosphorus) which is based on the assumption that these cycles are equilibrated with present day conditions (see below for limitations of this approach). The goals of this study are to: 1) establish a global data-driven diagnostics of C, N and P fluxes and pools in order to compare nutrient use efficiencies, nutrient turnover rates and other relevant indicators across biomes; and 2) provide a new dataset that can be used to evaluate the results of global terrestrial biosphere models with consistent state of C, N and P cycles. In GOLUM-CNP, the C, N and P cycles are estimated for different biomes assuming steady state with present-day input of carbon (NPP), nitrogen (N deposition and N fixation) and phosphorus (P deposition and release from rock weathering) (see Sect. 3.2). The reason for this steady-state computation lies in the fact that only few global long-term observations associated with N and

P cycles are available and are insufficient to constrain a transient simulation under the model framework. For example, field-scale manipulation experiments have shown that warming, elevated atmospheric $CO_2$, and N and P fertilization can drive changes in stoichiometries and nutrient resorption (Sistla and Schimel, 2012; Mayor et al., 2014; Yang et al., 2014b; Yuan and Chen, 2015) in terrestrial ecosystems, but these data are insufficient to infer these changes in terrestrial ecosystems during the past decades. As more data becomes available the model framework can be adjusted to simulate a transient present day state. Although, the steady-state assumption hampers the comparison of stocks with present day observations, a direct comparison with simulated steady states of DGVM is possible as these model can simulate the steady-state for present day conditions.

Starting from a CARbon DAta MOdel fraMework (CARDAMOM) …we incorporated observed stoichiometric ratios (C:N:P) in each pool, N and P external input fluxes, transformations and losses in ecosystems and  the fraction of gaseous losses of N to total (gaseous and leaching) losses of N from a global dataset of $^{15}$N measurements in soils. Although the diagnostics is presented for steady state, the methods used to compute fluxes and pools are generic and could be extended to non-steady state (see Sect. 2 and equations in Appendix A-C) when more data will become available in the future (see Sect. 5.3).

We first present the model structure (Sect. 2) and the data sets used to derive its outputs consisting of pools…"

2. The model is meant to be applied to natural ecosystems, and so the model-data fusion approach should be valid in grid cells that are dominated by nature ecosystems. However, the problem is that very few such grid cells remain (Ellis and Ramankutty 2008). I would therefore expect biases to emerge when the natural ecosystem model is applied to grid cells that are biogeochemically influenced by humans. I am bothered that anthropogenic effects are not accounted for in the global and (natural) biome scale averages computed by the authors.

**Response:**

We agree with the reviewer that current terrestrial ecosystem are largely influenced by anthropogenic activities. Here, however, "natural ecosystem" refers to non-agricultural ecosystems. The inputs of C, N and P to these natural system in our framework account for anthropogenic influences. For example, the estimates of NPP from CARDAMOM represent NPP under current level of atmospheric $CO_2$ concentration and climate change. The estimate of atmospheric N and P deposition (Wang et al., 2017) includes anthropogenic emissions of reactive N and of P from fuel combustion. Our definition of "natural ecosystems" follows the terminology used in previous studies, such as Cleveland et al. (2013).

In the revised manuscript, we specify this point on page 3 line 14: "The GOLUM-CNP framework describes the C, N and P cycles in natural (i.e. non-agricultural) terrestrial ecosystems (Fig. 1). We follow the model structure …" We change "natural vegetation" to "non-agricultural vegetation" and "natural biomes" to "non-agricultural biomes" all through the manuscript.

3. In a way, not enough information is presented. I would be interested in seeing spatial maps of at least some quantities and some confirmation that the model-data fusion approach is

**Response:**

Thanks for the suggestion. We add the global gridded map of N deposition (input data, Fig. S3a), N fixation (input data, Fig. S3b), P deposition (input data, Fig. S4a), release of P from weathering (input data, Fig. S4b), openness of N (output of GOLUM-CNP, Fig. S5a), openness of P (output of GOLUM-CNP, Fig. S5b), $\tau_{N,eco}$ (output of GOLUM-CNP, Fig. S6a), $\tau_{P,eco}$ (output of GOLUM-CNP, Fig. S6b), NUE (output of GOLUM-CNP, Fig. S7a) and PUE (output of GOLUM-CNP, Fig. S7b) in the supplementary materials and remind the readers of these spatial maps in the main text when necessary.

4. In another way, too much information is presented, making the paper confusing (see Figs. 3, 7). It would help to have a better-defined narrative arc. While many numbers were presented, I think the authors could do a better job in drawing attention to new, qualitative insights.
**Response:**

We think Fig. 3 depict the full C, N and P cycles including the pool sizes, residence times of different pools and fluxes. We prefer these plots rather than some bar plots which may be easier to read but only show specific aspects of the C, N and P cycling. Following thre reviewer's comments, we revise Sect. 4.2:

- In Sect. 4.2, we first compare the C, N and P cycles in different biomes. Such comparisons include the plant uptake, stocks in vegetation, litter and SOM, and the loss fluxes from the ecosystems.

- We then compare our estimates of global N and P stocks with previous studies, showing that our estimates are close to the estimates from previous studies.

- At last, we compare the global estimates of total nutrient inputs and losses with previous studies. We highlight that the loss of N by fire accounted for 26% of total N loss, while less fraction of P is lost through fire.

Fig. 7 shows the sensitivity of the selected output variables to all input variables. Sensitivities (dy/dx), not like the attribution of which input variable contribute the most to the overall uncertainties in the output (for example, Fig. 8), have no priority for different variables but they show how output change as a result of change in the input. The colors in these figures give qualitative information about the sensitivities' magnitude. The numbers in each grid cell give the quantitative information with precise values for readers interested in the exact numbers. As suggested by the reviewer, we improve the texts in Sect. 4.4.

5. I would like to know approximately what level of bias the steady-state assumption incurs. These ecosystems are unlikely to be at steady state for many reasons (CO2 fertilization, changes in nutrient deposition, climate extremes such as droughts, etc.).
**Response:**

The reviewer mentions several processes that related to both short-term (climate extremes) and long-term (e.g. $CO_2$ fertilization, changes in nutrient deposition) disequilibrium. Firstly, short-term disturbance causes temporal changes in the C sink and source at yearly to decadal scales, but has no impact on long-term C fluxes and pools unless the disturbance change its regime (e.g. become more frequent) (Luo and Weng, 2011).

Regarding the long-term effect, some previous studies have investigated the changes in C, N stocks between pre-industrial times and present day. Thornton et al. (2007) investigated the C stock changes from 1850 to present day (1976-2000) using CLM-CN model. They found that the C stocks in vegetation, litter and SOM increased by 35, 1 and 10 Pg, which equalled to only 5%, 6% and 3% of the respective initial C stocks in 1850. Zaehle et al. (2010) estimated the C and N stock changes in vegetation and SOM, using the O-CN terrestrial biosphere model. By excluding the impact of land use change, vegetation C stocks increased by 62 Pg and soil C stocks increased by 39 Pg from 1860 to 2002, being 13% and 3% of initial C stocks for vegetation and SOM respectively. The N stocks increased by 376 Tg and 2836 Tg in vegetation and SOM, being 11% and 3% of initial N stocks. Zaehle et al. (2013) further accounted for the effect of land use change on the global C and N cycles. The results also showed that the cumulative effect of anthropogenic disturbance from preindustrial times are small (<5%) on the natural C and N cycles. These evidences indicate that anthropogenic disturbance on the natural C, N and P cycles are not so significant at global scale. However, at local scales, some places are experiencing significant anthropogenic disturbances, mostly through deposition and fertilizer (Jornard et al., 2015). However, the extent to which natural ecosystems are affected by these disturbances are not clear (Wang et al., 2017). To check the long-term effect, we compare the recomputed pool sizes for C with the original CARDAMOM estimates (Fig. S1 in the revised supporting information). The steady-state transformed pool sizes are within the [25, 75$^{th}$] percentile range of the original CARDAMOM results at more than 90% forest grid cells. The major part of the deviation comes from the treatment of grassland in GOLUM and are not associated to the steady-state assumption: In GOLUM-CNP grasslands are considered as a distinct biome, while the original CARDAMOM did not use land cover information and provided some woody biomass pools for grassland dominated regions. As GOLUM does not consider woody biomass for grass, we transfer wood growth from CARDAMOM into non-woody tissue for grasslands. This makes our C pools more different from the original CARDAMOM at grassland-dominated pixels than at forest-dominated pixels. However, although the biomass at grassland grid cells in GOLUM-CNP are much lower than the biomass in CARDAMOM, the litter and SOM pool sizes at grassland grid cells in GOLUM-CNP are still within the [5, 95$^{th}$] percentile range of the original CARDAMOM results (Fig. S1b and S1c in the revised supporting information).

Here, we also remind that we develop a steady-state model as a first attempt of a CNP data-driven diagnostic model. The objective of this study is not to reproduce the present day state, but to provide a steady-state estimates of C, N and P cycle with current input of C, N and P, stoichiometry of N:C and P:C ratios, and residence times of different pools, because there is a lack of data to constraint a transient state (see point 1). As a result, we choose to constrain an equilibrium state. Although it does not correspond in some aspects to the present day state, it is still useful for evaluating global models as these model are able to simulate a equilibrium state for present day conditions and a direct comparison is possible (see point 2 and point 3).

1) Although we described processes in the C, N and P cycles by a set of differential equations (Eqs. A1-A5, B1-B7, C1-C7), only few global long-term observations associated with N and P were available to constrain a transient simulation. For example, the synthesis of stoichiometries in different pools are based on published literature during

the last four decades, and almost no data are available for preindustrial period; field-scale manipulation experiments have shown that warming, elevated atmospheric $CO_2$, and N and P fertilization can drive changes in stoichiometries and nutrient resorption in terrestrial ecosystems (Sistla and Schimel, 2012; Sardans et al., 2012; Sardans and Peñuelas, 2012; Li et al., 2013; Mayor et al., 2014; Yang et al., 2014; Yuan and Chen, 2015; Sardans et al., 2016; Sardans et al., 2017), but these data are insufficient to infer these changes in terrestrial ecosystems during the past years. As a result, for a data-driven model framework, we computed the equilibrium estimates and this equilibrium corresponds to the state where the inputs of C, N and P (e.g. NPP for C cycle, N deposition and fixation for N cycle, P deposition and release of P by rock weathering for P cycle) and residence times equal to the estimates under present-day conditions.

2) Given the state-of-the-art estimates about the magnitude of anthropogenic disturbance on natural C, N and P cycles and large uncertainties as mentioned above, it is still useful to take steady-state estimates as a diagnostic to evaluate DGVMs in simulating global C, N and P cycling (Wang et al., 2010; Xu-ri et al., 2008; Yang et al., 2009; Lawrence et al., 2011). The estimates based on steady state assumptions listed above match well with a wide range of *in-situ* observations in recent years.

3) CARDAMOM is the only spatially explicit data-driven product of C cycle, which does not rely on a steady-state assumption. In this study, our aim is to develop a steady-state model (see point 1) in which a consistency between different datasets are achieved (which is a novelty). We chose to use the variables in the CARDAMOM products: NPP, residence times and fire fractions, which were constrained by global long-term observations of MODIS LAI and MODIS burned area. For other C variables whose direct constraints do not have a global coverage in CARDAMOM (such as biomass), we recalculated them based on the equilibrium assumption. With this steady state C cycle, we compute the associated N and P pools and fluxes. For example, the N and P in SOM pool equal to the product of N:C/P:C ratios and equilibrated C content.

   In summary, given the three concerns mentioned above, we computed an equilibrium estimate of C, N and P cycle, where the inputs of C, N and P into the terrestrial ecosystems (NPP, N and P deposition, N fixation and release of P by rock weathering), stoichiometries of N:C and P:C ratios, the residence times of different pools and the land cover are held constant as the period 2001-2010. Although this steady-state estimates deviate from the actual state of present-day disturbed cycles, we compare the GOLUM-CNP with the original CARDAMOM to show that the steady-state estimates in GOLUM-CNP are still within the range of such large uncertainties from data on the present day state.

6. The model is simple, which is okay for a first attempt. But there does not seem to be any substantial discussion of how to make the model more realistic by including other known fluxes and feedbacks (especially appropriate in 5.3, Future research and data needs).
**Response:**
   Thanks for the suggestion. We substantially extended the discussion about future development for GOLUM-CNP towards a more realistic / complex diagnostic.

7. The manuscript needs a careful proofreading by a fluent English speaker. Starting with the title, "model" would make more sense than "modeling".

**Response:**

Thank you for your suggestion. We have requested several fluent English speakers to play a more active role in the re-writing and editing of the whole manuscript. We hope that the revised manuscript could satisfactorily address all of your concerns.

Specific comments:
8. Page 2, lines 17-21: Not clear if you are talking about N cycling, P cycling, or both

**Response:**

We revise this paragraph: "… Many of the underlying processes are not fully understood, and comprehensive data for evaluation are lacking to constrain the representation of some key processes (Zaehle et al., 2014), so model structure, the processes included and the prescribed parameters differ widely among LSMs (Zaehle and Dalmonech, 2011). For example, some models assume constant stoichiometry (N:C and P:C ratios) in plant tissues (Thornton et al., 2007; Weng and Luo, 2008), but others have a flexible stoichiometry (Wang et al., 2010; Xu-Ri and Prentice, 2008; Yang et al., 2009; Zaehle and Friend, 2010). For the N cycle, for instance, some models do not include losses of gaseous N  from denitrification (Medvigy et al., 2009), some use the "hole-in-the-pipe" approach to simulate the denitrification flux (Thornton et al., 2007; Wang et al., 2010), assuming it is proportional to net N mineralization, and others calculate this flux as a function of soil N-pool size and soil conditions (temperature, moisture, pH, etc.) (Parton et al., 2010; Xu-Ri and Prentice, 2008; Zaehle and Friend, 2010). For models of the terrestrial P cycle, for instance, Jahnke (2000) estimated a global amount of soil P of 200 Pg and that P contained in plants was 3 Pg, based on empirical P content of soils (0.1%) and soil thickness (60 cm). These estimates were questioned by Wang et al. (2010) and Goll et al. (2012), who estimated that P in plants ranged between 0.23 and 0.39 Pg and that P in soil was only 26.5 Pg based on P:C ratios derived from more comprehensive stoichiometric data sets. Furthermore,  terrestrial ecosystem models are usually only evaluated for specific ecosystems or at a limited number of sites (Goll et al., 2017a; Yang et al., 2014). The application of these models for simulations with global coverage is thus highly uncertain (Goll et al., 2012; Wang et al., 2010; Zhang et al., 2011)."

9. Page 2, lines 28-30: Too bad that this is not discussed further

**Response:**

We add some discussions about the different quantities of P pool sizes. Please refer to the response to Comment 8.

10. Page 3, lines 17-19, state that an inorganic P pool was added to the model, and it is assumed that this P is accessible to plants. However, in reality, very little inorganic P is accessible to plants.

**Response:**

P exists in various forms in soil with different accessibilities to plants (Yang and Post, 2011). P may enter the solution through desorption or dissolution of inorganic P, or by

mineralization of organic P, and then is utilized by plants.

In the original manuscript, we referred only to "inorganic P that are accessible by plants" for an integration of all the forms that can be easily used by plants. In the model developed by Wang et al. (2007) they defined four P pools (labile P, sorbed P, strongly sorbed P and occluded P) in soil and argued that the time for reaching equilibrium between labile inorganic P and sorbed inorganic P in soil is less than one hour. In this context, although the inorganic P that is directly accessible to plants is very small, there are still considerable amount of other forms of P that can be fast transformed to inorganic P that can be utilized by plants. A recent synthesis of a database of soil P fractions by Yang and Post (2011) also confirmed that the amount of labile P in soils (defined as the sum of resin Pi, bicarbonate Pi and Po using Hedley fractionation method) is generally much higher than vegetation demand, even in highly weathered soils commonly considered as P limited. They argued that P uptake by microbial (immobilization) competing with plant uptake strongly controls the P availability of plants, citing evidence by Cole et al. (1977) "…two grassland sites…, root uptake of P vs. microbial uptake of P was 5.26 kg P ha$^{-1}$ yr$^{-1}$ vs. 24.57 kg P ha$^{-1}$ yr$^{-1}$ and 12.00 kg P ha$^{-1}$ yr$^{-1}$ vs. 31.60 kg P ha$^{-1}$ yr$^{-1}$ respectively". These evidences all show that although some P in soils is not readily accessible to plants, a large amount of this P can be transformed quickly into a form that can be used by plants, and that this transformation is usually faster than the time scale (one year) of our computation. In consequence, it is possible to define a single pool to represent labile soil P available for plants. And our results (original Fig. 3) about the immobilization fluxes and plant uptake in most biomes are supported by Yang and Post (2011) indeed.

To address the comments by the referee, we use "labile soil P" instead of "inorganic P" all through the manuscript and revised the text: "…Two additional pools not present in CARDAMOM are added, representing soil inorganic N and labile soil P. These two N and P pools are assumed to represent nutrients accessible by plants (see Sect. 2.1 and 2.2). Of note is that these inorganic N and labile P pools represent an integration of various forms of N and P. For example, P has various forms in soil and can be transformed between those forms (Wang et al., 2007; Yang and Post, 2011). Some forms of organic P (e.g. bicarbonate Po in Hedley method) can also be easily mineralized and thus were implicitly included in our labile soil P pool. Other forms of P that are not easily accessible to plants are referred to as "occluded P" and labile soil P can become occluded P (Wang et al., 2010; Goll et al., 2017a). Fluxes connecting the pools are described by the differential equations …"

11. Page 3, line 31: What do you think about N from rock weathering (Houlton et al. 2018)?

**Response:**

Thank you for pointing out this newly published paper. The paper of Houlton et al. (2018) pointed out that about 19 to 31 Tg N yr$^{-1}$ are mineralized from near-surface rocks. Their results also suggested that rock N inputs are particularly important in montane and high-latitude ecosystems. However, they also discussed the fact that at some places where rock weathering occurs deep beneath the soil, N from rocks may be released to groundwater and transported to fluvial systems. The ability of plants to use weathered N in deep soils depends on the depth of their roots and how much of the rock N inputs are actually used by plants was not quantified in their study. In another synthesis mentioned by the referee

(Augusto et al., 2017) in Comment 14, soil N availability was shown not significantly dependent on soil parent material or soil weathering stage. Augusto et al., (2017) argued that soil parent material and weathering stage only have indirect effects on soil N availability through its influence on edaphic factors such as soil pH. These evidences together show that rock weathering may contribute a significant amount of N to the soil, but the amount of these N are accessible to plants are still unknown and needed to be assessed.

Computationally, in our modelling framework, if fluxes of rock N inputs are taken into account, they should act as an input of N to non-agricultural ecosystems. Qualitatively, considering the computation of the model outputs (Appendix E) only the size of inorganic N is related to the total input of N to non-agricultural ecosystems (Eq. E17). Quantitatively, our estimate of the total input of N to non-agricultural ecosystems (188 Tg yr$^{-1}$) is at the higher end of the estimate by Houlton et al. (2018) even when they considered the rock N inputs (mean 147 Tg yr$^{-1}$, and range between 99.1 and 185.1 Tg yr$^{-1}$). Compared with estimates from Houlton et al. (2018), the fluxes from rock N inputs ignored in GOLUM-CNP are compensated by the larger estimate of N fixation and atmospheric N deposition used by GOLUM-CNP. In consequence, our estimates of total N inputs, and thus the associated model output (inorganic N) cover the uncertainty related to missing N inputs from rocks.

In the revised manuscript, we mention the recent paper by Houlton et al. (2018) on page 3 line 35: "… The N cycle includes a specific soil inorganic-N pool in addition to the five pools of the C cycle. The inputs of N to ecosystems include atmospheric N deposition and N fixation (Nd+Nfix in Fig. 1), both of which are assumed to enter the inorganic-N pool. We did not consider the flux of N mobilized from near-surface rocks, although a recent paper by Houlton et al. (2018) pointed out this flux may be an important N sources in montane and high-latitude ecosystems. The total N-fixation flux in this study includes both symbiotic and asymbiotic fixation…" In the discussion section on page 12 line 37, we add more discussions about it: "In addition, some processes, such as the N inputs from rock weathering (Houlton et al., 2018) were not considered in this study, because 1) as stated in Houlton et al. (2018), it is still unknown how much of rock-released N can be used by plants when rock weathering happens deep beneath the soils; 2) in GOLUM-CNP, adding rock N inputs has the same effect than N fixation and N deposition (Eqs. B6 and E17); and 3) the estimate of total input of N to ecosystems (188 Tg N yr-1) in this study are already at the higher end of the estimate (mean 147 Tg yr-1, and range between 99.1 and 185.1 Tg yr-1) of Houlton et al. (2018), even if rock N inputs are not accounted for, due to our larger estimates of N fixation and N deposition than Houlton et al. (2018). In the future, the rock N inputs and the fraction of these N inputs are accessible to plants should be further quantified and the quantity of total N inputs to the ecosystems should be reconciled between different studies. With these improvements, the future development of data-driven GOLUM-CNP should take into all these processes and fluxes."

12. Page 3, lines 37-38: N fixation is not entirely controlled by plants. In fact, the authors mentioned asymbiotic fixation a few lines above.

**Response:**
We agree with the referee. Biological N fixation occur in two ways: symbiotic and asymbiotic (or free-living). However, there are a diversity of the relationship between the

N$_2$-fixing organisms and plants and these is no clear line separating symbiotic and asymbiotic N fixiation and how these N can be used by plants and microbes (Reed et al., 2011). In this study, we do not separate the two and this is not a problem under steady state, since the inputs of N fixation are either used by plants, used by microbes (e.g. through immobilization) or even lost through leaching.

To address the concerns by the referee, we revised the manuscript: "The total N-fixation flux in this study includes both symbiotic and asymbiotic fixation, separately estimated from a previous study (see Sect. 3.1). We do not separate the two fixation processes and assume that they together contribute to the inorganic-N pool, although these two pathways of N fixation are differed in terms of the relationships between N$_2$-fixing microorganisms and plants. We did not consider the flux of N mobilized from near-surface rocks, although a recent paper by Houlton et al. (2018) pointed out this flux may be an important N sources in montane and high-latitude ecosystems. N uptake (F$_N$) by plants is assumed…"

13. Page 4, lines 21-22: much inorganic P is neither strongly sorbed, nor immediately available to plants. See, for example, Yang et al. (2013).
**Response:**
We agree with the referee. Please refer to revision of the manuscript in the response to Comment 10.

14. Page 5, lines 2-6: why the biome-scale analysis? Soil parent material is perhaps the most important factor governing nutrient limitation (Augusto et al. 2017) and has substantial sub-biome scale variability.
**Response:**
Firstly, Augusto et al. (2017) grouped their database into four climate classes (tropical, dry, temperate and cold) rather than different ecosystems/biomes, because "… generally low number of values per climate–ecosystem combination" in their database. And because of the limited number of data, "For a given climate class (e.g., temperate class), we found no statistical difference of nutrient limitation among ecosystem types (e.g., forests vs. grasslands)" – their analyses were only made for different ecosystems within their climate classes, but were not made for ecosystem across the globe.

In Augusto et al. (2017), they used three independent approaches and defined three indexes to address nutrient limitation: 1) plant growth response to nutrient addition (fertilization) (index 1 hereafter); 2) leaf chemistry including N:P ratios and nutrient resorption efficiency based on a leaf database containing values of N content and/or P content for green leaves and senescent leaves (index 2 hereafter); and 3) a model framework quantifying soil N and P in soils that are available to plants (index 3 hereafter). Of note is that the former two indices are defined from a vegetation perspective, while the third index is defined from a perspective of soil properties.

Augusto et al. (2017) found that N limitation (all three indexes) was best explained by climate (Fig. 3d-f and Sect. 3.2 in Augusto et al., 2017) and not directly linked to soil parent materials (Fig. S10g in Augusto et al., 2017) and weathering stage (Fig. S10a in Augusto et al., 2017). They found P-limited plant growth (index 1) was not directly linked to climate or latitude (Fig. 3h, j, k in Augusto et al., 2017) but partly related to P atmospheric deposition

(Sect 3.3 in Augusto et al., 2017). They found that soil P availability (index 3) could be "*partly*" explained by soil types and soil parent materials. Of note is that only one of the three indexes, and only for P, was attributed to soil properties in Augusto et al. (2017). In addition, even grouped into different classes of soil parent materials, there are substantial sub-classes variability related to soil properties (Fig. S10b, e, h in Augusto et al., 2017).

Augusto et al. (2017) further investigated the relative importance of actual N limitation versus P limitation and found soil parent materials appeared to be "*slightly*" more important than climate (Fig. 4c) based on the plant growth response to fertilization (index 1). However, when looking at plant N:P ratios, the relative limitation index of the plant leaf influenced by climate (+0.73) was more strongly than by soil parent material (-0.2) as shown by Fig. 4d in Augusto et al. (2017).

As shown above, there are different indexes for nutrient stresses, which differ with respect to the perspective (soil, individual plant, vegetation, ecosystem, etc). In the nutrient cycle, the vegetation and soil are tightly coupled: plants take nutrients from the soil for growing and return the nutrients to soil when they die. So all indexes have their use and a combination of indexes is the best to cover the big picture, like Augusto et al. (2017) did. In this study, we aimed to provide a dataset to evaluate the results of LSMs in their simulation of interactions between C, N and P cycling and LSMs mainly work at ecosystem scale. We focused on openness, nutrient use efficiencies and residence times in ecosystems. The openness is defined as the percentage of the total **plant** uptake of nutrients from new nutrient inputs (N fixation and N deposition for N; P deposition and release of P from rock weathering for P), and the nutrient use efficiencies are defined as the quotient between GPP and **plant** uptake of N and P from inorganic N and labile soil P. These indicators all work at ecosystem scale, so that we think it is more appropriate to made analysis at biome-scale rather than at soil scale in this study.

At last, other studies discussing about similar indicators also made analysis at biome scale. For example, Cleveland et al. (2013) discussed the patterns of new versus recycled nutrients, which is a similar concept as the openness defined in our study, for different biomes. The analysis of nutrient efficiencies made by Gill and Finzi (2016) were also made at the biome scale. We think, as already highlighted above, whether the analysis should be made at biome-scale or soil-type-scale should largely depend on what variables we are looking at.

To address the point raised by the referee and highlight the fact different studies define nutrient cycles at different scales, we revise the manuscript on page 5 line 2: "Different indices have been used to describe nutrient cycling from different perspectives (soil, individual plant, vegetation, ecosystem, etc) (Augusto et al., 2017; Cleveland et al., 2013; Gill and Finzi, 2016). In this study, we focused on the openness, nutrient use efficiencies and the residence time (Sect. 3.2) which are defined at ecosystem scale and thus correspond to the scale at which DGVMs are typically defined. For the presentation of results, we distinguish seven biomes: tropical rainforests (TRF), temperate deciduous forests (TEDF), …"

15. Page 5, lines 12-13: I am concerned at the ECMWF soil moisture product. Is it any good? Doesn't it have fixed layer depths? I'm not sure if that is appropriate here.

**Response:**

ERA-Interim/Land has four fixed layer depths: 0-0.07 m, 0.07-0.28 m, 0.28-1 m and

1-2.89 m. We admit that we do not have precise data set for the total soil content globally up till now, so that we only rely on some model results which were also constrained and evaluated against in situ measurements (Balsamo et al., 2015; Martens et al., 2017).

In our computation, the soil moisture impacts the estimate of inorganic N and labile soil P (Eqs. B7, E17 and E28). To address the question by the referee, we choose another widely used data set of root-zone soil moisture, GLEAM v3.2 (Martens et al., 2017). In GLEAM v3.2, the depth of the root zone is a function of the land-cover type and comprises three model layers for the fraction of tall vegetation (0–10, 10–100, and 100–250 cm), two for the fraction of low vegetation (0–10, 10–100 cm), and only one for the fraction of bare soil (0–10 cm). We multiplied the root-zone soil moisture by the depth of root zone derived from the biome map we showed in our study to estimate the total water content. To convert from the biome map to the classes defined in GLEAM v3.2, we assume forests biomes correspond to tall vegetation in GLEAM v3.2, grass biomes correspond to low vegetation and other land surfaces correspond to bare soil. The comparison between the resulting pool sizes of inorganic N and labile soil P using these two different data sets of soil moisture are shown in Fig. R1. It is not surprising that these two products have large uncertainties at grid scale, but the large-scale patterns of inorganic N and labile soil P using these two products are very similar.

To address the referee's concern, we revise the sentence on page 5, lines 12-13: "The spatially explicit estimate of total soil moisture was derived from the European Centre for Medium-Range Weather Forecasts (ECMWF) Interim Reanalysis (ERA-Interim/Land; Albergel et al., 2013; Balsamo et al., 2015). Gridded soil water content was provided in ERA-Interim/Land in four discretized layers until 2.89 m below ground, which is the soil depth we considered for the total soil moisture. Although some uncertainties exist at grid scale, the large-scale patterns in soil moisture from ERA-Interim/Land are consistent with other products (Rötzer et al., 2015), enabling us to use it to represent the large-scale spatial gradients in soil water moisture. The global gridded estimate of runoff data was obtained from the Global Runoff Data Centre (GRDC, http://www.grdc.sr.unh.edu/), …"

[Figure]

**Figure R1** Comparison between inorganic N (a, c) and labile soil P (b, d) using different products of soil water content.

16. Page 8, lines 3-4: There is not much take-away here. I guess it is not too surprising that the number falls within the large range. But what is the range of your computed number? Is the ranage reduced or increased compared to CARDAMOM?
**Response:**
This section used input of C (NPP) to ecosystems and the residence time of C pools to compute a steady-state C cycle. The aim of this computation is not to improve CARDAMOM results, but to justify that the steady-state C cycle transformed from CARDAMOM is within the range of our best knowledge about the C cycle corresponding to non-steady state present-day conditions (provided by CARDAMOM). This section justifies the use of steady state transformation to derive the N and P cycle in GOLUM-CNP. For instance, the pool sizes of N and P in vegetation are computed as the product of C pool sizes (computed based on steady-state assumption) and the stoichiometry ratios of N:C and P:C based on a mass-balance approach.

We revised this section and highlighted the justification of using this steady-state C cycle rather than the original CARDAMOM in our GOLUM-CNP.

17. Page 8, lines 16-19: All of this seems like speculation, more appropriate for a "discussion" section than for "Results".
**Response:**
Thanks for the suggestion. We revise the manuscript:
-    Sect. 4.1: "Table 2 shows the global C-pool sizes and main fluxes of the steady-state

C cycle transformed from CARDAMOM under the climate condition of 2001-2010, which are compared with the means and percentile ranges from the original non-steady-state CARDAMOM results. Although the steady-state C stocks do not exactly represent the C stocks at present day, the differences between the steady-state transformed pool sizes and the original non-steady-state CARDAMOM results were within 10% for most C pools and fluxes. The largest differences were for  biomass (foliar, fine-root and wood) pools. The larger foliar and fine-root pools in the steady-state GOLUM-CNP model were due to adjustments done for grass dominated grid cells for which CARDAMOM provided some wood growth inconsistent with the biome distribution used in GOLUM-CNP. In these cases, we allocated wood growth from CARDAMOM into growth of fine root and foliage. . These pools in GOLUM-CNP, however, remained within the [5, 95th] percentile range of the original CARDAMOM values. The pool size for global woody biomass was 37% smaller in the steady-state model (469 Pg) than the original CARDAMOM results but remained within its inter-quartile range (364-984 Pg). The differences between the gridded maps from original CARDAMOM and GOLUM-CNP are shown in Fig S1. The steady-state transformed C stocks in biomass, litter and SOM were within the 25[th] and 75[th] percentiles of the original CARDAMOM results at more than 90% of forest grid cells, indicating that our steady-state transformed C stocks are close to the actual C stocks at present day, given the large uncertainties in the state-of-the-art estimates. Due to the adjustment made for the grass dominated grid cells (see above), the C pools for grassland differ more strongly than forest-dominated area from the original CARDAMOM."

- Sect. 5.3 page 12 lines 35-37: "… Our results are a first step for evaluating global biogeochemical cycles. Although our steady-state C pool sizes (given the NPP and residence time at the condition of current climate) were within the [25, 75th] percentile range of the original non-steady-state CARDAMOM results (Fig. S1) at most grid cells, the biomass C stocks at 5%-10% of forest grid cells exceed the uncertainty range of CARDAMOM. In addition, independent remote-sensing estimates for 30˚N to 80˚N were 4.76 ± 1.78 kg C m-2 for mean forest C density and 79.8 ± 29.9 Pg C for total forest C (Thurner et al., 2014), which were lower than the GOLUM-CNP estimates (6.51 kg C m-2 for mean forest C density across pixels defined as forest in Fig. 2, and 181 Pg C for total forest C) for this region. This inconsistency was largely due to the fact that northern temperate and boreal forests may deviate substantially from their equilibrium for the current NPP (Pan et al., 2011), because of climate change and elevated CO2. Residual overestimation could be also due to the fact that biomass removal by harvesting and from disturbance other than fires was not explicitly constrained in CARDAMOM and thus not represented in GOLUM-CNP. A transient simulation of N and P cycling will be needed in future studies…"

18. Page 11, lines 4-18: There are also a fair amount of "results" in the "Discussion" section. Furthermore, these are some very ad hoc ways of making corrections when unexpected results

**Response:**

  We think that these corrections are empirical because they are not directly a "data-driven" product like CARDAMOM (although CARDAMOM lacks strong constraints for the C allocation fractions, especially at high latitudes). This part of discussion aims to provide an example how to identify the major contributor of the uncertainties in the model and how to qualitatively estimate the impact of some biased input data on the results. In this context, we think that this part is more appropriate for the "Discussion" results rather than the "Results" section.

  In order to clarify the logic of this section, we made the following revisions in the manuscript:

- Page 3 line 10-12: "… We first present the model structure (Sect. 2) and the data sets used to derive its outputs consisting of pools, fluxes and turnovers of C, N and P (Sect. 3). The model results and their sensitivities to the input observation-based data sets are then further analyzed in Sect. 4. In Sect. 5, we show examples of the application of this sensitivity analysis to identify the major differences in the results from our model framework and a synthesis of in situ measurements, and a qualitative example of how to compare the model and the independent data. These differences identify critical observations to reduce uncertainty in global C and nutrient cycling and highlight the future demand for model development, calibration and evaluation."

- Page 10 line 16: "… The data-driven estimates of steady-state global C, N and P cycles are the first that are fully consistent with a large set of global observation-based data sets, under the condition of current climate, deposition and CO2 concentration. The indicators for the coupling between nutrient and C cycling, which are emerging properties of GOLUM-CNP, are used to evaluate the capabilities of GOLUM-CNP to capture observed patterns among biomes. We found that there are some differences between our data-driven estimates and previous studies about the nutrient efficiencies at biome scales (Gill and Finzi, 2016) and the openness (Cleveland et al., 2013). In this section, we discussed the major uncertainties in our model and show how these uncertainties affect the computation of nutrient efficiencies and the openness (Sect. 5.1). Of note is that most of our discussions are for the C cycle (based on the sensitivity analysis, see below), and since CARDAMOM is the only data-driven C cycle to our knowledge, the modifications of the CARDAMOM dataset we made in this section are more qualitative and diagnostic rather than deterministic. Such an example highlights some important variables that should be investigated or considered in future data-driven products."

**Response:**

  We revise the sentence: "Our estimates of P openness were also larger than those of Cleveland et al. (2013), which  we attributed to the large differences in the estimates of P deposition between the two studies: Cleveland et al. (2013) used  P deposition (0.26

Tg yr$^{-1}$) from Mahowald et al. (2008), which were  an order of magnitude lower than  recent estimates from Wang et al. (2017)  used in this study (5.8 Tg yr$^{-1}$),  because  Wang et al. (2017) revised the contribution of anthropogenic P emissions and  P in particles with diameters >10 μm (Wang et al., 2015, 2017). …"

**References:**

Augusto, L., Achat, D. L., Jonard, M., Vidal, D. and Ringeval, B.: Soil parent material—A major driver of plant nutrient limitations in terrestrial ecosystems, Global Change Biology, 23(9), 3808–3824, doi:10.1111/gcb.13691, 2017.

Balsamo, G., Albergel, C., Beljaars, A., Boussetta, S., Brun, E., Cloke, H., Dee, D., Dutra, E., Muñoz-Sabater, J., Pappenberger, F. and others: ERA-Interim/Land: a global land surface reanalysis data set, Hydrol. Earth Syst. Sci., 19(1), 389–407, 2015.

Cleveland, C. C., Houlton, B. Z., Smith, W. K., Marklein, A. R., Reed, S. C., Parton, W., Grosso, S. J. D. and Running, S. W.: Patterns of new versus recycled primary production in the terrestrial biosphere, Proc. Natl. Acad. Sci., 110(31), 12733–12737, doi:10.1073/pnas.1302768110, 2013.

Cole, C. V., Innis, G. S. and Stewart, J. W. B.: Simulation of Phosphorus Cycling in Semiarid Grasslands, Ecology, 58(1), 1–15, doi:10.2307/1935104, 1977.

Gill, A. L. and Finzi, A. C.: Belowground carbon flux links biogeochemical cycles and resource-use efficiency at the global scale, Ecol Lett, 19(12), 1419–1428, doi:10.1111/ele.12690, 2016.

Goll, D. S., Brovkin, V., Parida, B. R., Reick, C. H., Kattge, J., Reich, P. B., van Bodegom, P. M. and Niinemets, Ü.: Nutrient limitation reduces land carbon uptake in simulations with a model of combined carbon, nitrogen and phosphorus cycling, Biogeosciences, 9, 3547–3569, doi:10.5194/bg-9-3547-2012, 2012.

Houlton, B. Z., Morford, S. L. and Dahlgren, R. A.: Convergent evidence for widespread rock nitrogen sources in Earth's surface environment, Science, 360(6384), 58–62, doi:10.1126/science.aan4399, 2018.

Jonard, M., Fürst, A., Verstraeten, A., Thimonier, A., Timmermann, V., Potočić, N., Waldner, P., Benham, S., Hansen, K., Meriläi, P., Ponette, Q., Cruz, A. C. de la, Roskams, P., Nicolas, M., Croisé, L., Ingerslev, M., Matteucci, G., Decinti, B., Bascietto, M. and Rautio, P.: Tree mineral nutrition is deteriorating in Europe, Global Change Biology, 21(1), 418–430, doi:10.1111/gcb.12657, 2015.

Lawrence, D. M., Oleson, K. W., Flanner, M. G., Thornton, P. E., Swenson, S. C., Lawrence, P. J., Zeng, X., Yang, Z.-L., Levis, S., Sakaguchi, K., Bonan, G. B. and Slater, A. G.: Parameterization improvements and functional and structural advances in Version 4 of the Community Land Model, Journal of Advances in Modeling Earth Systems, 3(1), doi:10.1029/2011MS00045, 2011.

Luo, Y. and Weng, E.: Dynamic disequilibrium of the terrestrial carbon cycle under global change, Trends in Ecology & Evolution, 26(2), 96–104, 2011.

Martens, B., Miralles, D. G., Lievens, H., van der Schalie, R., de Jeu, R. A. M., Fernández-Prieto, D., Beck, H. E., Dorigo, W. A. and Verhoest, N. E. C.: GLEAM v3: satellite-based land evaporation and root-zone soil moisture, Geosci. Model Dev., 10(5), 1903–1925, doi:10.5194/gmd-10-1903-2017, 2017.

Mayor, J. R., Wright, S. J. and Turner, B. L.: Species-specific responses of foliar nutrients to long-term nitrogen and phosphorus additions in a lowland tropical forest, Journal of Ecology, 102(1), 36–44, doi:10.1111/1365-2745.12190, 2014.

Reed, S. C., Cleveland, C. C. and Townsend, A. R.: Functional Ecology of Free-Living Nitrogen Fixation: A Contemporary Perspective, Annu. Rev. Ecol. Evol. Syst., 42(1), 489–512, doi:10.1146/annurev-ecolsys-102710-145034, 2011.

Sardans, J. and Peñuelas, J.: The role of plants in the effects of global change on nutrient availability and stoichiometry in the plant-soil system, Plant Physiology, pp.112.208785, doi:10.1104/pp.112.208785, 2012.

Sardans, J., Rivas-Ubach, A. and Peñuelas, J.: The C:N:P stoichiometry of organisms and ecosystems in a changing world: A review and perspectives, Perspectives in Plant Ecology, Evolution and Systematics, 14(1), 33–47, doi:10.1016/j.ppees.2011.08.002, 2012.

Sardans, J., Alonso, R., Janssens, I. A., Carnicer, J., Vereseglou, S., Rillig, M. C., Fernández‐Martí

nez, M., Sanders, T. G. M. and Peñuelas, J.: Foliar and soil concentrations and stoichiometry of nitrogen and phosphorous across European Pinus sylvestris forests: relationships with climate, N deposition and tree growth, Functional Ecology, 30(5), 676–689, doi:10.1111/1365-2435.12541, 2016.

Sardans, J., Grau, O., Chen, H. Y. H., Janssens, I. A., Ciais, P., Piao, S. and Peñuelas, J.: Changes in nutrient concentrations of leaves and roots in response to global change factors, Global Change Biology, 23(9), 3849–3856, doi:10.1111/gcb.13721, 2017.

Sistla, S. A. and Schimel, J. P.: Stoichiometric flexibility as a regulator of carbon and nutrient cycling in terrestrial ecosystems under change, New Phytologist, 196(1), 68–78, doi:10.1111/j.1469-8137.2012.04234.x, 2012.

Thornton, P. E., Lamarque, J.-F., Rosenbloom, N. A. and Mahowald, N. M.: Influence of carbon-nitrogen cycle coupling on land model response to $CO_2$ fertilization and climate variability, Glob. Biogeochem. Cycles, 21(4), GB4018, doi:10.1029/2006GB002868, 2007.

Wang, R., Goll, D., Balkanski, Y., Hauglustaine, D., Boucher, O., Ciais, P., Janssens, I., Penuelas, J., Guenet, B., Sardans, J. and others: Global forest carbon uptake due to nitrogen and phosphorus deposition from 1850 to 2100, Glob. Change Biol., 2017.

Wang, Y.-P., Houlton, B. Z. and Field, C. B.: A model of biogeochemical cycles of carbon, nitrogen, and phosphorus including symbiotic nitrogen fixation and phosphatase production, Global Biogeochem. Cycles, 21(1), GB1018, doi:10.1029/2006GB002797, 2007.

Wang, Y. P., Law, R. M. and Pak, B.: A global model of carbon, nitrogen and phosphorus cycles for the terrestrial biosphere, Biogeosciences, 7(7), 2010.

Xu-Ri and Prentice, I. C.: Terrestrial nitrogen cycle simulation with a dynamic global vegetation model, Glob. Change Biol., 14(8), 1745–1764, doi:10.1111/j.1365-2486.2008.01625.x, 2008.

Yang, X., Wittig, V., Jain, A. K. and Post, W.: Integration of nitrogen cycle dynamics into the Integrated Science Assessment Model for the study of terrestrial ecosystem responses to global change, Glob. Biogeochem. Cycles, 23(4), 2009.

Yang, X. and Post, W. M.: Phosphorus transformations as a function of pedogenesis: A synthesis of soil phosphorus data using Hedley fractionation method, Biogeosciences, 8(10), 2011.

Yang, Y., Fang, J., Ji, C., Datta, A., Li, P., Ma, W., Mohammat, A., Shen, H., Hu, H., Knapp, B. O. and Smith, P.: Stoichiometric shifts in surface soils over broad geographical scales: evidence from China's grasslands, Global Ecology and Biogeography, 23(8), 947–955, doi:10.1111/geb.12175, 2014b.

Yuan, Z. Y. and Chen, H. Y. H.: Decoupling of nitrogen and phosphorus in terrestrial plants associated with global changes, Nature Climate Change, 5(5), 465–469, doi:10.1038/nclimate2549, 2015.

Zaehle, S., Friend, A. D., Friedlingstein, P., Dentener, F., Peylin, P. and Schulz, M.: Carbon and nitrogen cycle dynamics in the O-CN land surface model: 2. Role of the nitrogen cycle in the historical terrestrial carbon balance, Global Biogeochemical Cycles, 24(1), doi:10.1029/2009GB003522, 2010.

Zaehle, S.: Terrestrial nitrogen–carbon cycle interactions at the global scale, Philos. Trans. R. Soc. Lond. B Biol. Sci., 368(1621), 20130125, doi:10.1098/rstb.2013.0125, 2013.

---

## Author Response (AR2)

**Response to comments on "GOLUM-CNP v1.0: a data-driven modeling of carbon, nitrogen and phosphorus cycles in major terrestrial biomes" by Y. Wang et al.**

We thank the referee for reviewing our manuscript. Please find attached a point-by point reply to each of the comments raised by the referee with legible text and figures organized along the text.

All the main points have been adequately addressed, though some smaller issues remain.

**Response:**

We would like to thank the referee for the valuable comments and suggestions for improving our manuscript. Following the reviewer's comments, we carefully revised our manuscript. Please find below the point-to-point responses (in black) to all referee comments (in blue). For your convenience, changes in the revised manuscript are highlighted with dark red. All the pages and line numbers correspond to the original version of text.

The authors have significantly improved the explanation of what GOLUM is. However, it is still not sufficiently clear. On pages 2/3 where it is explained, no-where does it say that GOLUM is an offline component of CARDOMON - which is what it is if, as it seems from the description, it has to be run with CARDOMON output. Being an offline component is fine, you don't have to hide what the model is.

**Response:**

The GOLUM-CNP model is not an offline component of CARDAMOM. The GOLUM-CNP has a similar model structure as CARDAMOM C cycle for describing the different pools in the ecosystems (leaf, fine roots, wood, litter and SOM). In this study, GOLUM-CNP used CARDAMOM output because CARDAMOM is the only global data-driven C model that combined multiple observations of C pools and fluxes (satellite observations of leaf area and biomass, and global soil carbon data). GOLUM-CNP can also use the outputs from other observation-based C models if they have similar model structures. In this sense, GOLUM-CNP is not an offline component of any model, but an independent diagnostic model that combines multiple observation-based datasets.

The commenting of the code is much improved. The file names, "anal" etc. seem to be an abbreviation of "analysis". I suggest you choose another abbreviation, as the current one is a bit tasteless.

**Response:**

Thanks for the suggestion. We use Globe_SteadyState.py and SteadyState_pixel.py instead.

Page 3, line 4 The phrasing here suggests that GOLOM is providing "a new dataset", as if it were observations. It's not. It's a model. The second 'goal' is not achievable and it is misleading

**Response:**

We change the word "dataset" to "observation-based estimates".

We think the second goal "provide new observation-based estimates that can be used to evaluate the results of global terrestrial biosphere models with consistent state of C, N and P cycles" is feasible because other DGVMs can simulate the steady-state under present day conditions. For example, Wang et al. (2010) have shown their estimates of steady-state pool sizes and fluxes for 1990's; CLM4CN has a specific equilibrium simulation with vegetation, $CO_2$, aerosol and nitrogen deposition data for the year 2000. Direct comparison between the results from these model and those from GOLUM-CNP is thus possible. In addition, these models were usually validated against some in-situ observations or subsets of some variables in the C, N and P cycles but were quite hard to evaluate *consistently* against multiple datasets with global coverage. In this sense, this study provides new estimates of all the pool sizes and fluxes that are consistent with available datasets. Considering these two aspects, we would like to highlight that this is the most important asset of this study.

The authors justify their approach by saying that Cleveland et al. (2013) used the same technique, but they use a different nomenclature (Ix rather than NPPnew). Introducing different nomenclature for the same concept is deeply unhelpful for the community and disrespectful of prior research. I understand that the authors will now be reluctant to change their nomenclature, though I think that is what they ought to do. At the very least, a brief note around page 7, line 13, saying that this is equivalent to Cleveland's NPPnew, is essential.

**Response:**

In Cleveland et al. (2013), they defined "the new terrestrial NPP" as the ratio between "the NPP fueled by new nutrient inputs" and "the total NPP nutrient demand". Their definition is from the perspective of C cycle (NPP). In this study, we defined the "openness" from the perspective of nutrient cycles, as "the ratio of nutrients inputs (Ix, $X \in \{N,P\}$ over the amount of nutrients in production".

In addition, although the general principle that the ratio between the "new" fluxes and "total" fluxes are the same in Cleveland et al. (2013) and in this study, the practical computations are not (see below the changes in the supplementary for detail). In Cleveland et al. (2013), they had $C_j$ (NPP allocation fraction in their study) in the parentheses of their Eq. S7, which means the allocation fractions of new N in different vegetation pools are the same as the C allocation fractions. This assumption is neither right nor wrong, but lack evidences. In this study, we define the openness only based on the nutrient fluxes (equilibrium states) rather than converting them to C fluxes like what Cleveland et al. (2013) did. As long as indexes are precisely defined as they are in each study, one can be calculated from the other and this is not a source of confusion.

Our definition of the "openness" is thus not the same index than the "new NPP" of Cleveland et al. (2013). To make this point clearer, we revised our manuscript:

- Page 7 line 15: "… We calculated some ecologically relevant quantities from the GOLUM-CNP output. We defined the openness of N and P cycles as NO and PO that were calculated as the ratio of nutrients inputs (Ix, $X \in \{N,P\}$, …"

- We add in page 7 line 20: "The openness quantifies how much nutrients is from external inputs, which is similar but not strictly equal to the "proportion of new NPP fueled by new nutrient inputs". In general, the "openness" used this study and the "proportion of new NPP fueled by new nutrient inputs" in Cleveland et al. (2013) both quantify the ratios between fluxes that are related to external inputs and the "total" fluxes, but "openness" used in this study was defined from nutrient cycles, while the index used in Cleveland et al. (2013) was defined from NPP-carbon. In addition, the practical computation of the openness in this study are slightly different from that of Cleveland et al. (2013), which was quantitatively compared in the supplementary material."
- In the supplementary, we add a paragraph that compares the computation of "openness" used in this study and that of the "proportion of new NPP fueled by new nutrient inputs" used in Cleveland et al. (2013):

"**S1 Comparison of the "openness" in this study and in Cleveland et al. (2013)**

In this section, we compare the computation of "openness" index used in this study versus "proportion of new NPP fueled by new nutrient inputs" used in Cleveland et al. (2013). We take the indexes for N as an example, but the computation of corresponding indexes for P are similar.

Assume NPP is allocated to leaf, wood and roots by 0.5, 0.3 and 0.2, those pools with C:N ratios of 25, 150 and 50. In Cleveland's index, N is allocated in fractions of 0.5, 0.3 and 0.2 in NPP-N ($C_j$ can be moved before $CtoN_j$ in their Eq. S7). The new NPP is $(0.5*25 + 0.3*150 + 0.2*50) * I_N = 67.5\ I_N$, where $I_N$ represent external nutrients inputs, i.e. the sum of deposition and biological fixation for N which is available to vegetation. They assume that the remaining NPP must be totally fueled by resorption and net mineralization in the soil. They modelled the amount of N resorption but computed the amount of mineralization as the difference between nutrient demand, new nutrient inputs, and nutrient resorption (their Eqs. S5). So the "proportion of new NPP fueled by new nutrient inputs" index in Cleveland et al. (2013) equals $67.5\ I_N/NPP$.

In this study, the N allocation is 0.5/25 : 0.3/150 : 0.2/50. Normalizing the values to ensure the sum of the fractions equals to 1 gives allocation fractions of N in NPP-N of 0.77 : 0.08 : 0.15. Under steady state, we have the relationship that the total N demand = NPP*0.5/25+NPP*0.3/150+NPP*0.2/50=$F_N$+$RSB_N$, where $F_N$ is the uptake from inorganic N soil pool and $RSB_N$ is the flux of resorbed N. As a result, the openness index NO = $I_N / (F_N + RSB_N) = 38.5\ I_N/NPP$.

Because there is no evidence that how much the new N inputs is allocated in the vegetation, we chose to define the openness index only based on N and P fluxes rather than to convert N and P fluxes into NPP like Cleveland et al. (2013) did."

There is no figure "R1" included. I assume the authors mean the figure labelled R2. R2 is very substantially better than Figure 3 in the paper. It's a huge improvement and I suggest the authors consider their readers and use it in the main paper.

**Response:**

We think we depict the full C, N and P cycles including the pool sizes, residence times of different pools and fluxes with the original flow charts, which is not achievable by bar plots.

All the co-authors, who also have a wide range of expertise (measurements, models, etc.) in the community of global biochemical cycles, prefer the flow charts than the bar plots and we think that the flow charts contain critical information about how the different variables in the C, N and P cycles are linked.

To facilitate the readers, we put both figures in the main text (Figure R2 was put in the text as Figure 4). We re-ordered all the figures and change the texts that refer to different figures.

**Table S1** C:N, C:P and N:P molar (atomic) ratios across major biomes from Zechmeister-Boltenstern et al. (2015). Targeted biomes are: tropical rain forests (TRF), temperate deciduous forests (TEDF), temperate coniferous forests (TECF), boreal coniferous forests (BOCF), tundra (TUN), tropical/C4 grasslands (TRG), and temperate/C3 grasslands (TEG). Note that N:P ratios in Zechmeister-Boltenstern et al. (2015) are not exactly equal to the ratio of C:P to C:N, but the differences are small.

|  |  | TRF | TEDF | TECF | BOCF | TRG | TEG | TUN |
|---|---|---|---|---|---|---|---|---|
| C:N | Foliage | 25 | 25 | 59 | 49 | 39 | 25 | 49 |
|  | Root | 47 | 59 | 67 | 57 | 39 | 88 | 54 |
|  | Wood | 148 | 471 | 844 | 525 | -- | -- | -- |
|  | Soil | 16 | 19 | 20 | 32 | 25 | 10 | 13 |
| C:P | Foliage | 1027 | 867 | 1232 | 1049 | 753 | 1278 | 2167 |
|  | Root | 3125 | 1962 | 1186 | 1574 | 1300 | 2829 | 1300 |
|  | Wood | 13574 | 11179 | 24297 | 19734 | -- | -- | -- |
|  | Soil | 159 | 366 | 302 | 960 | 509 | 130 | 138 |
| N:P | Foliage | 43 | 36 | 23 | 23 | 20 | 53 | 45 |
|  | Root | 52 | 22 | 18 | 30 | 32 | 27 | 20 |
|  | Wood | 93 | 24 | 29 | 38 | -- | -- | -- |
|  | Soil | 13 | 20 | 15 | 31 | 31 | -- | 11 |

**Table S2** Mean residence time of C, N and P in ecosystems (unit: years). Targeted biomes are:

tropical rain forests (TRF), temperate deciduous forests (TEDF), temperate coniferous forests (TECF), boreal coniferous forests (BOCF), tundra (TUN), tropical/C4 grasslands (TRG), and temperate/C3 grasslands (TEG).

|   | TRF | TEDF | TECF | BOCF | TRG | TEG | TUN | Globe |
|---|-----|------|------|------|-----|-----|-----|-------|
| C | 29 | 49 | 48 | 106 | 40 | 67 | 101 | 38 |
| N | 382 | 1016 | 637 | 4834 | 987 | 3075 | 7896 | 1586 |
| P | 2520 | 3263 | 4413 | 6167 | 4483 | 6291 | 7077 | 6092 |

[Figure]

**Fig. S1** Comparison between C pool sizes of transformed steady-state C cycle and original CARDAMOM results.

[Figure]

[Figure]

**Fig. S2** Global external N inputs. a) N deposition from Wang et al. (2017). b) N fixation from Peng et al. (submitted)

[Figure]

[Figure]

**Fig. S3** Global external P inputs. a) P deposition from Wang et al. (2017). b) Release of P from rock weathering from Hartmann et al. (2014)

[Figure]

**Figure S4** Pool sizes and fluxes of C (black), N (green) and P (yellow) computed from GOLUM-CNP. The targeted biomes are tropical rainforests (TRF, a), temperate deciduous forests (TEDF, b), temperate coniferous forests (TECF, c), boreal coniferous forests (BOCF, d), tropical/C4 grasslands (TRG, e), temperate/C3 grasslands (TEG, f) and tundra (TUN, g).

[Figure]

[Figure]

**Fig.**  S4 Global nutrient openness computed from GOLUM-CNP. a) Openness of N. b) Opennes of P.

[Figure]

**Fig.**  S5 Global residence times of nutrients in the ecosystems. a) Residence times of N in the ecosystems. b) Residence times of P in the ecosystems

[Figure]

[Figure]

**Fig.**  S6 Global nutrient use efficiencies (the nutrient uptake by plants divided by GPP). a) N use efficiency. b) P use efficiency.

[Figure]

**Figure  S7** Mean sensitivity of the estimates of rates of nutrient uptake, inorganic nutrients, nutrient-use efficiencies, openness, turnover time of nutrients in the ecosystem and turnover time of nutrients in plants to the input variables for temperate deciduous forests.

[Figure]

**Figure**  S8 Mean **s**ensitivity of the estimates of rates of nutrient uptake, inorganic nutrients, nutrient-use efficiencies, openness, turnover time of nutrients in the ecosystem and turnover time of nutrients in plants to the input variables for temperate coniferous forests.

[Figure]

**Figure**  **S9** Mean sensitivity of the estimates of rates of nutrient uptake, inorganic nutrients, nutrient-use efficiencies, openness, turnover time of nutrients in the ecosystem and turnover time of nutrients in plants to the input variables for boreal coniferous forests.

[Figure]

**Figure**  **S10** Mean **s**ensitivity of the estimates of rates of nutrient uptake, inorganic nutrients, nutrient-use efficiencies, openness, turnover time of nutrients in the ecosystem and turnover time of nutrients in plants to the input variables for tropical/C4 grasslands.

[Figure]

**Figure**  **S11** Mean **s**ensitivity of the estimates of rates of nutrient uptake, inorganic nutrients, nutrient-use efficiencies, openness, turnover time of nutrients in the ecosystem and turnover time of nutrients in plants to the input variables for temperate/C3 grasslands.

[Figure]

**Figure**  **S12** Mean sensitivity of the estimates of rates of nutrient uptake, inorganic nutrients, nutrient-use efficiencies, openness, turnover time of nutrients in the ecosystem and turnover time of nutrients in plants to the input variables for tundra.

[Figure]

**Figure**  S13 NPP allocation fractions to woody biomass in original CARDAMOM (a) and adjusted (see Sect. 5.1) carbon cycle model (b).

[Figure]

**Figure**  S14 NPP allocation fractions to fine roots in original CARDAMOM (a) and adjusted (see Sect. 5.1) carbon cycle model (b).

[Figure]

**Figure**  S15 Leaf longevity in original CARDAMOM (a) and adjusted (see Sect. 5.1) carbon cycle model (b).

[Figure]

**Figure**  **S16** NPP allocation fractions to foliage in original CARDAMOM (a) and adjusted (see Sect. 5.1) carbon cycle model (b).